# Softmax is not Enough (for Adaptive Conformal Classification)

**Navid Akhavan Attar, Hesam Asadollahzadeh, Ling Luo & Uwe Aickelin**
School of Computing and Information Systems, Faculty of Engineering and IT
The University of Melbourne, Victoria, Australia
`{n.akhavanattar,h.asadollahzadeh,ling.luo,uwe.aickelin}@unimelb.edu.au`

## Abstract

The merit of Conformal Prediction (CP), as a distribution-free framework for uncertainty quantification, depends on generating prediction sets that are efficient, reflected in small average set sizes, while adaptive, meaning they signal uncertainty by varying in size according to input difficulty. A central limitation for deep conformal classifiers is that the nonconformity scores are derived from softmax outputs, which can be unreliable indicators of how certain the model truly is about a given input, sometimes leading to overconfident misclassifications or undue hesitation. In this work, we argue that this unreliability can be inherited by the prediction sets generated by CP, limiting their capacity for adaptiveness. We propose a new approach that leverages information from the pre-softmax logit space, using the Helmholtz Free Energy as a measure of model uncertainty and sample difficulty. By reweighting nonconformity scores with a monotonic transformation of the energy score of each sample, we improve their sensitivity to input difficulty. Our experiments with four state-of-the-art score functions on multiple datasets and deep architectures show that this energy-based enhancement improves the adaptiveness of the prediction sets, leading to a notable increase in both efficiency and adaptiveness compared to baseline nonconformity scores, without introducing any post-hoc complexity.

## 1 Introduction

Deploying machine learning models in critical, real-world applications requires not just high accuracy, but also trustworthy uncertainty quantification. Conformal Prediction (CP) has emerged as an effective framework for this challenge (Vovk et al., 2005). It provides a model-agnostic method to construct prediction sets, $C(X)$, that are guaranteed to contain the true class, $Y$, with a user-specified probability:

$$P(Y \in C(X)) \geq 1 - \alpha.$$

This distribution-free guarantee is a significant asset. However, the practical utility of CP depends on the characteristics of these prediction sets. Ideally, they should be **adaptive** and **efficient**: small for inputs that the model finds easy, and appropriately larger for inputs that are difficult or ambiguous.

This adaptiveness is governed by the nonconformity score. While many nonconformity scores are designed to produce adaptive sets, they are typically derived from a model's final softmax probabilities. This choice inherits a fundamental weakness, as softmax outputs are often unreliable indicators of a model's true uncertainty. They can exhibit overconfidence even for misclassified or out-of-distribution (OOD) inputs. Post-hoc calibration helps reduce this issue, but only to a limited extent, as it cannot fully correct the underlying limitations in uncertainty quantification. (Guo et al., 2017; Lee et al., 2018a; Hein et al., 2019). Consequently, the adaptiveness of these scores is by design limited, which can lead to inefficiently large sets for simple inputs, or misleadingly small sets for difficult ones.

One approach to improve adaptiveness involves adjusting the score based on an input-specific measure of difficulty, such as the variance of ensemble predictions, the error predicted by an auxiliary model (Hernández-Hernández et al., 2022), or the variance estimated via Monte-Carlo dropout (MCD) with a neural network (Cortés-Ciriano & Bender, 2019). This principle is related to Normalized Conformal Prediction, which has been shown to produce tighter and more informative sets

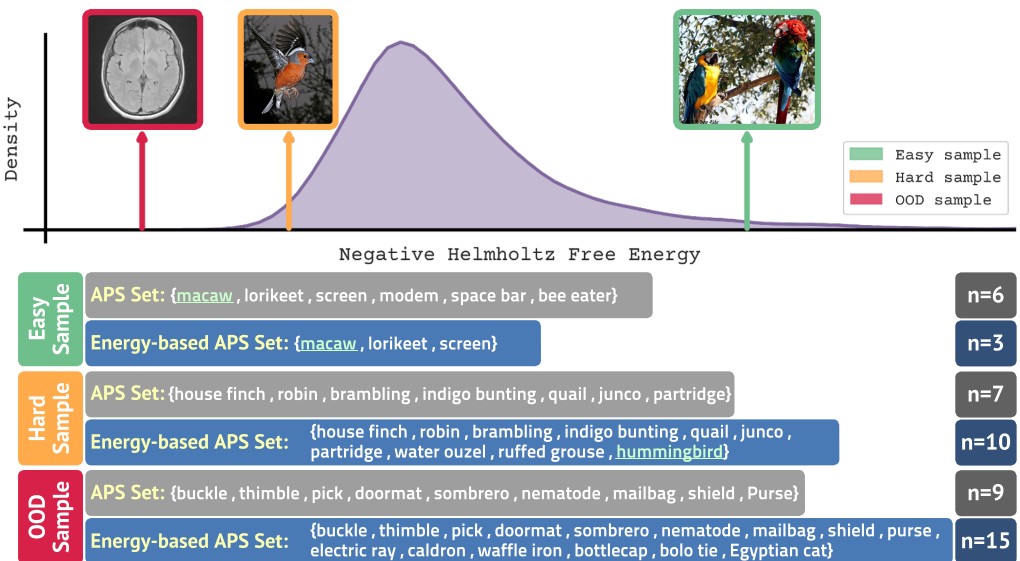

Figure 1: Prediction sets from a standard method (APS Romano et al. (2020)) versus our energy-based variant, demonstrating improved adaptiveness on ImageNet. (i) For an easy input like the image of a *Macaw*, whose clear visual cues (vivid colors, long tail) make it simple to classify, our energy-based method produces a smaller, more efficient set. (ii) For a hard input, a bird image labeled as *Hummingbird*, its appearance deviates from typical hummingbirds (e.g., a thicker, less tapered beak) and shares features with other bird classes, making the image difficult for the model. In this case, the energy-based method returns a larger prediction set, signaling higher uncertainty. (iii) Finally, for an out-of-distribution (OOD) input like a brain MRI that the model wasn't trained on, our method generates a much larger set, warning the user that the prediction is unreliable. This improvement in adaptive behavior is guided by the Helmholtz free energy, which captures the model's uncertainty about an input.

in regression by scaling a base score with an uncertainty estimate (Papadopoulos et al., 2002a; Papadopoulos & Haralambous, 2011; Boström et al., 2017). Building on this principle, we argue that a reliable signal for sample difficulty and model uncertainty exists in the pre-softmax logit space. We propose using the *Helmholtz free energy* computed from the logits, as a principled measure of a model's familiarity with an input. Inputs aligned with the training data distribution are assigned low energy (high certainty), while atypical or ambiguous inputs receive high energy (low certainty).

The energy signal is then incorporated into the conformal framework by applying per-sample reweighting of the nonconformity scores. For "easy" inputs where the model is certain, the energy-based term magnifies the base score, yielding smaller, more efficient prediction sets. For "hard" or OOD inputs, this term dampens the score, producing larger sets that signal the model's uncertainty. This improved adaptiveness is exemplified in Figure 1. Compared to existing adaptive scores, our proposed energy-based variants, by leveraging Helmholtz free energy derived from the pre-softmax logit space for per-sample reweighting of nonconformity scores, increase both adaptiveness and efficiency. This approach improves prediction sets across state-of-the-art score functions, while preserving the theoretical coverage guarantees of Conformal Prediction.

We summarize our contributions as follows:

- We provide a theoretical and empirical motivation for moving beyond softmax-based scores. We establish the connection between Helmholtz free energy and model uncertainty and demonstrate that this energy signal distinguishes sample difficulty more effectively than standard softmax metrics.

- We introduce a general framework of **Energy-Based Nonconformity Scores**, which modulates a base nonconformity score with the free energy of each sample to create more adaptive prediction sets that are smaller for easy inputs and larger for difficult or out-of-distribution inputs.

- We provide theoretical and empirical evidence showing that our energy-based enhancement improves the efficiency and adaptiveness of prediction sets for multiple deep learning architectures across a range of different scenarios.

## 2 MOTIVATION AND METHOD

In this section, we (i) explain why the softmax probabilities that underpin conventional nonconformity scores for deep classifiers are often unreliable for efficiently capturing model uncertainty, (ii) introduce Helmholtz free energy as a more robust measure of uncertainty derived directly from model logits, and (iii) use this concept to motivate and define a new class of energy-aware scores that produce more adaptive and efficient prediction sets. All notations are summarized in Appendix A.

### 2.1 SOFTMAX UNRELIABILITY AND IMPLICATIONS FOR CONFORMAL PREDICTION

Given logits $\mathbf{f}(x)$, a calibrated softmax with temperature $T > 0$ is

$$\hat{\pi}(y \mid x) = \operatorname*{softmax}_{y}\left(\frac{\mathbf{f}(x)}{T}\right) = \frac{\exp[f_y(x)/T]}{\sum_{k=1}^{K} \exp[f_k(x)/T]}. \tag{1}$$

The common nonconformity scores for classification are functions of $\hat{\pi}$ (as detailed in Appendix E.2). However, relying on softmax values alone is unreliable for uncertainty assessment. First, modern networks produce poorly calibrated and often overconfident posteriors (Guo et al., 2017), including spuriously high confidence on unrecognizable inputs (Nguyen et al., 2015). While temperature scaling can improve in-distribution calibration, it does not address epistemic uncertainty: OOD, "far" or even "hard" inputs may still map to representation regions that yield confident softmax outputs (Hein et al., 2019; Lee et al., 2018a). This sensitivity to representation geometry means that when class manifolds overlap or decision boundaries are poorly separated, softmax confidence can be misleading even after calibration (Cohen et al., 2020). Second, softmax posteriors entangle likelihoods with learned class priors, biasing scores under label shift or class imbalance. Margins for minority classes tend to be smaller, intensify uncertainty mis-estimation unless logits are explicitly adjusted (Ren et al., 2020). Collectively, these issues undermine CP adaptiveness: probability-based scores can produce (i) unnecessarily large sets for easy samples when tails are inflated, or (ii) deceptively small sets on ambiguous/OOD inputs that happen to receive high softmax confidence. For a comprehensive compilation of softmax criticism, see Appendix C.

These observations motivate adjusting nonconformity scores with an additional signal that reflects the model's holistic signal about Its familiarity with $x$. In the next subsection, we use the *Helmholtz free energy* computed from the logits as a principled, model-aware measure of epistemic uncertainty, that also correlates with sample difficulty, assigning low energy to easy in-distribution inputs and high energy to hard, ambiguous, or OOD inputs.

### 2.2 FREE ENERGY AS A MEASURE OF EPISTEMIC UNCERTAINTY

To quantify a model's uncertainty in its predictions, we seek a measure that reflects its familiarity with the input data. We turn to the framework of Energy-Based Models (EBMs) (LeCun et al., 2006). An EBM defines a scalar *energy* for every configuration of variables, where lower energy corresponds to higher probability. Any standard discriminative classifier can be interpreted through the lens of an EBM (Grathwohl et al., 2020). We refer to Appendix D for more details on EBMs.

For a classifier with a logit function $f(x) : \mathbb{R}^D \to \mathbb{R}^K$, we can define a joint energy function over inputs $x$ and labels $y$ as:

$$E(x, y; f) = -f_y(x), \quad y \in \{1, \dots, K\}. \tag{2}$$

This formulation connects the classifier's outputs directly to an energy landscape. The conditional probability $p(y|x)$ is then given by the Gibbs-Boltzmann distribution:

$$p(y|x) = \frac{\exp(-E(x, y))}{\sum_{k=1}^{K} \exp(-E(x, k))} = \frac{\exp(f_y(x))}{\sum_{k=1}^{K} \exp(f_k(x))}, \tag{3}$$

which is identical to the standard softmax function.

By marginalizing over the labels, we can derive an unnormalized density over the input space. This process yields the *Helmholtz free energy*, $F(x)$, which acts as the energy function for the marginal distribution $p(x)$:

$$F(x; f) = -\tau \log \sum_{k=1}^{K} \exp\left(\frac{-E(x, k)}{\tau}\right) = -\tau \log \sum_{k=1}^{K} \exp\left(\frac{f_k(x)}{\tau}\right), \tag{4}$$

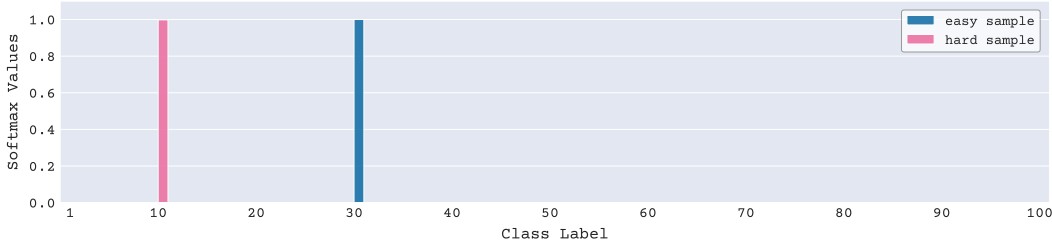

(a) Softmax Scores: 1.0 (easy sample) vs. 0.998 (hard sample)

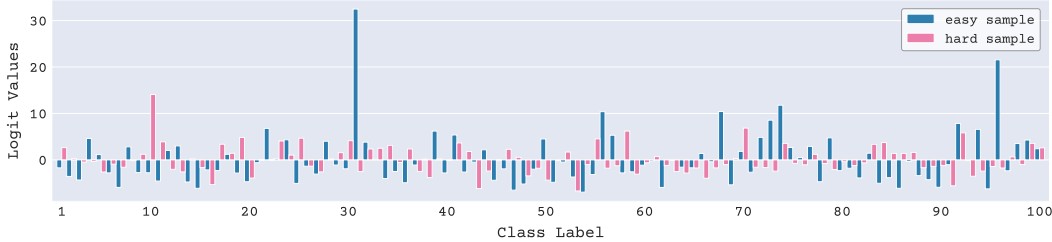

(b) Negative Energy Scores: 32.49 (easy sample) vs. 14.08 (hard sample)

Figure 2: (a) Softmax probability distributions and (b) raw logit outputs of two CIFAR-100 samples computed by a trained ResNet-56. Both samples receive similarly high softmax confidence scores, despite differing significantly in difficulty (1 vs. 27). In contrast, their negative energy scores more clearly reflect this difference.

where $\tau$ is a temperature parameter. The free energy represents a soft minimum of the joint energies for a given input $x$. A low free energy indicates that the model assigns high certainty to at least one class, suggesting the input is familiar. Conversely, a high free energy indicates that the model is uncertain across all classes.

This relationship allows us to define a model-implied marginal density over the input space $\mathcal{X}$:

$$p(x) = \frac{\exp(-F(x)/\tau)}{Z}, \quad \text{where} \quad Z = \int_{x' \in \mathcal{X}} \exp(-F(x')/\tau)\, dx', \tag{5}$$

is the partition function, a constant that ensures the distribution integrates to one. This formulation implies that inputs corresponding to high-density regions of the data distribution (i.e., "typical" examples) are assigned low energy, while those in low-density regions have high energy (Liu et al., 2020). We now formalize the connection between free energy and epistemic uncertainty.

**Proposition 2.1.** *The Helmholtz free energy $F(x)$ is a valid measure of epistemic uncertainty, as it is linearly proportional to the negative log-likelihood of the model-implied data density $p(x)$.*

We refer to Appendix G.2 for proof. This alignment makes the energy score a desirable quantity for epistemic uncertainty (Fuchsgruber et al., 2024; Zong & Huang, 2025) and thus suitable for OOD detection (Liu et al., 2020; Wang et al., 2021).

## 2.3 ENERGY-BASED NONCONFORMITY SCORES

To clarify our motivation for energy-based conformal classification, we illustrate with a real example how integrating free energy into conformal classification can be beneficial, as it provides additional information not necessarily captured in the softmax space.

Following the definition in Angelopoulos et al. (2021), we quantify the *difficulty* of a sample $(x, y_{\text{true}})$ as

$$D(x, y_{\text{true}}) = o_x(y_{\text{true}}), \tag{6}$$

where $o_x(y_{\text{true}})$ denotes the rank of the true label $y_{\text{true}}$ in the model's predicted class-probability ordering (from most to least likely). Formally,

$$o_x(y) = \big|\{k \in [K] : \hat{\pi}(k \mid x) \geq \hat{\pi}(y \mid x)\}\big|. \tag{7}$$

Inspired by the analysis in Liu et al. (2020), Figure 2(a) displays the softmax probability distributions produced by a pretrained ResNet-56 model for two samples from the CIFAR-100 dataset, while Figure 2(b) shows the corresponding raw logit outputs for each sample.

The first sample is considered *"easy"*, with a difficulty of 1 (i.e., the true label has the highest predicted probability), while the second is *"hard"*, with a difficulty of 27 (i.e., misclassified by the model). Notably, despite the substantial difference in difficulty, both samples exhibit nearly identical softmax confidence scores, which would make them indistinguishable under standard softmax-based uncertainty metrics. In contrast, their negative energy scores ($-F(x)$), computed from the logits, are significantly more separable. This suggests that $F(x)$ captures a different, and potentially more nuanced, aspect of uncertainty. Easy or high-density samples yield large $-F(x)$, while hard, ambiguous, low-density or OOD samples yield smaller $-F(x)$.

To further investigate this behaviour, Figure 3 shows the distribution of energy scores across the CIFAR-100 test set calculated with a trained ResNet-56 model, stratified by sample difficulty. As the figure illustrates, energy distributions shift noticeably across difficulty levels, suggesting that logits (and their derived energy scores) retain richer information about a model's confidence than the softmax outputs alone. This highlights energy as an informative signal for uncertainty that can improve the efficiency of nonconformity scores, particularly in cases where softmax probabilities are overconfident or poorly aligned with true sample difficulty.

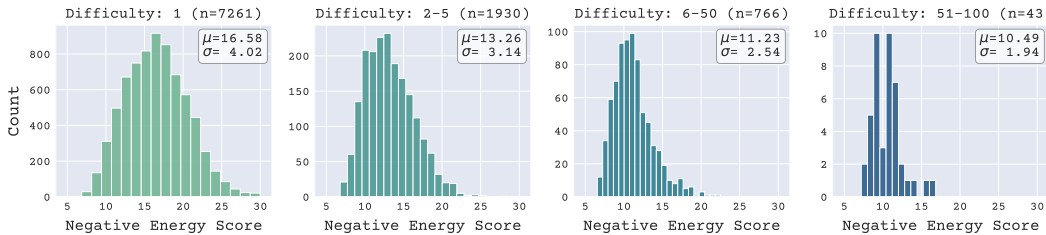

Figure 3: Distribution of negative energy scores ($-F(x)$), stratified by sample difficulty. As difficulty increases, the distribution shifts toward lower energy values, indicating reduced model confidence.

**Theorem 2.2** (Monotonicity of Expected Confidence with Sample Difficulty). *Consider two difficulty levels $d_1$ and $d_2$ such that $1 \leq d_1 < d_2 \leq K$. Let $\mathbb{E}_{(X,Y)\sim\mathcal{D}}[\cdot \mid D(X, Y_{true}) = d]$ denote the expectation over the data distribution conditional on samples having difficulty d. For a classifier successfully trained to convergence on a representative dataset, the expected negative free energy is a strictly monotonically decreasing function of difficulty:*

$$\mathbb{E}[-F(X) \mid D(X, Y_{true}) = d_1] > \mathbb{E}[-F(X) \mid D(X, Y_{true}) = d_2]. \tag{8}$$

The proof of Theorem 2.2 is provided in Appendix G.3.

Having established that the free energy, derived from the logit space, captures epistemic uncertainty and sample difficulty more effectively than softmax probabilities, we propose to integrate the energy score into the nonconformity scores, an approach that aligns with the principles of normalized conformal prediction. By using free energy as a sample-specific difficulty measure, we aim to scale the base nonconformity scores to produce prediction sets that better adjust to the model's uncertainty regarding each sample.

We define the energy-based variant of a base adaptive nonconformity score, $S(x, y)$, as:

$$S_{\text{Energy-Based}}(x, y) = S(x, y) \cdot \frac{1}{\beta} \log\left(1 + e^{-\beta F(x)}\right). \tag{9}$$

Here, the scaling factor is a softplus function of the negative free energy, $-F(x)$, which ensures a positive, input-dependent weight. The parameter $\beta > 0$ controls the sharpness of this function. This modulation re-calibrates the nonconformity score on a per-sample basis, leveraging the model's epistemic uncertainty.

The intuition behind this formulation is as follows:

- **For "easy" in-distribution samples**, the model is certain, resulting in a large negative free energy (i.e., large and positive $-F(x)$). This yields a large scaling factor, which

magnifies the base score $S(x, y)$. This reweighting causes the scores of incorrect labels to more readily exceed the fixed conformal quantile $\hat{q}$, leading to smaller and more efficient prediction sets.

- **For "hard" or OOD samples**, the model is uncertain, and $-F(x)$ is small or negative. The scaling factor becomes small, thereby dampening the base score. This dampening reduces the magnitude of all scores for the given input, causing more plausible labels to fall below the conformal quantile $\hat{q}$ and thus producing adaptively larger sets that reflect the model's uncertainty.

As shown in Proposition 2.3, scaling the score is equivalent to adjusting the quantile threshold $\hat{q}$ for each input, tightening it for confident predictions and relaxing it for uncertain ones. In our experiments, we apply this modulation to several state-of-the-art scores, including APS, RAPS, and SAPS. An extension of this modulation to the LAC score is provided in Appendix J.

**Proposition 2.3** (Equivalence to Sample-Dependent Thresholding). *Let $S(x, y)$ be any adaptive base nonconformity score and let $G(x)$ be a positive, sample-dependent scaling function (e.g., $G(x) = \text{softplus}(-F(x); \beta) = \frac{1}{\beta} \log(1 + e^{-\beta F(x)})$), assuming this is positive. Let $\mathcal{C}_G(x)$ be the prediction set constructed using the scaled score $S_G(x, y) = G(x)S(x, y)$ and its corresponding quantile $\hat{q}_{1-\alpha}^{(G)}$ derived from the calibration set $\mathcal{D}_{cal} = \{(x_i, y_i)\}_{i=1}^{N}$.*

*This construction is mathematically equivalent to using the original base score $S(x, y)$ with a sample-dependent threshold $\theta(x)$ that varies for each test sample:*

$$\mathcal{C}_G(x) = \{y \in \{1, \ldots, K\} \mid S(x, y) \leq \theta(x)\}, \tag{10}$$

*where the threshold is defined as:*

$$\theta(x) = \frac{\hat{q}_{1-\alpha}^{(G)}}{G(x)}. \tag{11}$$

We refer to Appendix G.4 for proof.

## 3 EXPERIMENTS

We present a comprehensive empirical evaluation of our proposed energy-scaled nonconformity scores, comparing them across various data regimes and distributional challenges. The objective of CP is to produce prediction sets $\mathcal{C}(X)$ for a test instance $X$ such that its unknown true label $Y$ is included with a user-specified probability $1 - \alpha$, i.e., $\mathbb{P}(Y \in \mathcal{C}(X)) \geq 1 - \alpha$.

### 3.1 BALANCED TRAINING DATA

We first evaluate performance when models are trained on datasets where the prior distribution over class labels is uniform, i.e., $\mathbb{P}_{\text{train}}(Y = y) = 1/|\mathcal{Y}|$ for all $y \in \mathcal{Y}$. This includes standard ImageNet-Val, Places365, and CIFAR-100 training sets. For $\alpha \in \{0.01, 0.025, 0.05, 0.1\}$, we report empirical coverage and average prediction set size in Table 1. This establishes whether energy-based methods maintain coverage while potentially improving adaptiveness and efficiency under standard, balanced training conditions. Detailed difficulty-stratified results are also reported in Appendix I.

### 3.2 IMBALANCED TRAINING DATA

We then study performance on data with an imbalanced class prior. For this, we use CIFAR-100-LT training variants, which are designed to simulate long-tailed distributions where class frequencies decay exponentially ($\mathbb{P}_{\text{train}}(Y = j) \propto \exp(-\lambda \cdot j)$). The parameter $\lambda$ controls the severity of this imbalance, with higher values indicating a stronger imbalance, as illustrated in Figure 7.

Modern deep networks trained on such long-tailed data exhibit a *"familiarity bias"*, where the model shows higher confidence for majority classes and lower confidence for minority classes (Wallace & Dahabreh, 2012; Samuel et al., 2021). This makes conformal prediction with standard softmax scores to under-cover minority classes. To address this, our energy-based variants dampen the non-conformity scores of minority classes more than those of majority classes. This helps to expand prediction sets for minority classes, fostering their labels' inclusion.

Table 1: Performance comparison of `APS`, `RAPS`, and `SAPS` nonconformity score functions and their energy-based variants on CIFAR-100, ImageNet, and Places365 at miscoverage levels $\alpha \in \{0.01, 0.025, 0.05, 0.1\}$. Results are averaged over 10 trials. For the **Set Size** column, lower is better. **Bold** values indicate the best performance within each method family (e.g., `APS` with and without `Energy`).

| Method | | $\alpha = 0.1$ Coverage | Set Size | $\alpha = 0.05$ Coverage | Set Size | $\alpha = 0.025$ Coverage | Set Size | $\alpha = 0.01$ Coverage | Set Size |
|---|---|---|---|---|---|---|---|---|---|
| **CIFAR-100 (ResNet-56)** | | | | | | | | | |
| APS | w/o Energy | $0.90 \pm 0.01$ | $3.17 \pm 0.09$ | $0.95 \pm 0.00$ | $6.91 \pm 0.24$ | $0.975 \pm 0.002$ | $13.29 \pm 0.44$ | $0.99 \pm 0.00$ | $25.79 \pm 1.20$ |
| APS | w/ Energy | $0.90 \pm 0.01$ | $\mathbf{3.16} \pm 0.08$ | $0.95 \pm 0.00$ | $\mathbf{6.49} \pm 0.24$ | $0.974 \pm 0.001$ | $\mathbf{11.48} \pm 0.25$ | $0.99 \pm 0.00$ | $\mathbf{22.90} \pm 0.82$ |
| RAPS | w/o Energy | $0.90 \pm 0.00$ | $3.13 \pm 0.07$ | $0.95 \pm 0.01$ | $8.17 \pm 0.47$ | $0.974 \pm 0.002$ | $16.38 \pm 0.81$ | $0.99 \pm 0.00$ | $30.88 \pm 1.90$ |
| RAPS | w/ Energy | $0.90 \pm 0.01$ | $3.13 \pm 0.08$ | $0.95 \pm 0.00$ | $\mathbf{6.18} \pm 0.25$ | $0.974 \pm 0.002$ | $\mathbf{11.34} \pm 0.32$ | $0.99 \pm 0.00$ | $\mathbf{23.63} \pm 0.86$ |
| SAPS | w/o Energy | $0.90 \pm 0.01$ | $2.87 \pm 0.09$ | $0.95 \pm 0.00$ | $7.47 \pm 0.43$ | $0.974 \pm 0.002$ | $15.08 \pm 0.72$ | $0.99 \pm 0.00$ | $29.80 \pm 1.82$ |
| SAPS | w/ Energy | $0.90 \pm 0.01$ | $\mathbf{2.87} \pm 0.11$ | $0.95 \pm 0.00$ | $\mathbf{5.94} \pm 0.16$ | $0.974 \pm 0.001$ | $\mathbf{10.73} \pm 0.23$ | $0.99 \pm 0.00$ | $\mathbf{22.90} \pm 0.82$ |
| **ImageNet (ResNet-50)** | | | | | | | | | |
| APS | w/o Energy | $0.90 \pm 0.00$ | $1.60 \pm 0.02$ | $0.95 \pm 0.00$ | $3.99 \pm 0.18$ | $0.976 \pm 0.001$ | $11.72 \pm 0.26$ | $0.99 \pm 0.00$ | $39.08 \pm 1.18$ |
| APS | w/ Energy | $0.90 \pm 0.00$ | $1.66 \pm 0.03$ | $0.95 \pm 0.00$ | $\mathbf{3.84} \pm 0.17$ | $0.976 \pm 0.001$ | $\mathbf{10.11} \pm 0.30$ | $0.99 \pm 0.00$ | $\mathbf{32.93} \pm 1.25$ |
| RAPS | w/o Energy | $0.90 \pm 0.00$ | $1.77 \pm 0.03$ | $0.95 \pm 0.00$ | $4.22 \pm 0.06$ | $0.976 \pm 0.001$ | $10.56 \pm 0.21$ | $0.99 \pm 0.00$ | $37.01 \pm 1.33$ |
| RAPS | w/ Energy | $0.90 \pm 0.00$ | $\mathbf{1.76} \pm 0.04$ | $0.95 \pm 0.00$ | $\mathbf{3.88} \pm 0.07$ | $0.976 \pm 0.001$ | $\mathbf{9.18} \pm 0.29$ | $0.99 \pm 0.00$ | $\mathbf{31.47} \pm 1.28$ |
| SAPS | w/o Energy | $0.90 \pm 0.00$ | $1.67 \pm 0.01$ | $0.95 \pm 0.00$ | $3.67 \pm 0.08$ | $0.976 \pm 0.001$ | $9.75 \pm 0.31$ | $0.99 \pm 0.00$ | $35.97 \pm 1.46$ |
| SAPS | w/ Energy | $0.90 \pm 0.00$ | $\mathbf{1.66} \pm 0.03$ | $0.95 \pm 0.00$ | $\mathbf{3.66} \pm 0.06$ | $0.976 \pm 0.001$ | $\mathbf{8.50} \pm 0.29$ | $0.99 \pm 0.00$ | $\mathbf{30.24} \pm 1.12$ |
| **Places365 (ResNet-50)** | | | | | | | | | |
| APS | w/o Energy | $0.90 \pm 0.00$ | $7.56 \pm 0.13$ | $0.95 \pm 0.00$ | $14.28 \pm 0.24$ | $0.975 \pm 0.002$ | $24.92 \pm 0.79$ | $0.99 \pm 0.00$ | $46.81 \pm 1.93$ |
| APS | w/ Energy | $0.90 \pm 0.00$ | $\mathbf{7.11} \pm 0.11$ | $0.95 \pm 0.00$ | $\mathbf{12.98} \pm 0.23$ | $0.975 \pm 0.002$ | $\mathbf{22.32} \pm 0.68$ | $0.99 \pm 0.00$ | $\mathbf{40.73} \pm 1.69$ |
| RAPS | w/o Energy | $0.90 \pm 0.00$ | $7.37 \pm 0.16$ | $0.95 \pm 0.00$ | $14.37 \pm 0.27$ | $0.976 \pm 0.001$ | $26.34 \pm 0.57$ | $0.99 \pm 0.00$ | $50.64 \pm 1.68$ |
| RAPS | w/ Energy | $0.90 \pm 0.00$ | $\mathbf{6.85} \pm 0.11$ | $0.95 \pm 0.00$ | $\mathbf{12.67} \pm 0.23$ | $0.975 \pm 0.002$ | $\mathbf{22.35} \pm 0.64$ | $0.99 \pm 0.00$ | $\mathbf{41.59} \pm 1.32$ |
| SAPS | w/o Energy | $0.90 \pm 0.00$ | $7.20 \pm 0.14$ | $0.95 \pm 0.00$ | $14.11 \pm 0.30$ | $0.976 \pm 0.001$ | $25.76 \pm 0.49$ | $0.99 \pm 0.00$ | $49.80 \pm 1.74$ |
| SAPS | w/ Energy | $0.90 \pm 0.00$ | $\mathbf{6.79} \pm 0.09$ | $0.95 \pm 0.00$ | $\mathbf{12.51} \pm 0.18$ | $0.975 \pm 0.002$ | $\mathbf{22.19} \pm 0.69$ | $0.99 \pm 0.00$ | $\mathbf{41.19} \pm 1.29$ |

Indeed, energy scores capture this training imbalance (Liu et al., 2024a). Figure 4 visually demonstrates how the distributions of negative energy scores shift across different class bins under both balanced and imbalanced training conditions.

**Theorem 3.1** (Free Energy as an Indicator of Class Imbalance). *Let $f$ be a classifier trained on a dataset drawn from a distribution $P_{train}(X, Y)$ with imbalanced class priors. Consider two classes, a majority class $y_{maj}$ and a minority class $y_{min}$, such that their training priors satisfy $P_{train}(Y = y_{maj}) > P_{train}(Y = y_{min})$.*

*Let the model be evaluated on a balanced test distribution $P_{test}$. Assume the classes are of comparable intrinsic complexity. Then, the expected negative free energy for test samples from the majority class will be greater than that for the minority class:*

$$\mathbb{E}_{X \sim P_{test}(X|Y=y_{maj})}[-F(X)] > \mathbb{E}_{X \sim P_{test}(X|Y=y_{min})}[-F(X)]. \tag{12}$$

See Appendix G.5 for proof. This setup allows us to evaluate how energy influences adaptiveness when a model's representations are shaped by imbalanced training. We report marginal coverage and average set size with the standard balanced CIFAR-100 calibration set and test set, with results presented in Table 2. We refer to Appendix K for additional experiments on imbalanced scenario.

### 3.3 RELIABILITY UNDER DISTRIBUTIONAL SHIFT

An important test for any uncertainty quantification method is its response to out-of-distribution (OOD) data. This scenario is particularly challenging for conformal prediction because the assumption of exchangeability between the calibration and test data is violated. Consequently, the formal

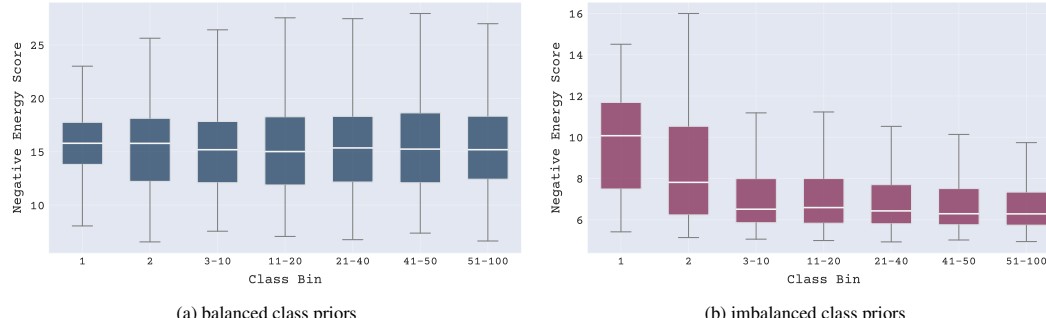

Figure 4: Distributions of negative energy scores across various class bins under balanced and imbalanced training. Results are for CIFAR-100. (a) Balanced model: scores are consistent across class bins. (b) Imbalanced model ($\lambda = 0.03$): minority classes exhibit lower negative energy scores, reflecting reduced confidence.

Table 2: Performance comparison of different nonconformity scores and their energy-based variants on imbalanced CIFAR-100 with an imbalance factor of $\lambda = 0.005$ and at miscoverage levels $\alpha \in \{0.01, 0.025, 0.05, 0.1\}$. Results are averaged over 10 trials with a ResNet-56 model. For the **Set Size** column, lower is better. **Bold** values indicate the best performance within each method family (e.g., APS with and without Energy). Results for additional $\lambda$ values are provided in Appendix K.

| Method | | $\alpha = 0.1$ | | $\alpha = 0.05$ | | $\alpha = 0.025$ | | $\alpha = 0.01$ | |
|---|---|---|---|---|---|---|---|---|---|
| | | Coverage | Set Size | Coverage | Set Size | Coverage | Set Size | Coverage | Set Size |
| **CIFAR-100-LT ($\lambda = 0.005$) (ResNet-56)** | | | | | | | | | |
| APS | w/o Energy | $0.90 \pm 0.01$ | $8.44 \pm 0.30$ | $0.95 \pm 0.00$ | $17.22 \pm 0.55$ | $0.973 \pm 0.003$ | $28.32 \pm 0.86$ | $0.99 \pm 0.00$ | $45.10 \pm 0.75$ |
| | w/ Energy | $0.90 \pm 0.01$ | $\mathbf{7.41} \pm 0.22$ | $0.95 \pm 0.01$ | $\mathbf{13.30} \pm 0.65$ | $0.973 \pm 0.004$ | $\mathbf{21.72} \pm 1.27$ | $0.99 \pm 0.00$ | $\mathbf{34.78} \pm 1.72$ |
| RAPS | w/o Energy | $0.90 \pm 0.01$ | $8.88 \pm 0.30$ | $0.95 \pm 0.00$ | $18.54 \pm 0.73$ | $0.972 \pm 0.003$ | $30.09 \pm 0.94$ | $0.99 \pm 0.00$ | $51.65 \pm 1.75$ |
| | w/ Energy | $0.90 \pm 0.01$ | $\mathbf{7.59} \pm 0.25$ | $0.95 \pm 0.01$ | $\mathbf{13.26} \pm 0.69$ | $0.973 \pm 0.004$ | $\mathbf{22.27} \pm 1.24$ | $0.99 \pm 0.00$ | $\mathbf{35.43} \pm 1.88$ |
| SAPS | w/o Energy | $0.90 \pm 0.01$ | $8.59 \pm 0.35$ | $0.95 \pm 0.00$ | $17.96 \pm 0.66$ | $0.972 \pm 0.003$ | $29.25 \pm 1.04$ | $0.99 \pm 0.00$ | $50.30 \pm 1.86$ |
| | w/ Energy | $0.90 \pm 0.01$ | $\mathbf{7.58} \pm 0.23$ | $0.95 \pm 0.01$ | $\mathbf{13.19} \pm 0.68$ | $0.973 \pm 0.004$ | $\mathbf{21.99} \pm 1.27$ | $0.99 \pm 0.00$ | $\mathbf{35.18} \pm 1.91$ |

guarantee of marginal coverage no longer holds. This challenge is amplified in real-world deployments where a model, calibrated on in-distribution samples, inevitably encounters novel inputs. These inputs can range from simple *covariate shifts* (e.g., familiar objects in new contexts) to more severe *semantic shifts*, where the inputs belong to classes entirely unseen during training.

In the absence of coverage guarantees, a reliable conformal classifier should not provide a small, incorrect prediction without some indication of its uncertainty. This motivates the following desiderata for the behavior of a conformal predictor $C(\cdot)$ when presented with an OOD input $x_{\text{ood}}$ drawn from an OOD distribution $P_{\text{ood}}$, compared to an in-distribution input $x_{\text{id}}$ drawn from $P_{\text{id}}$.

**Desiderata for a Reliable Conformal Classifier on OOD Data** We establish the following desiderata for a conformal predictor's behavior when encountering OOD data, where the standard exchangeability assumption is violated and coverage guarantees no longer hold.

**Desideratum 1** (Adaptive Uncertainty Response). *When faced with an out-of-distribution input, a reliable conformal predictor must adapt its output to signal increased uncertainty. This signal should manifest as either an expansion of the prediction set size or as a principled abstention via an empty set. This response is characterized by one or both of the following outcomes:*

*(i) A significant increase in the probability of abstention:*

$$P_{X \sim P_{ood}}(C(X) = \emptyset) \gg P_{X \sim P_{id}}(C(X) = \emptyset) \approx 0$$

*(i) An inflation in the size of non-empty prediction sets, such that the expected size of non-empty OOD sets is greater than the expected size of in-distribution sets:*

$$\mathbb{E}_{X \sim P_{ood}}[|C(X)|] > \mathbb{E}_{X \sim P_{id}}[|C(X)|]$$

**Desideratum 2** (Avoidance of False Confidence). *The predictor should minimize the probability of producing a small, non-empty set (e.g., of size 1 or 2) for an OOD input.*

$$minimize\ P_{X \sim P_{ood}}(1 \leq |C(X)| \leq k)\ for\ small\ k$$

In summary, as also discussed in Appendix P, a larger or empty set is an informative and appropriate outcome in this scenario, whereas a small, incorrect set is problematic. To assess how our method aligns with the OOD desiderata, we designed an experiment under a semantic shift. A ResNet-56 model was calibrated on in-distribution CIFAR-100 data and evaluated on the Places365 as the OOD dataset. As coverage is not a meaningful metric in this context, our analysis focuses on prediction set size.

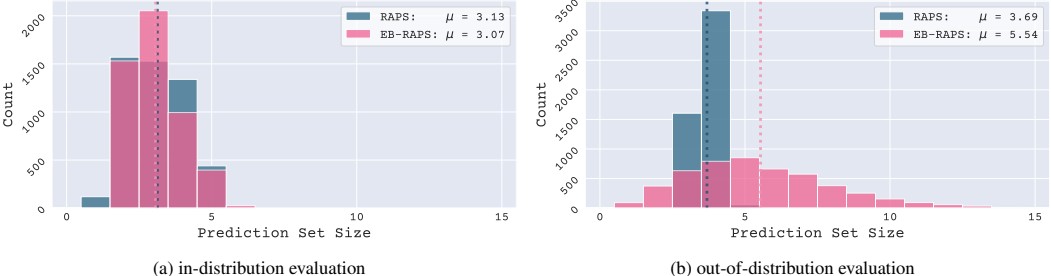

(a) in-distribution evaluation      (b) out-of-distribution evaluation

Figure 5: Prediction set size distributions for the `SAPS` score and its energy-based variant with $\alpha = 0.05$, on (a) in-distribution CIFAR-100 and (b) out-of-distribution Places365 data. The energy-based variant produces larger prediction sets on OOD data. Here, $\mu$ represents the overall set size.

The results demonstrate an alignment with our desiderata. As shown in Table 3, energy-based scores produce larger average sets compared to their base counterparts. Figure 5 provides a visual illustration of this adaptive behavior, comparing the `RAPS` score with its energy-based variant. The Energy-based `RAPS` produces smaller prediction sets on ID data and larger prediction sets for OOD samples. This response, shows improvement towards Desideratum 1, compared to baseline `RAPS`.

Table 3: Comparison of average prediction set sizes for a ResNet-56 model trained on CIFAR-100. The model is evaluated on both in-distribution (CIFAR-100) and out-of-distribution (Places365) data. Energy-based variants demonstrate adaptiveness to the distributional shift by maintaining small sets on ID data while producing significantly larger sets for OOD inputs. **Bold** values indicate the preferred result: the smallest average set size for ID (efficiency) and the largest for OOD (uncertainty awareness).

| Method | | $\alpha = 0.1$ Set Size ID (in distribution) | $\alpha = 0.1$ Set Size OOD (out of distribution) | $\alpha = 0.05$ Set Size ID (in distribution) | $\alpha = 0.05$ Set Size OOD (out of distribution) |
|---|---|---|---|---|---|
| **APS** | w/o `Energy` | $3.17 \pm 0.09$ | $6.18 \pm 0.25$ | $6.91 \pm 0.24$ | $14.91 \pm 0.81$ |
| | w/ `Energy` | $\mathbf{3.16} \pm 0.08$ | $\mathbf{86.76} \pm 0.94$ | $\mathbf{6.49} \pm 0.24$ | $\mathbf{93.40} \pm 0.53$ |
| **RAPS** | w/o `Energy` | $\mathbf{3.13} \pm 0.07$ | $3.70 \pm 0.04$ | $8.17 \pm 0.47$ | $8.95 \pm 0.47$ |
| | w/ `Energy` | $\mathbf{3.13} \pm 0.08$ | $\mathbf{5.53} \pm 0.07$ | $\mathbf{6.18} \pm 0.25$ | $\mathbf{9.05} \pm 0.49$ |
| **SAPS** | w/o `Energy` | $\mathbf{2.87} \pm 0.09$ | $3.78 \pm 0.05$ | $7.47 \pm 0.43$ | $8.82 \pm 0.46$ |
| | w/ `Energy` | $\mathbf{2.87} \pm 0.11$ | $\mathbf{5.55} \pm 0.10$ | $\mathbf{5.94} \pm 0.16$ | $\mathbf{9.53} \pm 0.18$ |

## 4 CONCLUSION

This paper demonstrates that the reliability of conformal classifiers is enhanced by moving beyond softmax probabilities to leverage information about model uncertainty from the logit space. Our proposed energy-based framework adjusts standard nonconformity scores on a per-sample basis, leveraging this principled measure of model certainty to make each score sensitive to the model's confidence in that specific input. Our evaluations on common nonconformity scores, across multiple datasets and architectures, confirm that our approach yields prediction sets with improved efficiency and adaptiveness, all while preserving the theoretical coverage guarantee.

## 5 REPRODUCIBILITY STATEMENT

The empirical results presented in this paper are fully reproducible. Our implementation, based on PyTorch and using the TorchCP library, are available at https://github.com/navidattar/Energy-Based-Conformal-Classification. Detailed descriptions of hyperparameters, and environment specification for running experiments, are provided in Appendix F.

### ACKNOWLEDGMENTS

This research was supported under the Australian Research Council's Industrial Transformation Research Program (ITRP) funding scheme (project number IH210100051). The ARC Digital Bioprocess Development Hub is a collaboration between The University of Melbourne, University of Technology Sydney, RMIT University, CSL Innovation Pty Ltd, Cytiva (Global Life Science Solutions Australia Pty Ltd) and Patheon Biologics Australia Pty Ltd.

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

APPENDIX

## A  NOTATION

Table 4: Notation used in this work.

| Symbol | Meaning |
|--------|---------|
| $x \in \mathbb{R}^D$ | Input (feature) vector |
| $K$ | Total number of classes |
| $y \in \{1, \ldots, K\}$ | Class label |
| $\mathbf{f}(x) = (f_1, \ldots, f_K)$ | Pre-softmax logit vector produced by the classifier |
| $\hat{\pi}(y \mid x)$ | Model's softmax probability for class $y$, Equation 1 |
| $f_{\max}(x)$ | $\max_k f_k(x)$ |
| $S(x, y)$ | General nonconformity score |
| $T$ | Temperature used in the calibrated softmax |
| $\tau$ | Temperature used in the energy calculation |
| $\alpha$ | Desired miscoverage level / target error rate |
| $\hat{q}_{1-\alpha}$ | Quantile threshold for prediction set construction |
| $E(x, y)$ | Joint energy, $E(x, y) = -f_y(x)$ |
| $F(x)$ | Helmholtz free energy score, Equation 4 |
| $\beta$ | Softplus sharpness parameter |
| $D(x, Y_{\text{true}})$ | Sample difficulty measure |
| $o_x(y)$ | Rank of label $y$ in the model's predicted class-probability ordering |
| $\mathbb{H}(x)$ | Shannon entropy of $\hat{\pi}(y \mid x)$ |
| $\mathcal{D} = \{(x_i, y_i)\}_{i=1}^{N}$ | Dataset with $N$ samples, inputs $x_i$ and labels $y_i$ |
| $\delta(x)$ | Geometric distance of $x$ from the decision boundary |

## B  RELATED WORKS

CP is a statistical framework that provides distribution-free, finite-sample coverage guarantees for predictions (Vovk et al., 2005). This robust approach to uncertainty quantification has seen widespread adoption across numerous real-world applications. These include regression (Lei & Wasserman, 2014; Romano et al., 2019), classification (Sadinle et al., 2019), structured prediction (Bates et al., 2021), large language models (LLMs) (Su et al., 2024; Cherian et al., 2024; Kumar et al., 2023; Ren et al., 2023; Quach et al., 2024), and diffusion models (Horwitz & Hoshen, 2022; Teneggi et al., 2023). graph neural networks (GNNs) (Zargarbashi et al., 2023; Huang et al., 2023; Wijegunawardana et al., 2020; Clarkson, 2023; Song et al., 2024), and image generative models (Horwitz & Hoshen, 2022). Further applications are found in robotic control (Kang et al., 2024; Luo & Zhou, 2024), hyperspectral imaging (Liu et al., 2024b), healthcare (Lindemann et al., 2024), finance (Bellotti, 2021), autonomous systems and automated vehicles (Lindemann et al., 2024; Zecchin et al., 2024; Bang et al., 2024), human-in-the-loop decision making (Straitouri et al., 2023; Cresswell et al., 2024), bioprocessing (Pham et al., 2025), and scientific machine learning (Moya et al., 2024; Podina et al., 2024).

The foundational inductive conformal prediction framework (Vovk et al., 2005) often employs a split conformal (or inductive conformal) approach, where the dataset is divided into a training set for model fitting and a disjoint calibration set for uncertainty quantification (Papadopoulos et al., 2002b; Vovk et al., 2005; Shafer & Vovk, 2008; Angelopoulos & Bates, 2021; Lei et al., 2015). This is done to manage the computational aspects of CP. Beyond this common split, other CP variants

include methods based on cross-validation (Vovk, 2015) or the jackknife (i.e., leave-one-out) technique (Barber et al., 2021). The primary goals within CP research are to enhance the efficiency of prediction sets (i.e., reduce their size) and to ensure and improve the validity of coverage rates.

**Improving Prediction Set Efficiency.** Strategies to improve the efficiency of prediction sets predominantly fall into two categories: training-time modifications and post-hoc adjustments.

One line of research focuses on developing new training algorithms or regularizations to learn models that inherently produce smaller prediction sets while maintaining coverage (Bellotti, 2021; Colombo & Vovk, 2020; Chen et al., 2021; Stutz et al., 2022; Einbinder et al., 2022; Bai et al., 2022; Fisch et al., 2021; Yang & Kuchibhotla, 2021; Correia et al., 2024). For example, the uncertainty-aware conformal loss aims to optimize APS (Romano et al., 2020) by encouraging non-conformity scores towards a uniform distribution (Einbinder et al., 2022), while ConfTr introduces a regularization term to minimize average set size (Stutz et al., 2022). However, such training methods can be computationally intensive due to model retraining and optimization complexity. Early stopping has also been explored as a technique to select models leading to more compact prediction sets under guaranteed coverage (Liang et al., 2023).

The other avenue involves post-hoc techniques applied to pre-trained models. This includes the design of novel non-conformity score functions. Notable examples are LAC (Sadinle et al., 2019), APS (Romano et al., 2020), RAPS (Angelopoulos et al., 2021), SAPS (Huang et al., 2024b), Top-K (Angelopoulos et al., 2021; Luo & Zhou, 2024), and others (Ghosh et al., 2023). Post-hoc learning methods have also been proposed (Xi et al., 2024). Research also addresses unique settings such as federated learning (Lu et al., 2023; Plassier et al., 2023), multi-label classification (Cauchois et al., 2021; Fisch et al., 2022; Papadopoulos, 2022), outlier detection (Bates et al., 2023; Guan & Tibshirani, 2022), and out-of-distribution (OOD) detection (Chen et al., 2023; Novello et al., 2024). A common challenge for some post-hoc methods is their reliance on potentially unreliable probability outputs from models.

Recent efforts have sought to make conformal prediction sets more adaptive by explicitly incorporating epistemic uncertainty. One line of work proposes methods that operate on richer, second-order predictions. For instance, Javanmardi et al. (2025) introduce Bernoulli Prediction Sets (BPS), which construct provably optimal (i.e., smallest) prediction sets under the assumption that the true data distribution is contained within a given "credal set" derived from models like deep ensembles or Bayesian neural networks. A complementary, model-agnostic approach is taken by Cabezas et al. (2025) with EPICSCORE, which enhances any standard nonconformity score by training a separate Bayesian model to learn its conditional distribution. This transforms the score to reflect epistemic uncertainty, achieving asymptotic conditional coverage. However, our energy-based framework improves the adaptiveness of Conformal Classifiers by leveraging uncertainty information from pre-softmax logits via the Helmholtz free energy, thus avoiding the need for second-order predictors or additional post-hoc computational costs.

**Ensuring Validity and Enhancing Coverage Rates.** A significant body of work is dedicated to ensuring the validity of the marginal coverage rate and improving it, particularly under challenging conditions, as well as striving for stronger conditional coverage guarantees (Shi et al., 2013; Löfström et al., 2015). Efforts have been made to maintain marginal coverage by adapting CP to scenarios involving adversarial examples (Gendler et al., 2022; Kang et al., 2024), covariate shift (Tibshirani et al., 2019; Deng et al., 2023), label shift (Podkopaev & Ramdas, 2021; Plassier et al., 2023), and noisy labels (Feldman et al., 2023; Sesia et al., 2023).

Beyond marginal coverage, many CP algorithms pursue forms of conditional coverage (Vovk, 2012). This includes training-conditional validity, which aims to ensure that most training dataset realizations result in valid marginal coverage on future test data (Bian & Barber, 2023; Pournaderi & Xiang, 2024). Group-conditional CP methods seek to guarantee coverage across predefined groups within the population (Javanmard et al., 2022; Gibbs et al., 2025; Melki et al., 2023). While achieving exact pointwise conditional coverage is known to be impossible in general (Foygel Barber et al., 2021), practical approaches for approximate or class-conditional coverage exist. For instance, LAC demonstrates the possibility of efficient class-conditional coverage (Sadinle et al., 2019). Clustered CP improves class-conditional coverage by leveraging the label space taxonomy, particularly when the number of classes is large (Ding et al., 2023a). Other methods, like $k$-Class-conditional CP, cali-

brate class-specific score thresholds based on top-$k$ errors. The goal remains to enhance conditional coverage properties while still producing efficient and informative prediction sets.

## C ON THE LIMITATIONS OF SOFTMAX OUTPUTS FOR MEASURING MODEL UNCERTAINTY

Our motivation for developing energy-based nonconformity scores, as detailed throughout Section 2, is the inadequacy of softmax outputs for reliably quantifying model uncertainty. To provide a broader context, we reproduce a selection of established criticisms originally compiled in the appendix of Pearce et al. (2021), along with a few more recent perspectives published thereafter.

- *"[The softmax output] is often erroneously interpreted as model confidence."* (Gal & Ghahramani, 2016)
- *"Deterministic models can capture aleatoric uncertainty but cannot capture epistemic uncertainty."* (Gal et al., 2017)
- *"NNs . . . until recently have been unable to provide measures of uncertainty in their predictions."* (Malinin & Gales, 2018)
- *"When asked to predict on a data point unlike the training data, the NN should increase its uncertainty. There is no mechanism built into standard NNs to do this . . . standard NNs cannot estimate epistemic uncertainty."* (Pearce, 2020)
- *"NNs are poor at quantifying predictive uncertainty."* (Lakshminarayanan et al., 2017)
- *"Deep neural networks with the softmax classifier are known to produce highly overconfident posterior distributions even for such abnormal samples."* (Lee et al., 2018b)
- *"The output of the [softmax] classifier cannot identify these [far from the training data] inputs as out-of-distribution."* (Hein et al., 2019)
- *"The only uncertainty that can reliably be captured by looking at the softmax distribution is aleatoric uncertainty."* (van Amersfoort et al., 2020)
- *"Softmax entropy is inherently inappropriate to capture epistemic uncertainty."* (Mukhoti et al., 2021)
- *"For [softmax] classifiers . . . misclassification will occur with high confidence if the unknown is far from any known data."* (Boult et al., 2019)
- *". . . softmax output only reflects the total predictive uncertainty instead of the model uncertainty, leading to false confidence under distribution shift."* (Wang et al., 2024)
- *". . . the raw softmax output is neither very reliable . . . nor can it represent all sources of uncertainty"* (Gawlikowski et al., 2023)
- *"Furthermore, the softmax output cannot be associated with model uncertainty."* (Gawlikowski et al., 2023)
- *". . . the softmax output is often erroneously interpreted as model confidence. In reality, a model can be uncertain in its predictions even with a high softmax output."* (Mobiny et al., 2021)

Additional commentary and remarks also reinforce this view:

- *"Softmax is not telling you anything about . . . model uncertainty."* — Elise Jennings, Training Program on Extreme-Scale Computing (2019)[1]
- *"The [softmax] network has no way of telling you 'I'm completely uncertain about the outcome and don't rely on my prediction'."* — Florian Wilhelm, PyData Berlin (2019)[2]
- *"Just adding a softmax activation does not magically turn outputs into probabilities."* — Tucker Kirven, Neural Network Prediction Scores are not Probabilities (2020)[3]

---

[1] https://youtu.be/Puc_ujh5QZs?t=1323
[2] https://youtu.be/LCDIqL-8bHs?t=262
[3] https://jtuckerk.github.io/prediction_probabilities.html

# D  ENERGY-BASED MODELS

An Energy-Based Model (EBM) defines a probability distribution over an input space $\mathbb{R}^D$ through an energy function $E_\theta : \mathbb{R}^D \to \mathbb{R}$, which is typically parameterized by a neural network with parameters $\theta$. For any input vector $x \in \mathbb{R}^D$, the probability density is given by the Boltzmann distribution:

$$p_\theta(x) = \frac{\exp(-E_\theta(x))}{Z_\theta}, \tag{13}$$

where $Z_\theta = \int_{x'} \exp(-E_\theta(x'))dx'$ is the partition function. This normalization constant is a key challenge in EBMs, as its computation involves integrating over the entire high-dimensional input space, which is generally intractable.

This framework can be extended to model a joint distribution over inputs and class labels, $p(x, y)$. A standard discriminative classifier, which produces a logit vector $\mathbf{f}(x)$, can be re-interpreted as an EBM by defining a joint energy function $E(x, y)$. Following common practice LeCun et al. (2006); Grathwohl et al. (2020), we define the joint energy as the negative logit corresponding to class $y$:

$$E(x, y) = -f_y(x). \tag{14}$$

From this joint model, the conditional probability $p(y \mid x)$ can be derived as:

$$p(y \mid x) = \frac{p(x, y)}{p(x)} = \frac{p(x, y)}{\sum_{k=1}^K p(x, k)} = \frac{\exp(-E(x, y))}{\sum_{k=1}^K \exp(-E(x, k))} = \frac{\exp(f_y(x))}{\sum_{k=1}^K \exp(f_k(x))}, \tag{15}$$

which is precisely the standard softmax probability $\hat{\pi}(y \mid x)$. The marginal probability $p(x)$ can then be associated with a *"free energy"* function $E(x) = -\frac{1}{\tau} \log \sum_{k=1}^K \exp(f_k(x)/\tau)$, where $\tau$ is a temperature parameter.

Training EBMs often proceeds via Maximum Likelihood Estimation (MLE), which aims to shape the energy function $E_\theta(x)$ such that it assigns low energy to data points from the true distribution and high energy elsewhere. The objective is to maximize the log-likelihood of the observed data $\mathcal{D}$:

$$\arg\max_\theta \mathbb{E}_{x \sim p_{\text{data}}}[\log p_\theta(x)]. \tag{16}$$

The gradient of the log-likelihood with respect to the parameters $\theta$ is given by:

$$\nabla_\theta \log p_\theta(x) = \nabla_\theta(-E_\theta(x) - \log Z_\theta) \tag{17}$$

$$= -\nabla_\theta E_\theta(x) - \frac{1}{Z_\theta} \nabla_\theta \int_{x'} \exp(-E_\theta(x'))dx' \tag{18}$$

$$= -\nabla_\theta E_\theta(x) - \int_{x'} \frac{\exp(-E_\theta(x'))}{Z_\theta}(-\nabla_\theta E_\theta(x'))dx' \tag{19}$$

$$= -\nabla_\theta E_\theta(x) + \mathbb{E}_{x' \sim p_\theta}[\nabla_\theta E_\theta(x')]. \tag{20}$$

Remarkably, this gradient can be computed without explicitly evaluating the intractable partition function $Z_\theta$. Updating the parameters via stochastic gradient ascent on the log-likelihood is equivalent to descending on the following loss function:

$$\mathcal{L}_{\text{MLE}} = \mathbb{E}_{x \sim p_{\text{data}}}[E_\theta(x)] - \mathbb{E}_{x' \sim p_\theta}[E_\theta(x')]. \tag{21}$$

This objective can be intuitively understood as a force that "pulls down" the energy of "positive" samples drawn from the data distribution ($p_{\text{data}}$) while "pushing up" the energy of "negative" samples synthesized from the model's current distribution ($p_\theta$).

To operationalize this training procedure, we must be able to draw samples $x'$ from the model distribution $p_\theta(x)$. Since direct sampling is infeasible, this is typically approximated using Markov Chain Monte Carlo (MCMC) methods. A prevalent choice is Stochastic Gradient Langevin Dynamics (SGLD) Welling & Teh (2011), which iteratively refines an initial sample $x_0$ (e.g., drawn from a simple noise distribution or a buffer of previous samples) according to the rule:

$$x_{t+1} = x_t - \alpha_t \nabla_x E_\theta(x_t) + \sqrt{\eta_t}\epsilon, \quad \text{where} \quad \epsilon \sim \mathcal{N}(0, I), \tag{22}$$

where $\alpha_t$ is the step size and $\eta_t$ controls the scale of the injected Gaussian noise. After a sufficient number of steps, the resulting sample $x_T$ is treated as an approximate sample from $p_\theta(x)$. Different strategies for initializing and running the MCMC chain lead to various training algorithms, such as Contrastive Divergence (CD) (Hinton, 2002), which re-initializes the chain from data points at each step, and Persistent Contrastive Divergence (PCD) (Tieleman, 2008), which maintains a persistent chain across training iterations to obtain higher-quality samples.

# E    EXPERIMENTAL PRELIMINARIES

This section details the experimental design, including the conformal prediction framework, the nonconformity scores used, and evaluation metrics used to validate our proposed Energy-based non-conformity scores.

## E.1    CONFORMAL PREDICTION FRAMEWORK

We employ the standard Split Conformal Prediction (CP) framework in our experiments. A base model is trained on a proper training set, and its outputs on a held-out calibration set are used to compute non-conformity scores and determine the quantile threshold $\hat{q}$ required to form prediction sets. The procedure is formally outlined in Algorithm 1.

---

**Algorithm 1** Split Conformal Prediction

---

1: **Input:** Dataset $\mathcal{D}$, desired error rate $\alpha \in (0, 1)$, non-conformity score function $S(x, y)$.
2: Partition $\mathcal{D}$ into a training set $\mathcal{D}_{\text{train}}$ and a calibration set $\mathcal{D}_{\text{cal}}$, such that $\mathcal{D}_{\text{train}} \cap \mathcal{D}_{\text{cal}} = \emptyset$. Let $n = |\mathcal{D}_{\text{cal}}|$.
3: Train the classifier on $\mathcal{D}_{\text{train}}$ to learn the mapping $x \mapsto \mathbf{f}(x)$.
4: For each example $(x_i, y_i) \in \mathcal{D}_{\text{cal}}$, compute the non-conformity score $s_i = S(x_i, y_i)$.
5: Calculate the quantile threshold $\hat{q}$ from the set of calibration scores $\{s_1, \ldots, s_n\}$. Specifically, $\hat{q}$ is the $\frac{\lceil (n+1)(1-\alpha) \rceil}{n}$-th empirical quantile of these scores.
6: **Output:** For a new input $x_{\text{new}}$, the prediction set is constructed as:

$$C(x_{\text{new}}) = \{y \in \{1, \ldots, K\} : S(x_{\text{new}}, y) \leq \hat{q}\}$$

---

## E.2    NONCONFORMITY SCORES FOR DEEP CLASSIFIERS

The non-conformity score function $S(x, y)$ measures how poorly the label $y$ fits the input $x$ according to the trained model. The prediction set $C(x_{\text{new}})$ is then formed by including all labels whose non-conformity scores do not exceed the calibrated threshold $\hat{q}$. A key property of this procedure is that the resulting set is guaranteed to contain the true label with a probability of at least $1 - \alpha$, assuming the test and calibration data points are exchangeable. Exchangeability is a statistical assumption that the joint distribution of the data is invariant to permutation, making it a suitable assumption for scenarios like simple random sampling.

We evaluate a range of established non-conformity scores, each based on a different principle for measuring how much a model's prediction disagrees with a given label, as summarized in Table 5. We compare these established baselines against their Energy-based counterparts, as defined in Equation 9.

Table 5: Nonconformity scores considered in this work. All scores are functions of $\hat{\pi}(y \mid x)$ where $u \sim U[0, 1]$; The function $o_x(y)$ returns the rank position of label $y$ among all possible labels, ordered by the model's predicted probabilities (with rank 1 being the most likely); $\hat{\pi}_{\max}(x) = \max_k \hat{\pi}(k \mid x)$; $(\cdot)^+$ denotes the positive part; $\lambda$ and $k_{\text{reg}}$ are hyperparameters.

| | Method | Nonconformity Score |
|---|---|---|
| | LAC / THR (Sadinle et al., 2019) | $S_{\text{LAC}}(x, y) = 1 - \hat{\pi}(y \mid x) \equiv -\hat{\pi}(y \mid x)$ |
| Adaptive Scores | APS (Romano et al., 2020) | $S_{\text{APS}}(x, y) = \sum_{k=1}^{K} \hat{\pi}(k \mid x) \mathbb{I}\{\hat{\pi}(k \mid x) > \hat{\pi}(y \mid x)\} + u \cdot \hat{\pi}(y \mid x)$ |
| | RAPS (Angelopoulos et al., 2021) | $S_{\text{RAPS}}(x, y) = S_{\text{APS}}(x, y) + \lambda \left( o_x(y) - k_{\text{reg}} \right)^+$ |
| | SAPS (Huang et al., 2024b) | $S_{\text{SAPS}}(x, y) = \begin{cases} u \cdot \hat{\pi}_{\max}(x), & o_x(y) = 1, \\ \hat{\pi}_{\max}(x) + (o_x(y) - 2 + u)\lambda, & \text{otherwise,} \end{cases}$ |

Below, we briefly describe how each method works:

**Least Ambiguous Class (LAC/THR)** (Sadinle et al., 2019) is one of the simplest and earliest scores. Its non-conformity is defined as $S_{\text{LAC}}(x,y) = 1 - \hat{\pi}(y \mid x)$. The score is thus inversely proportional to the model's confidence; a high probability for the true class yields a low non-conformity score. It is worth noting that using the negative probability, $S(x,y) = -\hat{\pi}(y \mid x)$, is mathematically equivalent for constructing the prediction set, as the "+1" in the original formula merely shifts the range from $[-1,0]$ to $[0,1]$ for non-negative interpretability without altering the relative ordering, where higher scores indicate greater nonconformity (less conformity between label and sample). This is because the conformal procedure relies on the rank-ordering of scores to determine the quantile $\hat{q}$, and the transformation from $1 - \hat{\pi}$ to $-\hat{\pi}$ is monotonic, preserving the rank order. For the Energy-based counterpart of this score, we use $-\pi$ (without the bias term). This method provably yields the smallest expected prediction sets compared to other methods proposed after this model, while preserving the marginal coverage, assuming the predicted probabilities are correct. However, this score is non-adaptive, meaning it tends to produce prediction sets of similar size regardless of the sample's intrinsic difficulty, which opened the room for adaptive nonconformity scores and their variants to be proposed later.

**Adaptive Prediction Sets (APS)** (Romano et al., 2020) introduced the concept of adaptiveness to conformal prediction. The score $S_{\text{APS}}(x,y)$ is the cumulative probability mass of all classes deemed more likely than class $y$. Mathematically, this is the sum of softmax probabilities for all labels $k$ whose probability $\hat{\pi}(k \mid x)$ is greater than $\hat{\pi}(y \mid x)$, plus a randomized term to handle ties. This design has a crucial effect: for "easy" examples where the model is confident (i.e., $\hat{\pi}_{\max}(x)$ is high and Entropy is low), the scores for incorrect labels grow rapidly, leading to small prediction sets. Conversely, for "hard" examples where the model is uncertain (a flatter softmax distribution), the scores grow slowly, resulting in larger, more inclusive sets that reflect this uncertainty.

**Regularized Adaptive Prediction Sets (RAPS)** (Angelopoulos et al., 2021) builds directly upon APS by adding a regularization term. Its score is $S_{\text{RAPS}}(x,y) = S_{\text{APS}}(x,y) + \lambda(o_x(y) - k_{\text{reg}})^+$, where $o_x(y)$ is the rank of label $y$'s probability. This term penalizes the inclusion of labels with a low rank (i.e., large $o_x(y)$), effectively preventing the prediction sets from becoming excessively large, especially for uncertain inputs. The hyperparameters $k_{\text{reg}}$ and $\lambda$ control the onset and strength of this size-regularizing penalty.

**Sorted Adaptive Prediction Sets (SAPS)** (Huang et al., 2024b) is a more recent refinement that aims to mitigate the effects of probability miscalibration in the softmax tail. It treats the top-ranked class differently from all others. For labels not ranked first, the score is based on the maximum probability $\hat{\pi}_{\max}(x)$ plus a penalty that increases linearly with the label's rank, weighted by a hyperparameter $\lambda$. This approach avoids summing many small, potentially noisy tail probabilities (as APS does) and instead relies on the more stable top probability and the rank ordering.

### E.3 Evaluation Metrics

We assess the performance of all methods using a target miscoverage level $\alpha \in \{0.01, 0.025, 0.05, 0.1\}$. Let $\{(x_i, y_i)\}_{i=1}^{n_{\text{test}}}$ be the test set. The primary metrics are:

- **Empirical Coverage:** The fraction of test samples where the true label is included in the prediction set.

$$\text{Coverage} = \frac{1}{n_{\text{test}}} \sum_{i=1}^{n_{\text{test}}} \mathbb{I}[y_i \in \mathcal{C}(x_i)] \tag{23}$$

- **Macro-Coverage (MacroCov):** While empirical coverage reflects marginal reliability over the entire test distribution, MacroCov measures the average per-class coverage, giving each class equal weight regardless of its frequency. Let $\hat{c}_y = \frac{1}{|I_y|} \sum_{i \in I_y} \mathbb{I}[y_i \in \mathcal{C}(x_i)]$ denote the empirical coverage for class $y$, where $I_y = \{i : y_i = y\}$. Then,

$$\text{MacroCov} = \frac{1}{K} \sum_{y=1}^{K} \hat{c}_y. \tag{24}$$

This metric is particularly informative in imbalanced or long-tailed settings, since it prevents head classes from dominating the overall coverage and highlights systematic under-coverage of rare classes.

- **Average Prediction Set Size:** The mean size of the prediction sets over the test data.

$$\text{Size} = \frac{1}{n_{\text{test}}} \sum_{i=1}^{n_{\text{test}}} |\mathcal{C}(x_i)| \tag{25}$$

To assess class-conditional reliability, we report the following metrics to verify that they are maintained while reducing the average prediction set size:

- **Average Class Coverage Gap (CovGap):** This metric measures the average absolute deviation of per-class coverage from the target coverage level $1 - \alpha$ (Ding et al., 2023b). Let $I_y = \{i : y_i = y\}$ be the indices of test samples for class $y$. The empirical coverage for class $y$ is $\hat{c}_y = \frac{1}{|I_y|} \sum_{i \in I_y} \mathbb{I}[y_i \in \mathcal{C}(x_i)]$. The gap is then:

$$\text{CovGap} = \frac{1}{K} \sum_{y=1}^{K} |\hat{c}_y - (1 - \alpha)| \tag{26}$$

We report this value as a percentage.

To measure the adaptiveness of the prediction sets, we use:

- **Size-Stratified Coverage Violation (SSCV):** As introduced by Angelopoulos et al. (2021), SSCV quantifies whether coverage is maintained across different prediction set sizes. We define disjoint set-size strata $\{S_j\}_{j=1}^{s}$ and group test indices into bins $\mathcal{J}_j = \{i : |\mathcal{C}(x_i)| \in S_j\}$. The SSCV is the maximum deviation from the target coverage across all bins:

$$\text{SSCV} = \sup_j \left| \frac{|\{i \in \mathcal{J}_j : y_i \in \mathcal{C}(x_i)\}|}{|\mathcal{J}_j|} - (1 - \alpha) \right| \tag{27}$$

To probe feature-conditional reliability beyond class-conditional metrics, we report the worst-slab coverage (Cauchois et al., 2021):

- **Worst-Slab Coverage (WSC):** Let $v \in \mathbb{R}^d$ be a unit direction and define a *slab* along $v$ as

$$\mathcal{S}(v, a, b) = \{x \in \mathbb{R}^d : a \le v^\top x \le b\}, \tag{28}$$

for thresholds $a < b$. For a mass parameter $\delta \in (0, 1]$, the empirical WSC along direction $v$ is

$$\text{WSC}_n(v; \delta) = \inf_{\substack{a < b: \\ \hat{P}_n(X \in \mathcal{S}(v,a,b)) \ge \delta}} \hat{P}_n(y \in \mathcal{C}(x) \mid x \in \mathcal{S}(v, a, b)), \tag{29}$$

where $\hat{P}_n$ denotes the empirical distribution over the test set. We estimate overall WSC by scanning a set of directions $\mathcal{V}$ (e.g., random unit vectors) and taking the worst case:

$$\text{WSC}_n(\delta) = \inf_{v \in \mathcal{V}} \text{WSC}_n(v; \delta). \tag{30}$$

Larger values indicate better approximate conditional coverage, while low WSC reveals a feature-space region of mass at least $\delta$ where the prediction sets under-cover.

# F  REPRODUCIBILITY DETAILS

To ensure reproducibility, we detail our experimental setup, key hyperparameters, and implementation. All source code will be made publicly available. All experiments are implemented based on the **TorchCP** library (Huang et al., 2024a), which provides a robust framework for conformal prediction on deep learning models. The pre-trained backbone models are sourced from **TorchVision** (maintainers & contributors, 2016).

## F.1  COMPUTATIONAL ENVIRONMENT

- **Operating System:** Linux kernel 5.14.0-427.42.1.el9_4.x86_64.
- **GPU Hardware:** NVIDIA H100 80GB HBM3.

- **CPU Hardware:** 8 cores.
- **System Memory:** 32 GB RAM.
- **NVIDIA Driver Version:** 550.144.03.
- **CUDA Version:** 12.2.
- **Python Version:** 3.9.21.
- **PyTorch Version:** 2.0.0+ (with CUDA support).

## F.2 DATASETS AND MODELS

Our experiments are conducted on several standard image classification benchmarks: **CIFAR-100** (Krizhevsky et al., 2009), **ImageNet-Val** (Deng et al., 2009), and **Places365** (Zhou et al., 2018). These benchmarks are chosen because they contain a large number of classes, which makes performance differences between methods more evident. We use pre-trained ResNet (He et al., 2016), VGG (Simonyan & Zisserman, 2015), ViT (Dosovitskiy et al., 2021), Swin Transformer (Liu et al., 2021), EfficientNet (Tan & Le, 2019), and ShuffleNet (Zhang et al., 2018) architectures from TorchVision as our base classifiers. To evaluate performance under distributional shift, we use a model trained on CIFAR-100 and test its out-of-distribution (OOD) performance on the Places365 dataset.

To investigate the methods' robustness to class imbalance, we create four long-tailed variants of CIFAR-100, denoted as **CIFAR-100-LT**. The number of training samples for class $j \in \{1, \ldots, 100\}$, denoted $n_j$, is set to be proportional to $\exp(-\lambda \cdot j)$. The imbalance factor $\lambda \in \{0.005, 0.01, 0.02, 0.03\}$ controls the severity of the class imbalance, with larger values of $\lambda$ creating a more pronounced long-tail distribution. For evaluation, we have considered two scenarios: (i) the calibration and test sets are balanced while the training data remain imbalanced, and (ii) the calibration and test sets follow the same imbalance ratios as the training data.

## F.3 HYPERPARAMETER SETTINGS

In our experiments, we set $k_{\text{reg}} = 2$ and $\lambda = 0.2$ for RAPS and we use $\lambda = 0.2$ for SAPS. The softmax probabilities $\hat{\pi}(y|x)$ used by all scores are computed with a temperature parameter $T$, while the free energy $F_\tau(x)$ is calculated with its own temperature $\tau$. Crucially, to ensure a fair comparison, the softmax temperature $T$ was tuned for all baseline and proposed methods to optimize their performance. We tune the temperature energy parameter $\tau$ with $\ln(\tau) \in [-9, 9]$ and the calibration temperature $T \in \{0.01, \ldots, 25\}$.

*Remark* F.1. A critical consideration in our proposed modulation is the positivity of the reweighting factor. When reweighting a base nonconformity score, it is critical the scaling factor must be strictly positive. A negative factor would reverse the score's ordering, invalidating the fundamental assumption of conformal prediction that lower scores indicate higher conformity. While our uncertainty signal is the negative free energy, $-F(x)$, it is not guaranteed to be positive. Mathematically, $-F(x)$ becomes negative if $\sum_{k=1}^{K} \exp(f_k(x)/\tau) < 1$, a condition which implies that the maximum logit $f_{\max}(x)$ is negative (a necessary, though not always sufficient, condition). This scenario signifies extreme model uncertainty, where the model lacks evidence for any class and typically occurs only for far out-of-distribution inputs.

Although we empirically observe that $-F(x)$ is positive for nearly all in-distribution and OOD samples in our experiments, to ensure the theoretical robustness of our method, we scale the base scores by the softplus of the negative free energy. The hyperparameter $\beta$ in the softplus function, $\frac{1}{\beta} \log(1 + e^{\beta z})$, controls its approximation to the Rectified Linear Unit (ReLU) function. By choosing a large value for $\beta$, the scaling factor $\text{softplus}(-F(x))$ behaves almost identically to $-F(x)$ when it is positive, but smoothly transitions to a value near zero in the rare cases where $-F(x) < 0$. This behavior is highly beneficial for conformal prediction. For such uncertain inputs, the near-zero scaling factor drives the modulated scores for all labels toward zero, causing most or all of them to fall below the conformal quantile $\hat{q}$. This correctly produces an extensively large prediction set, signaling the model's high epistemic uncertainty. Throughout our experiments, we set $\beta = 1$. See Appendix O for a detailed ablation study on $\beta$.

# G    PROOFS

## G.1    THEORETICAL VALIDITY OF ENERGY-BASED SCORES

In conformal prediction, the validity of the coverage guarantee relies on the exchangeability of the nonconformity scores. Specifically, for a set of exchangeable data points, their corresponding nonconformity scores must also be exchangeable. This property ensures that the prediction sets contain the true label with the desired probability. We now establish that our proposed Energy-modulated scores, as defined in Equation 9, satisfy this critical property under standard assumptions. This ensures that using these modulated scores yields valid prediction sets with the guaranteed marginal coverage central to conformal prediction theory (Vovk et al., 2005).

**Theorem G.1** (Exchangeability of Energy-Based Nonconformity Scores). *Let $(X_i, Y_i)_{i=1}^{n+1}$ be an exchangeable sequence of random variables drawn from a distribution $P_{XY}$. Assume that:*

*(i) The base nonconformity score $S(x, y)$ is a deterministic function of its arguments.*

*(ii) The free energy $F(x)$ is a deterministic function of $x$ as defined in Equation 4.*

*Define the modulated score for $i = 1, \ldots, n+1$ as:*

$$S_i' = S_{Energy\text{-}based}(X_i, Y_i) := S(X_i, Y_i) \cdot \frac{1}{\beta} \log\left(1 + e^{-\beta F(X_i)}\right). \tag{31}$$

*Then the sequence of modulated scores $(S_1', \ldots, S_{n+1}')$ is exchangeable.*

*Proof.* A sequence of random variables is exchangeable if its joint distribution is invariant under any finite permutation of its indices.

Let the transformation be defined as $h(x, y) = S(x, y) \cdot \frac{1}{\beta} \log\left(1 + e^{-\beta F(x)}\right)$. By assumptions (i) and (ii), the base score $S(x, y)$ and the energy function $F(x)$ are deterministic. Since the softplus function and multiplication are also deterministic operations, the entire mapping $h(x, y)$ is deterministic and measurable.

A fundamental property of exchangeable sequences is that they remain exchangeable after applying a measurable transformation. That is, if $(Z_i)$ is an exchangeable sequence and $g$ is a measurable function, then the sequence $(g(Z_i))$ is also exchangeable.

Applying this principle with $Z_i = (X_i, Y_i)$ and the transformation $g = h$, we find that the sequence of scores $(S_i') = (h(X_i, Y_i))$ inherits exchangeability from the data sequence $((X_i, Y_i))$. Formally, for any permutation $\sigma$ of $\{1, \ldots, n+1\}$,

$$(S_{\sigma(1)}', \ldots, S_{\sigma(n+1)}') \overset{d}{=} (S_1', \ldots, S_{n+1}'), \tag{32}$$

where $\overset{d}{=}$ denotes equality in distribution. Hence, the sequence $(S_i')_{i=1}^{n+1}$ is exchangeable. $\square$

## G.2    PROOF OF PROPOSITION 2.1

*Proof.* Let the epistemic uncertainty $U_E(x)$ be defined as the negative logarithm of the model-induced input density, $U_E(x) = -\log p(x)$. This definition captures the intuition that uncertainty is high where the model assigns low probability density.

Starting from the definition of the input density in Equation 5:

$$p(x) = \frac{\exp(-F(x)/\tau)}{Z}.$$

Taking the logarithm of both sides yields:

$$\log p(x) = \log\left(\exp(-F(x)/\tau)\right) - \log Z$$
$$\log p(x) = -F(x)/\tau - \log Z.$$

Multiplying by $-1$ and rearranging for $F(x)$, we obtain:

$$-\log p(x) = F(x)/\tau + \log Z$$
$$F(x) = \tau(-\log p(x)) - \tau \log Z.$$

Substituting $U_E(x) = -\log p(x)$ and letting $C = -\tau \log Z$ (a constant with respect to $x$), we arrive at:

$$F(x) = \tau \cdot U_E(x) + C.$$

This shows that the free energy $F(x)$ is linearly proportional to the epistemic uncertainty $U_E(x)$, scaled by the temperature $\tau$ and shifted by a constant. Therefore, a higher free energy value directly corresponds to higher epistemic uncertainty. $\square$

### G.3 PROOF OF THEOREM 2.2

*Proof.* The proof proceeds by first establishing the relationship between the negative free energy $-F(x)$ and the maximum logit, and then arguing that the expected maximum logit decreases as sample difficulty increases for a well-trained model.

**Step 1: Relating Negative Free Energy to the Maximum Logit.** The negative free energy, $-F(x)$, is the LogSumExp (LSE) of the scaled logits. The LSE function is a smooth approximation of the maximum function and is tightly bounded by it. For any vector $\mathbf{z} \in \mathbb{R}^K$, the sum of exponentials can be bounded relative to its maximum term, $z_{\max} = \max_k z_k$:

$$e^{z_{\max}} \leq \sum_{k=1}^{K} e^{z_k} \leq K \cdot e^{z_{\max}}. \tag{33}$$

By taking the logarithm across all parts of the inequality, we obtain the standard bounds for the LSE function:

$$\max_k z_k \leq \log \sum_{k=1}^{K} e^{z_k} \leq \max_k z_k + \log K. \tag{34}$$

Applying this to our scaled logits, $z_k = f_k(x)/\tau$, and multiplying by $\tau$ gives:

$$\max_k f_k(x) \leq -F(x) \leq \max_k f_k(x) + \tau \log K. \tag{35}$$

This inequality demonstrates that $-F(x)$ is a tight, monotonically increasing function of the maximum logit, $\max_k f_k(x)$. Therefore, proving Theorem 2.2 is equivalent to proving that the expected maximum logit is a strictly monotonically decreasing function of difficulty:

$$\mathbb{E}[\max_k f_k(X) \mid D(X, Y_{\text{true}}) = d_1] > \mathbb{E}[\max_k f_k(X) \mid D(X, Y_{\text{true}}) = d_2]. \tag{36}$$

**Step 2: Characterizing the Maximum Logit by Difficulty Level.** We analyze the maximum logit for samples conditioned on their difficulty.

- **Low Difficulty** ($d = 1$): A sample $(x, y)$ has difficulty $d = 1$ if and only if its true label $y$ receives the highest logit. Thus, for this subpopulation of data, the maximum logit is the logit of the true class:

$$D(x, y_{\text{true}}) = 1 \implies \max_k f_k(x) = f_y(x). \tag{37}$$

  A model trained via a standard objective like cross-entropy is explicitly optimized to increase the value of $f_y(x)$ for all training samples. Consequently, the set of samples where the model succeeds ($d = 1$) corresponds to inputs for which the model produces a large, dominant logit for the correct class.

- **High Difficulty** ($d > 1$): A sample $(x, y)$ has difficulty $d > 1$ if and only if the model's prediction is incorrect. This implies that the maximum logit corresponds to an incorrect class $k' \neq y$:

$$D(x, y_{\text{true}}) = d > 1 \implies \max_k f_k(x) = f_{k'}(x) \text{ for some } k' \neq y. \tag{38}$$

**Step 3: Comparing Conditional Expectations.** We compare the expected maximum logit over the subpopulation of correctly classified samples ($d_1 = 1$) versus incorrectly classified samples ($d_2 > 1$). The training objective directly pushes the values in the set $\{f_Y(X) \mid D(X, Y_{\text{true}}) = 1\}$ to be as large as possible. In contrast, the values in the set $\{\max_k f_k(X) \mid D(X, Y_{\text{true}}) > 1\}$ arise from the model's failure to generalize.

A fundamental property of a successfully trained and well-generalized, and well-calibrated classifier is that its confidence on the examples it classifies correctly is, on average, higher than its confidence on the examples it classifies incorrectly (Guo et al., 2017). If this were not the case, the model would not have learned a meaningful decision boundary from the data. Thus, the average maximum logit for the population of "easy" samples must be greater than that for the population of "hard" samples. Formally, for $d_1 < d_2$:

$$\mathbb{E}[\max_k f_k(X) \mid D(X, Y_{\text{true}}) = d_1] > \mathbb{E}[\max_k f_k(X) \mid D(X, Y_{\text{true}}) = d_2]. \tag{39}$$

Given the monotonic relationship established in Equation 35, it follows directly that the expected negative free energy also decreases with increasing difficulty. This completes the proof and aligns with our empirical observation in Section 2. □

### G.4 PROOF OF PROPOSITION 2.3

*Proof.* The proof follows directly from the definition of a conformal prediction set. By definition, the prediction set $\mathcal{C}_G(x)$ for a new instance $x$ includes all labels $y$ for which the scaled nonconformity score does not exceed the calibrated quantile $\widehat{q}_{1-\alpha}^{(G)}$.

$$\mathcal{C}_G(x) = \left\{ y \;\middle|\; S_G(x, y) \leq \widehat{q}_{1-\alpha}^{(G)} \right\}. \tag{40}$$

Substituting the definition of the scaled score, $S_G(x, y) = G(x)S(x, y)$, we have:

$$\mathcal{C}_G(x) = \left\{ y \;\middle|\; G(x)S(x, y) \leq \widehat{q}_{1-\alpha}^{(G)} \right\}. \tag{41}$$

Since we assume $G(x)$ is strictly positive, we can divide both sides of the inequality by $G(x)$ without changing its direction:

$$\mathcal{C}_G(x) = \left\{ y \;\middle|\; S(x, y) \leq \frac{\widehat{q}_{1-\alpha}^{(G)}}{G(x)} \right\}. \tag{42}$$

By defining the instance-adaptive threshold $\theta(x) = \widehat{q}_{1-\alpha}^{(G)}/G(x)$, we arrive at the equivalent formulation:

$$\mathcal{C}_G(x) = \{y \mid S(x, y) \leq \theta(x)\}. \tag{43}$$

This concludes the proof. □

*Remark* G.2. It is important to emphasize that the new quantile, $\widehat{q}_{1-\alpha}^{(G)}$, is fundamentally different from the baseline quantile, $\widehat{q}_{1-\alpha}$, which would be computed from the unscaled scores.

Let $\mathcal{S}_{\text{base}} = \{S(X_i, Y_i)\}_{i=1}^n$ be the set of baseline calibration scores, and $\mathcal{S}_G = \{G(X_i)S(X_i, Y_i)\}_{i=1}^n$ be the set of Energy-reweighted calibration scores.

- $\widehat{q}_{1-\alpha}$ is the $(1 - \alpha)$-quantile of the empirical distribution defined by $\mathcal{S}_{\text{base}}$.

- $\widehat{q}_{1-\alpha}^{(G)}$ is the $(1 - \alpha)$-quantile of the empirical distribution defined by $\mathcal{S}_G$.

There is no simple, closed-form relationship between $\widehat{q}_{1-\alpha}$ and $\widehat{q}_{1-\alpha}^{(G)}$. Scaling each score $S_i$ by a different factor $G(X_i)$ changes the distribution of scores, including the relative ordering of the calibration samples. For instance, a sample $(X_j, Y_j)$ that had a median score in $\mathcal{S}_{\text{base}}$ might have a very high score in $\mathcal{S}_G$ if its corresponding energy factor $G(X_j)$ is large.

Consequently, the sample that happens to fall at the $\lceil(1 - \alpha)(n + 1)\rceil$-th position (thus defining the quantile) will almost certainly be different in the baseline and Energy-based cases. In other words, one cannot simply take the baseline quantile $\widehat{q}_{1-\alpha}$ and scale it by some factor. The entire set of calibration scores must be re-computed and re-sorted to find the new, correct quantile $\widehat{q}_{1-\alpha}^{(G)}$.

## G.5 Proof of Theorem 3.1

*Proof.* The proof relies on the connection between the logits of a classifier trained with cross-entropy and the Bayesian posterior probability, which shows that the model's parameters internalize the training set's class priors.

**Step 1: Bayesian Decomposition of Logits.** A classifier trained to minimize cross-entropy loss learns to approximate the posterior probability $P(Y = y|X = x)$. Its logits $f_y(x)$ thus approximate the log-posterior, up to an instance-specific normalization constant. Using Bayes' rule, we can decompose the log-posterior:

$$\log P(Y = y|X = x) = \log P(X = x|Y = y) + \log P(Y = y) - \log P(X = x). \tag{44}$$

When trained on $P_{\text{train}}$, the model's logits learn to reflect this structure:

$$f_y(x) \approx \log P_{\text{train}}(X = x|Y = y) + \log P_{\text{train}}(Y = y) + C(x), \tag{45}$$

where the term $C(x)$ absorbs instance-dependent factors like $-\log P_{\text{train}}(X = x)$ and other model-specific biases. Critically, the logit $f_y(x)$ encodes the log-prior probability of class $y$ from the training distribution.

**Step 2: Connecting Negative Free Energy to the Maximum Logit.** As established in the proof of Theorem 2.2, the negative free energy $-F(x)$ is tightly and monotonically related to the maximum logit, $-F(x) \approx \max_k f_k(x)$. For a reasonably accurate classifier, the expectation of the maximum logit over test samples from a given class $y$ is dominated by instances where the model is correct. For a correct classification of a sample $(x, y)$, the maximum logit is the logit of the true class: $\max_k f_k(x) = f_y(x)$. Building on this, we can state that the expected negative free energy for class $y$ is primarily driven by the expected logit for that class:

$$\mathbb{E}_{X \sim P_{\text{test}}(X|Y=y)}[-F(X)] \approx \mathbb{E}_{X \sim P_{\text{test}}(X|Y=y)}[f_y(X)]. \tag{46}$$

**Step 3: Comparing Expected Logits for Majority and Minority Classes.** Using the decomposition from Equation 45, we can express the expected logit for a class $y$ as:

$$\mathbb{E}_{X \sim P_{\text{test}}(X|Y=y)}[f_y(X)] \approx \mathbb{E}_{X \sim P_{\text{test}}(X|Y=y)}[\log P_{\text{train}}(X|Y = y) + C(X)] + \log P_{\text{train}}(Y = y). \tag{47}$$

Let us define the term $A(y) = \mathbb{E}_{X \sim P_{\text{test}}(X|Y=y)}[\log P_{\text{train}}(X|Y = y) + C(X)]$. This term represents the average "data-fit" or "evidence" for class $y$, as learned by the model. Under our assumption that classes $y_{\text{maj}}$ and $y_{\text{min}}$ have comparable intrinsic complexity and are well-represented, this evidence term should be similar for both, i.e., $A(y_{\text{maj}}) \approx A(y_{\text{min}})$.

We can now compare the expected negative free energy for the two classes:

$$\mathbb{E}[-F(X)|Y = y_{\text{maj}}] \approx A(y_{\text{maj}}) + \log P_{\text{train}}(Y = y_{\text{maj}}) \tag{48}$$

$$\mathbb{E}[-F(X)|Y = y_{\text{min}}] \approx A(y_{\text{min}}) + \log P_{\text{train}}(Y = y_{\text{min}}) \tag{49}$$

By the proposition's premise, $P_{\text{train}}(Y = y_{\text{maj}}) > P_{\text{train}}(Y = y_{\text{min}})$, which implies $\log P_{\text{train}}(Y = y_{\text{maj}}) > \log P_{\text{train}}(Y = y_{\text{min}})$. Since $A(y_{\text{maj}}) \approx A(y_{\text{min}})$, the additive log-prior term learned during training becomes the dominant factor driving the difference. Therefore, we conclude that:

$$\mathbb{E}_{X \sim P_{\text{test}}(X|Y=y_{\text{maj}})}[-F(X)] > \mathbb{E}_{X \sim P_{\text{test}}(X|Y=y_{\text{min}})}[-F(X)]. \tag{50}$$

This result confirms that the model's systematically higher epistemic uncertainty (lower negative free energy) for minority classes is a bias inherited from the training distribution's class priors. This aligns with prior work showing that models have lower expected logits for minority classes (Ren et al., 2020; Lyu et al., 2025; Kato & Hotta, 2023; Chen & Su, 2023). □

## H  ANALYSIS OF SOFTMAX SATURATION AND ENERGY-BASED ADAPTIVITY

In this section, we provide a formal mechanism linking the limitations of softmax-based nonconformity scores to conformal inefficiency. We demonstrate that while softmax probabilities saturate rapidly away from the decision boundary, thereby losing information about sample difficulty, the Helmholtz Free Energy retains this geometric information. We validate this analysis with a toy experiment visualizing the decision landscapes.

### H.1  DISTANCE TO DECISION BOUNDARY AND LOGIT MAGNITUDE

Consider a deep classifier $f : \mathcal{X} \to \mathbb{R}^K$. For a given input $x$, let $\hat{y} = \arg\max_k f_k(x)$ be the predicted class. The decision boundary between the predicted class $\hat{y}$ and the second most likely class $j$ is defined by the hyperplane where $f_{\hat{y}}(x) = f_j(x)$.

It has been established that for neural networks, the magnitude of the logit vector typically scales with the distance of the input from the decision boundary (Hein et al., 2019). Let $\delta(x)$ denote the geometric distance of $x$ from the decision boundary. We observe the following proportionality:

$$\delta(x) \propto \max_k f_k(x). \tag{51}$$

Therefore, an "easy" sample (one far from the boundary in a high-density region) is characterized by logits with large magnitudes, while a "hard" sample (near the boundary) yields logits with smaller or entangled magnitudes. Ideally, an adaptive conformal predictor should produce smaller sets as $\delta(x)$ increases.

### H.2  THE SATURATION OF SOFTMAX AND ENTROPY

Standard conformal scores rely on the softmax distribution $\hat{\pi}(y|x) = \exp(f_y(x))/\sum_k \exp(f_k(x))$. A critical limitation of this mapping is *gradient saturation*.

Consider a sample $x$ moving away from the decision boundary such that its logit magnitude scales by a factor $\alpha > 1$. As $\alpha \to \infty$, $\hat{\pi}(\hat{y}|x) \to 1$. The gradient of the softmax output with respect to the dominant logit $f_{\hat{y}}$ is given by:

$$\frac{\partial \hat{\pi}(\hat{y}|x)}{\partial f_{\hat{y}}} = \hat{\pi}(\hat{y}|x)(1 - \hat{\pi}(\hat{y}|x)). \tag{52}$$

As $\hat{\pi} \to 1$, this gradient approaches 0. Similarly, the Shannon Entropy $\mathbb{H}(x)$ of the distribution approaches 0.

**Implication for Conformal Prediction:** Once a sample is sufficiently far from the boundary to achieve a high confidence (e.g., $\hat{\pi} > 0.99$), the softmax score saturates. The model becomes geometrically insensitive: a sample at distance $d$ and a sample at distance $10d$ yield indistinguishable conformal scores. This saturation restricts the adaptive capacity of the prediction sets. For high-confidence samples, the sets cannot achieve greater efficiency because the score yields no further signal.

### H.3  NON-SATURATION OF FREE ENERGY

In contrast, the negative Helmholtz Free Energy is defined as $-F(x) = \tau \log \sum_k \exp(f_k(x)/\tau)$. As derived in Appendix G.3, this quantity is bounded by the maximum logit:

$$-F(x) \approx \max_k f_k(x). \tag{53}$$

Unlike softmax, the Free Energy does not saturate. Its derivative with respect to the dominant logit is approximately 1:

$$\frac{\partial(-F(x))}{\partial f_{\hat{y}}} \approx 1. \tag{54}$$

This indicates that $-F(x)$ grows linearly with the logit magnitude, thereby acting as a faithful proxy for the distance $\delta(x)$ even in high-confidence regimes.

## H.4 EMPIRICAL VISUALIZATION OF UNCERTAINTY LANDSCAPES

To empirically validate this behavior, we trained a 3-layer Multilayer Perceptron (MLP) on a 2D toy dataset consisting of two concentric classes. Figure 6 visualizes the value of Max Softmax Confidence, Shannon Entropy, and Negative Free Energy across the input space $\mathcal{X}$.

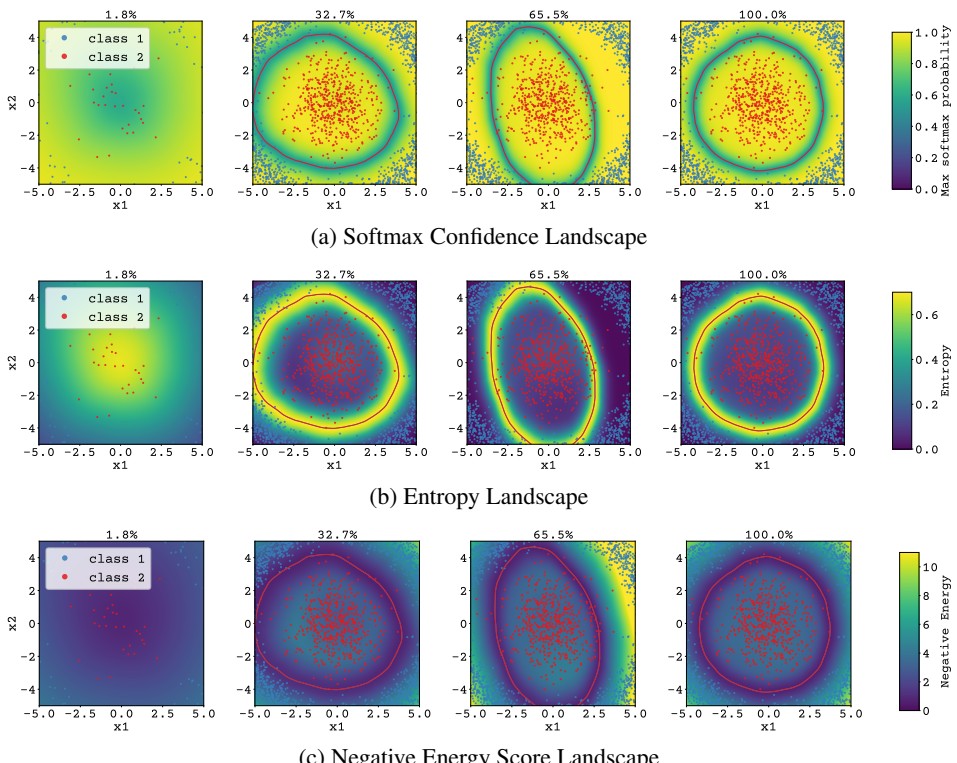

(a) Softmax Confidence Landscape

(b) Entropy Landscape

(c) Negative Energy Score Landscape

Figure 6: Evolution of uncertainty landscapes on a 2D toy dataset throughout the training process. Columns represent progressive checkpoints from the early training phase (left) to full convergence (right). The red line indicates the decision boundary. (a) Softmax probabilities saturate rapidly. The yellow region (confidence $\approx 1.0$) is flat, making points near the boundary indistinguishable from points far away. (b) Entropy exhibits similar saturation (dark blue region), vanishing to zero for most of the domain. (c) Negative Free Energy retains gradients throughout the domain. Note the continuous color transition scaling with the distance from the decision boundary, identifying "easier" points with higher values even when softmax is saturated.

## H.5 MECHANISM OF ENERGY-BASED EFFICIENCY

Our proposed method leverages this non-saturating property by modulating the base nonconformity score $S(x, y)$ with a sample-specific scaler $G(x) \propto \mathrm{softplus}(-F(x))$.

For "easy" samples (large $\delta(x)$), $-F(x)$ is large positive. This results in a large scaling factor $G(x) \gg 1$.

1. The nonconformity scores for incorrect labels (which are naturally non-zero) are magnified significantly by $G(x)$, pushing them well above the calibrated quantile $\hat{q}$.

2. The nonconformity score for the true label (typically near zero) remains small even after scaling.

This amplification forces the exclusion of incorrect classes that might otherwise have been included due to the looseness of the global quantile $\hat{q}$, thereby reducing the prediction set size. Because $F(x)$ does not saturate, this efficiency gain continues to improve as samples get "easier," a property unattainable with softmax-based modulation.

# I    PERFORMANCE ANALYSIS STRATIFIED BY SAMPLE DIFFICULTY

In this section, we report the performance of different nonconformity scores by stratifying test samples based on their difficulty. Sample difficulty is defined as the rank of the true label in the model's predicted probability ordering; a lower rank indicates an "easier" sample, while a higher rank signifies a "harder", often misclassified, one.

In Section 3.1, we established that energy-reweighting reduces the overall average prediction set size while maintaining target coverage. However, this aggregate metric does not reveal whether these efficiency gains are distributed evenly or are concentrated on specific types of samples. This stratified analysis provides a more granular view to answer that question.

We expect an adaptive method to produce the most significant set size reductions for easy samples, where the model is confident, while appropriately adjusting for harder samples where uncertainty is higher. Table 6 presents these stratified results, detailing how coverage and average set size vary across the different difficulty levels.

Table 6: Coverage and average set size on ImageNet, stratified by sample difficulty. Results are for a ResNet-50 model at $\alpha = 0.01$ and averaged over 10 trials. The table compares baseline adaptive scores with their energy-based variants, which generally produce smaller sets for easier samples while maintaining coverage.

| | | APS | | | | RAPS | | | | SAPS | | | |
| | | w/o Energy | | w/ Energy | | w/o Energy | | w/ Energy | | w/o Energy | | w/ Energy | |
| Difficulty | Count | Cov. | Set Size | Cov. | Set Size | Cov. | Set Size | Cov. | Set Size | Cov. | Set Size | Cov. | Set Size |
|---|---|---|---|---|---|---|---|---|---|---|---|---|---|
| 1 to 1 | 15990 | 1.00 | 39.08 | 1.00 | 32.31 | 1.00 | 36.97 | 1.00 | 30.53 | 1.00 | 35.69 | 1.00 | 29.11 |
| 2 to 3 | 2547 | 1.00 | 38.38 | 1.00 | 34.96 | 1.00 | 37.17 | 1.00 | 34.10 | 1.00 | 36.98 | 1.00 | 33.69 |
| 4 to 6 | 638 | 1.00 | 37.86 | 1.00 | 36.46 | 1.00 | 37.32 | 1.00 | 36.29 | 1.00 | 37.35 | 1.00 | 36.02 |
| 7 to 10 | 306 | 1.00 | 37.66 | 1.00 | 37.09 | 1.00 | 37.40 | 1.00 | 37.20 | 1.00 | 37.47 | 1.00 | 36.96 |
| 11 to 100 | 453 | 0.76 | 37.49 | 0.76 | 37.84 | 0.76 | 37.51 | 0.76 | 38.27 | 0.76 | 37.58 | 0.76 | 37.98 |
| 101 to 1000 | 66 | 0.00 | 37.82 | 0.00 | 38.45 | 0.00 | 37.60 | 0.00 | 38.60 | 0.00 | 37.34 | 0.00 | 38.05 |

## J    ENERGY-BASED LAC

The standard LAC score is inversely proportional to the softmax probability. For its energy-based variant, as mentioned in Appendix E.2, we use a base score of $S_{\text{LAC}}(x, y) = -\hat{\pi}(y|x)$. For a difficult input, which corresponds to a low negative free energy score, the objective is to produce a larger prediction set. This requires the nonconformity scores of more classes to fall below the calibrated threshold. Conversely, for an easy input (high negative free energy), the nonconformity scores should be scaled to produce a smaller set. Therefore, for the energy-based LAC, we divide the base score by the energy-based scaling factor. This adjustment ensures that for difficult inputs, the small scaling factor in the denominator makes the nonconformity scores smaller, including more labels in the set.

So formally, we define the Energy-based LAC nonconformity score as:

$$S_{\text{EB-LAC}}(x, y) = \frac{-\hat{\pi}(y|x)}{\frac{1}{\beta} \log(1 + e^{-\beta F(x)})} \tag{55}$$

where $\hat{\pi}(y|x)$ is the softmax probability, $F(x)$ is the Helmholtz free energy, and $\beta$ is the softplus sharpness parameter.

### J.1    ENERGY-BASED LAC PERFORMANCE IN STANDARD SCENARIO

Table 7: Performance of the LAC nonconformity score function and its energy-based variant on CIFAR-100, ImageNet, and Places365 at miscoverage levels $\alpha \in \{0.01, 0.025, 0.05, 0.1\}$. Results are averaged over 10 trials. For the **Set Size** column, lower is better. **Bold** values indicate the best performance within the method family (with and without Energy).

| Method | | $\alpha = 0.1$ | | $\alpha = 0.05$ | | $\alpha = 0.025$ | | $\alpha = 0.01$ | |
|---|---|---|---|---|---|---|---|---|---|
| | | Coverage | Set Size | Coverage | Set Size | Coverage | Set Size | Coverage | Set Size |
| | | **CIFAR-100** (ResNet-56) | | | | | | | |
| LAC | w/o Energy | $0.90 \pm 0.01$ | $2.54 \pm 0.06$ | $0.95 \pm 0.00$ | $5.26 \pm 0.17$ | $0.974 \pm 0.001$ | $9.50 \pm 0.21$ | $0.99 \pm 0.00$ | $21.14 \pm 0.74$ |
| | w/ Energy | $0.90 \pm 0.01$ | $\mathbf{2.52} \pm 0.04$ | $0.95 \pm 0.00$ | $\mathbf{5.21} \pm 0.15$ | $0.974 \pm 0.002$ | $\mathbf{9.09} \pm 0.16$ | $0.99 \pm 0.00$ | $\mathbf{20.65} \pm 0.83$ |
| | | **ImageNet** (ResNet-50) | | | | | | | |
| LAC | w/o Energy | $0.90 \pm 0.00$ | $1.49 \pm 0.01$ | $0.95 \pm 0.00$ | $\mathbf{2.68} \pm 0.05$ | $0.975 \pm 0.001$ | $5.48 \pm 0.18$ | $0.99 \pm 0.00$ | $14.79 \pm 0.80$ |
| | w/ Energy | $0.90 \pm 0.00$ | $\mathbf{1.48} \pm 0.01$ | $0.95 \pm 0.00$ | $\mathbf{2.68} \pm 0.04$ | $0.975 \pm 0.001$ | $\mathbf{5.42} \pm 0.15$ | $0.99 \pm 0.00$ | $\mathbf{13.89} \pm 0.64$ |
| | | **Places365** (ResNet-50) | | | | | | | |
| LAC | w/o Energy | $0.90 \pm 0.001$ | $6.21 \pm 0.03$ | $0.95 \pm 0.00$ | $11.36 \pm 0.04$ | $0.973 \pm 0.001$ | $19.55 \pm 0.41$ | $0.99 \pm 0.001$ | $37.12 \pm 0.93$ |
| | w/ Energy | $0.90 \pm 0.001$ | $\mathbf{6.19} \pm 0.03$ | $0.95 \pm 0.001$ | $\mathbf{11.11} \pm 0.11$ | $0.973 \pm 0.001$ | $\mathbf{19.28} \pm 0.43$ | $0.99 \pm 0.001$ | $\mathbf{35.28} \pm 0.63$ |

## J.2 ENERGY-BASED LAC PERFORMANCE WITH IMBALANCED TRAINING PRIORS

Table 8: Performance of `LAC` and its energy-based variant on imbalanced CIFAR-100 with varying imbalance factors ($\lambda \in \{0.005, 0.01, 0.02, 0.03\}$) and at miscoverage levels $\alpha \in \{0.01, 0.025, 0.05, 0.1\}$. Results are averaged over 10 trials with a ResNet-56 model. For the average set size, lower is better. **Bold** values indicate the best performance.

| Method | | $\alpha = 0.1$ Coverage | Set Size | $\alpha = 0.05$ Coverage | Set Size | $\alpha = 0.025$ Coverage | Set Size | $\alpha = 0.01$ Coverage | Set Size |
|---|---|---|---|---|---|---|---|---|---|
| | | **CIFAR-100-LT ($\lambda = 0.005$, mild imbalance) (ResNet-56)** | | | | | | | |
| LAC | w/o Energy | $0.897 \pm 0.007$ | $7.04 \pm 0.24$ | $0.947 \pm 0.004$ | $12.98 \pm 0.49$ | $0.973 \pm 0.003$ | $21.31 \pm 0.84$ | $0.988 \pm 0.002$ | $33.97 \pm 1.37$ |
| LAC | w/ Energy | $0.897 \pm 0.006$ | $\mathbf{6.91} \pm 0.19$ | $0.947 \pm 0.005$ | $\mathbf{12.68} \pm 0.51$ | $0.973 \pm 0.004$ | $\mathbf{20.56} \pm 1.01$ | $0.989 \pm 0.002$ | $\mathbf{32.99} \pm 0.87$ |
| | | **CIFAR-100-LT ($\lambda = 0.01$) (ResNet-56)** | | | | | | | |
| LAC | w/o Energy | $0.900 \pm 0.007$ | $11.92 \pm 0.41$ | $0.951 \pm 0.003$ | $20.75 \pm 0.51$ | $0.975 \pm 0.003$ | $30.58 \pm 0.68$ | $0.990 \pm 0.001$ | $45.93 \pm 1.02$ |
| LAC | w/ Energy | $0.900 \pm 0.007$ | $\mathbf{11.52} \pm 0.32$ | $0.950 \pm 0.003$ | $\mathbf{20.32} \pm 0.39$ | $0.976 \pm 0.003$ | $\mathbf{30.51} \pm 0.64$ | $0.990 \pm 0.001$ | $\mathbf{44.49} \pm 0.79$ |
| | | **CIFAR-100-LT ($\lambda = 0.02$) (ResNet-56)** | | | | | | | |
| LAC | w/o Energy | $0.901 \pm 0.007$ | $27.78 \pm 0.73$ | $0.950 \pm 0.007$ | $42.03 \pm 1.67$ | $0.975 \pm 0.003$ | $54.88 \pm 1.13$ | $0.990 \pm 0.002$ | $68.21 \pm 1.00$ |
| LAC | w/ Energy | $0.900 \pm 0.007$ | $\mathbf{27.63} \pm 0.65$ | $0.951 \pm 0.006$ | $\mathbf{41.49} \pm 1.19$ | $0.976 \pm 0.003$ | $\mathbf{54.13} \pm 0.93$ | $0.990 \pm 0.001$ | $\mathbf{67.28} \pm 1.47$ |
| | | **CIFAR-100-LT ($\lambda = 0.03$, severe imbalance) (ResNet-56)** | | | | | | | |
| LAC | w/o Energy | $0.901 \pm 0.006$ | $28.34 \pm 0.47$ | $0.951 \pm 0.004$ | $41.79 \pm 0.73$ | $0.976 \pm 0.002$ | $54.71 \pm 0.82$ | $0.990 \pm 0.002$ | $68.73 \pm 1.14$ |
| LAC | w/ Energy | $0.901 \pm 0.006$ | $\mathbf{27.93} \pm 0.42$ | $0.952 \pm 0.004$ | $\mathbf{41.50} \pm 0.71$ | $0.975 \pm 0.003$ | $\mathbf{54.02} \pm 1.07$ | $0.990 \pm 0.003$ | $\mathbf{68.20} \pm 1.45$ |

## J.3 ENERGY-BASED LAC PERFORMANCE ANALYSIS STRATIFIED BY SAMPLE DIFFICULTY

Table 9: Coverage and average set size for the LAC method on ImageNet, stratified by sample difficulty. Results are shown for $\alpha = 0.01$ and $\alpha = 0.025$. The table compares the baseline LAC with its energy-based variant.

| | | **LAC** | | | | | | | |
|---|---|---|---|---|---|---|---|---|---|
| | | $\alpha = 0.01$ | | | | $\alpha = 0.025$ | | | |
| | | w/o Energy | | w/ Energy | | w/o Energy | | w/ Energy | |
| Difficulty Level | Count | Cov. | Set Size | Cov. | Set Size | Cov. | Set Size | Cov. | Set Size |
| 1 to 1 | 15990 | 1.00 | 10.77 | 1.00 | 9.24 | 1.00 | 4.34 | 1.00 | 4.25 |
| 2 to 3 | 2547 | 1.00 | 21.47 | 1.00 | 20.61 | 0.99 | 8.86 | 0.99 | 8.78 |
| 4 to 6 | 638 | 0.99 | 32.07 | 0.98 | 32.96 | 0.93 | 13.29 | 0.93 | 13.36 |
| 7 to 10 | 306 | 0.96 | 38.18 | 0.96 | 40.00 | 0.84 | 15.86 | 0.85 | 16.14 |
| 11 to 25 | 275 | 0.92 | 44.22 | 0.91 | 47.14 | 0.60 | 17.07 | 0.56 | 17.05 |
| 26 to 50 | 104 | 0.62 | 47.37 | 0.61 | 50.88 | 0.04 | 17.60 | 0.06 | 17.59 |
| 51 to 100 | 74 | 0.32 | 56.19 | 0.39 | 64.61 | 0.00 | 19.99 | 0.00 | 21.07 |

# K  ADDITIONAL EXPERIMENTAL RESULTS FOR IMBALANCED DATA

To evaluate performance under class imbalance as described in Section 3.2, we construct several long-tailed variants of the CIFAR-100 dataset. In these variants, the number of training samples per class follows an exponential decay controlled by an imbalance factor $\lambda$. Figure 7 illustrates how different values of $\lambda$ create varying levels of imbalance in the training distribution.

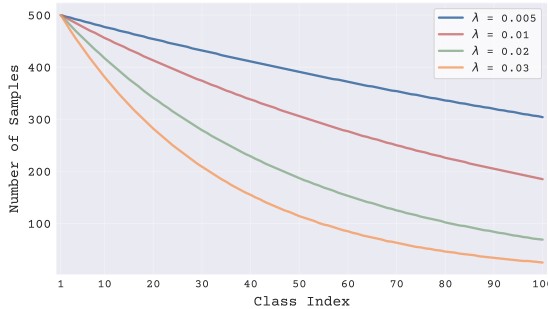

Figure 7: Class distributions under varying imbalance levels. The number of samples per class follows an exponential decay pattern proportional to $\exp(-\lambda \cdot j)$, where larger $\lambda$ values induce stronger imbalance.

## K.1  RESULTS FOR DIFFERENT IMBALANCE FACTOR $\lambda$

Table 10: Performance comparison of APS, RAPS, SAPS, and their Energy-based variants on imbalanced CIFAR-100 with varying imbalance factors ($\lambda \in \{0.01, 0.02, 0.03\}$) and at miscoverage levels $\alpha \in \{0.01, 0.025, 0.05, 0.1\}$. Results are averaged over 10 trials with a ResNet-56 model. For the **Set Size** column, lower is better. **Bold** values indicate the best performance within each method family.

| Method | | $\alpha = 0.1$ Coverage | $\alpha = 0.1$ Set Size | $\alpha = 0.05$ Coverage | $\alpha = 0.05$ Set Size | $\alpha = 0.025$ Coverage | $\alpha = 0.025$ Set Size | $\alpha = 0.01$ Coverage | $\alpha = 0.01$ Set Size |
|---|---|---|---|---|---|---|---|---|---|
| colspan CIFAR-100-LT ($\lambda = 0.01$, mild imbalance) (ResNet-56) | | | | | | | | | |
| APS | w/o Energy | $0.90 \pm 0.01$ | $14.56 \pm 0.46$ | $0.95 \pm 0.00$ | $25.59 \pm 0.25$ | $0.975 \pm 0.003$ | $37.56 \pm 0.98$ | $0.99 \pm 0.00$ | $55.26 \pm 2.49$ |
| APS | w/ Energy | $0.90 \pm 0.01$ | $\mathbf{11.86} \pm 0.37$ | $0.95 \pm 0.00$ | $\mathbf{21.44} \pm 0.52$ | $0.975 \pm 0.003$ | $\mathbf{31.70} \pm 0.97$ | $0.99 \pm 0.00$ | $\mathbf{46.40} \pm 0.72$ |
| RAPS | w/o Energy | $0.90 \pm 0.01$ | $15.48 \pm 0.47$ | $0.95 \pm 0.00$ | $27.62 \pm 0.33$ | $0.974 \pm 0.003$ | $40.87 \pm 1.00$ | $0.99 \pm 0.00$ | $60.13 \pm 1.86$ |
| RAPS | w/ Energy | $0.90 \pm 0.01$ | $\mathbf{11.77} \pm 0.42$ | $0.95 \pm 0.00$ | $\mathbf{21.57} \pm 0.56$ | $0.975 \pm 0.003$ | $\mathbf{31.93} \pm 0.98$ | $0.99 \pm 0.00$ | $\mathbf{47.30} \pm 1.01$ |
| SAPS | w/o Energy | $0.90 \pm 0.01$ | $15.02 \pm 0.50$ | $0.95 \pm 0.00$ | $26.98 \pm 0.32$ | $0.974 \pm 0.002$ | $40.38 \pm 1.26$ | $0.99 \pm 0.00$ | $59.12 \pm 1.76$ |
| SAPS | w/ Energy | $0.90 \pm 0.01$ | $\mathbf{11.80} \pm 0.40$ | $0.95 \pm 0.00$ | $\mathbf{21.35} \pm 0.55$ | $0.975 \pm 0.003$ | $\mathbf{31.84} \pm 1.02$ | $0.99 \pm 0.00$ | $\mathbf{47.17} \pm 1.05$ |
| colspan CIFAR-100-LT ($\lambda = 0.02$) (ResNet-56) | | | | | | | | | |
| APS | w/o Energy | $0.90 \pm 0.01$ | $29.62 \pm 0.56$ | $0.95 \pm 0.01$ | $45.76 \pm 1.32$ | $0.975 \pm 0.003$ | $59.26 \pm 0.82$ | $0.99 \pm 0.00$ | $73.49 \pm 1.36$ |
| APS | w/ Energy | $0.90 \pm 0.01$ | $\mathbf{28.23} \pm 0.74$ | $0.95 \pm 0.01$ | $\mathbf{42.36} \pm 1.13$ | $0.976 \pm 0.003$ | $\mathbf{54.95} \pm 0.91$ | $0.99 \pm 0.00$ | $\mathbf{69.58} \pm 1.92$ |
| RAPS | w/o Energy | $0.90 \pm 0.01$ | $32.86 \pm 0.81$ | $0.95 \pm 0.01$ | $52.01 \pm 1.55$ | $0.976 \pm 0.003$ | $66.83 \pm 1.18$ | $0.99 \pm 0.00$ | $79.97 \pm 1.36$ |
| RAPS | w/ Energy | $0.90 \pm 0.01$ | $\mathbf{28.72} \pm 0.77$ | $0.95 \pm 0.01$ | $\mathbf{42.72} \pm 1.02$ | $0.976 \pm 0.003$ | $\mathbf{56.31} \pm 0.86$ | $0.99 \pm 0.00$ | $\mathbf{70.72} \pm 1.56$ |
| SAPS | w/o Energy | $0.90 \pm 0.01$ | $32.37 \pm 0.83$ | $0.95 \pm 0.01$ | $51.47 \pm 1.48$ | $0.976 \pm 0.003$ | $66.28 \pm 1.21$ | $0.99 \pm 0.00$ | $79.14 \pm 1.54$ |
| SAPS | w/ Energy | $0.90 \pm 0.01$ | $\mathbf{28.73} \pm 0.78$ | $0.95 \pm 0.01$ | $\mathbf{42.63} \pm 1.09$ | $0.976 \pm 0.003$ | $\mathbf{56.16} \pm 0.89$ | $0.99 \pm 0.00$ | $\mathbf{70.48} \pm 1.56$ |
| colspan CIFAR-100-LT ($\lambda = 0.03$, severe imbalance) (ResNet-56) | | | | | | | | | |
| APS | w/o Energy | $0.90 \pm 0.01$ | $30.35 \pm 0.57$ | $0.95 \pm 0.00$ | $44.45 \pm 0.79$ | $0.975 \pm 0.002$ | $58.05 \pm 0.95$ | $0.99 \pm 0.00$ | $71.34 \pm 2.09$ |
| APS | w/ Energy | $0.90 \pm 0.00$ | $\mathbf{28.42} \pm 0.52$ | $0.95 \pm 0.00$ | $\mathbf{42.40} \pm 0.63$ | $0.975 \pm 0.003$ | $\mathbf{55.61} \pm 1.54$ | $0.99 \pm 0.00$ | $\mathbf{70.85} \pm 1.58$ |
| RAPS | w/o Energy | $0.90 \pm 0.01$ | $34.47 \pm 0.72$ | $0.95 \pm 0.01$ | $49.94 \pm 1.14$ | $0.975 \pm 0.003$ | $64.66 \pm 1.13$ | $0.99 \pm 0.00$ | $78.91 \pm 1.06$ |
| RAPS | w/ Energy | $0.90 \pm 0.01$ | $\mathbf{29.29} \pm 0.66$ | $0.95 \pm 0.00$ | $\mathbf{43.68} \pm 0.78$ | $0.975 \pm 0.003$ | $\mathbf{57.23} \pm 1.02$ | $0.99 \pm 0.00$ | $\mathbf{72.74} \pm 1.54$ |
| SAPS | w/o Energy | $0.90 \pm 0.01$ | $34.05 \pm 0.71$ | $0.95 \pm 0.01$ | $49.45 \pm 1.33$ | $0.974 \pm 0.003$ | $64.35 \pm 1.33$ | $0.99 \pm 0.00$ | $78.41 \pm 1.06$ |
| SAPS | w/ Energy | $0.90 \pm 0.01$ | $\mathbf{29.19} \pm 0.61$ | $0.95 \pm 0.01$ | $\mathbf{43.61} \pm 0.85$ | $0.975 \pm 0.003$ | $\mathbf{57.12} \pm 0.97$ | $0.99 \pm 0.00$ | $\mathbf{72.48} \pm 1.49$ |

### K.2 Performance Under Imbalanced Calibration and Test Sets

The results reported in Table 2 and Table 10 evaluate models trained on imbalanced data but calibrated and tested on balanced sets. We now consider a more realistic scenario where the calibration and test sets also follow the same imbalanced distribution as the training set.

This setting is particularly challenging for standard CP methods, as the limited number of calibration samples for minority classes can impede reliable coverage guarantees. In such cases, approaches like clustered conformal prediction Ding et al. (2023b), which can provide coverage with fewer calibration samples, are practical alternatives.

Our energy-based method is designed to address this challenge by adaptively enlarging prediction sets for uncertain inputs, which often correspond to minority class samples. To benchmark this behavior, we compare it against another principled reweighting strategy that directly uses class priors. This approach was introduced by Ding et al. (2025), who proposed the **Prevalence-Adjusted Softmax (PAS)** score. The PAS score modifies the non-adaptive score by dividing the negative softmax probability (LAC score without bias term) by the empirical class prior, $\hat{p}(y)$, to improve coverage for rare classes. The nonconformity score is defined as:

$$S_{\text{PAS}}(x, y) = \frac{-\hat{\pi}(y|x)}{\hat{p}(y)} \tag{56}$$

We extend this concept to adaptive nonconformity scores such as APS and RAPS. For these scores, a smaller value indicates higher conformity. To increase the likelihood of including labels from rare classes (which have a small $\hat{p}(y)$), we multiply the base score by the class prior. This makes the minority classes more likely to be included in the final prediction set. We refer to this method as **Prevalence-Adjusted (PA) Nonconformity Scores**. The general formulation is:

$$S_{\text{PA}}(x, y) = S_{\text{adaptive}}(x, y) \cdot \hat{p}(y) \tag{57}$$

where $S_{\text{adaptive}}(x, y)$ is an adaptive score like $S_{\text{APS}}(x, y)$.

Given this, Table 11 and Table 12 present a comparison between the standard adaptive scores, our energy-based variants, and the prevalence-adjusted variants in this fully imbalanced setting. We report marginal coverage, average set size, and MacroCov to provide a comprehensive view of performance.

Table 11: Performance on fully imbalanced CIFAR-100-LT for high confidence levels ($\alpha \in \{0.025, 0.01\}$). For each method, we compare the **Standard** baseline, the **Prevalence-Adj.** variant, and our **Energy-based** variant. Lower **Set Size** is better.

| Method | Variant | $\alpha = 0.025$ | | | $\alpha = 0.01$ | | |
|---|---|---|---|---|---|---|---|
| | | Cov | Size | MacroCov | Cov | Size | MacroCov |
| | **CIFAR-100-LT ($\lambda = 0.005$, mild imbalance)** | | | | | | |
| LAC | Standard | $0.97 \pm 0.00$ | $21.48 \pm 0.86$ | $0.97 \pm 0.00$ | $0.99 \pm 0.00$ | $34.60 \pm 1.77$ | $0.99 \pm 0.00$ |
| | Prevalence-Adj. (PAS) | $0.98 \pm 0.00$ | $22.84 \pm 0.99$ | $0.98 \pm 0.00$ | $0.99 \pm 0.00$ | $36.23 \pm 0.98$ | $0.99 \pm 0.00$ |
| | Energy-based | $0.97 \pm 0.00$ | $\mathbf{21.11} \pm 0.79$ | $0.97 \pm 0.00$ | $0.99 \pm 0.00$ | $\mathbf{33.18} \pm 1.44$ | $0.99 \pm 0.00$ |
| APS | Standard | $0.97 \pm 0.00$ | $28.61 \pm 1.23$ | $0.97 \pm 0.00$ | $0.99 \pm 0.00$ | $46.37 \pm 1.39$ | $0.99 \pm 0.00$ |
| | Prevalence-Adj. | $0.98 \pm 0.00$ | $29.72 \pm 1.80$ | $0.97 \pm 0.00$ | $0.99 \pm 0.00$ | $50.57 \pm 2.08$ | $0.99 \pm 0.00$ |
| | Energy-based | $0.97 \pm 0.00$ | $\mathbf{22.09} \pm 1.19$ | $0.97 \pm 0.00$ | $0.99 \pm 0.00$ | $\mathbf{35.78} \pm 1.43$ | $0.99 \pm 0.00$ |
| RAPS | Standard | $0.97 \pm 0.00$ | $30.52 \pm 1.44$ | $0.97 \pm 0.00$ | $0.99 \pm 0.00$ | $51.46 \pm 1.84$ | $0.99 \pm 0.00$ |
| | Prevalence-Adj. | $0.97 \pm 0.00$ | $31.45 \pm 1.66$ | $0.97 \pm 0.00$ | $0.99 \pm 0.00$ | $51.41 \pm 1.66$ | $0.99 \pm 0.00$ |
| | Energy-based | $0.97 \pm 0.00$ | $\mathbf{22.57} \pm 1.08$ | $0.97 \pm 0.00$ | $0.99 \pm 0.00$ | $\mathbf{36.75} \pm 1.42$ | $0.99 \pm 0.00$ |
| SAPS | Standard | $0.97 \pm 0.00$ | $29.54 \pm 1.48$ | $0.97 \pm 0.00$ | $0.99 \pm 0.00$ | $50.35 \pm 2.04$ | $0.99 \pm 0.00$ |
| | Prevalence-Adj. | $0.97 \pm 0.00$ | $30.41 \pm 1.62$ | $0.97 \pm 0.00$ | $0.99 \pm 0.00$ | $50.50 \pm 1.33$ | $0.99 \pm 0.00$ |
| | Energy-based | $0.97 \pm 0.00$ | $\mathbf{22.36} \pm 1.13$ | $0.97 \pm 0.00$ | $0.99 \pm 0.00$ | $\mathbf{36.39} \pm 1.23$ | $0.99 \pm 0.00$ |
| | **CIFAR-100-LT ($\lambda = 0.01$)** | | | | | | |
| LAC | Standard | $0.97 \pm 0.00$ | $30.22 \pm 0.83$ | $0.97 \pm 0.00$ | $0.99 \pm 0.00$ | $45.45 \pm 1.77$ | $0.99 \pm 0.00$ |
| | Prevalence-Adj. (PAS) | $0.98 \pm 0.00$ | $33.95 \pm 1.00$ | $0.98 \pm 0.00$ | $0.99 \pm 0.00$ | $50.56 \pm 1.59$ | $0.99 \pm 0.00$ |
| | Energy-based | $0.97 \pm 0.00$ | $\mathbf{29.93} \pm 0.99$ | $0.97 \pm 0.00$ | $0.99 \pm 0.00$ | $\mathbf{43.94} \pm 1.34$ | $0.99 \pm 0.00$ |
| APS | Standard | $0.97 \pm 0.00$ | $36.97 \pm 1.61$ | $0.97 \pm 0.00$ | $0.99 \pm 0.00$ | $54.90 \pm 2.32$ | $0.99 \pm 0.00$ |
| | Prevalence-Adj. | $0.98 \pm 0.00$ | $43.29 \pm 2.15$ | $0.98 \pm 0.00$ | $0.99 \pm 0.00$ | $66.30 \pm 2.14$ | $0.99 \pm 0.00$ |
| | Energy-based | $0.97 \pm 0.00$ | $\mathbf{31.20} \pm 1.12$ | $0.97 \pm 0.00$ | $0.99 \pm 0.00$ | $\mathbf{45.15} \pm 1.81$ | $0.99 \pm 0.00$ |
| RAPS | Standard | $0.97 \pm 0.00$ | $40.39 \pm 1.29$ | $0.97 \pm 0.00$ | $0.99 \pm 0.00$ | $60.09 \pm 2.93$ | $0.99 \pm 0.00$ |
| | Prevalence-Adj. | $0.98 \pm 0.00$ | $43.21 \pm 2.15$ | $0.98 \pm 0.00$ | $0.99 \pm 0.00$ | $65.82 \pm 2.35$ | $0.99 \pm 0.00$ |
| | Energy-based | $0.97 \pm 0.00$ | $\mathbf{31.64} \pm 1.24$ | $0.97 \pm 0.00$ | $0.99 \pm 0.00$ | $\mathbf{45.99} \pm 1.67$ | $0.99 \pm 0.00$ |
| SAPS | Standard | $0.97 \pm 0.00$ | $39.81 \pm 1.53$ | $0.97 \pm 0.00$ | $0.99 \pm 0.00$ | $59.11 \pm 2.61$ | $0.99 \pm 0.00$ |
| | Prevalence-Adj. | $0.98 \pm 0.00$ | $42.34 \pm 2.12$ | $0.98 \pm 0.00$ | $0.99 \pm 0.00$ | $65.18 \pm 2.45$ | $0.99 \pm 0.00$ |
| | Energy-based | $0.97 \pm 0.01$ | $\mathbf{31.46} \pm 1.26$ | $0.97 \pm 0.01$ | $0.99 \pm 0.00$ | $\mathbf{45.73} \pm 1.71$ | $0.99 \pm 0.00$ |
| | **CIFAR-100-LT ($\lambda = 0.02$)** | | | | | | |
| LAC | Standard | $0.97 \pm 0.00$ | $52.86 \pm 1.54$ | $0.97 \pm 0.00$ | $0.99 \pm 0.00$ | $66.53 \pm 1.36$ | $0.99 \pm 0.00$ |
| | Prevalence-Adj. (PAS) | $0.98 \pm 0.00$ | $61.74 \pm 1.37$ | $0.98 \pm 0.00$ | $0.99 \pm 0.00$ | $75.07 \pm 1.20$ | $0.99 \pm 0.00$ |
| | Energy-based | $0.97 \pm 0.00$ | $\mathbf{51.96} \pm 1.30$ | $0.97 \pm 0.00$ | $0.99 \pm 0.00$ | $\mathbf{65.48} \pm 2.27$ | $0.99 \pm 0.00$ |
| APS | Standard | $0.96 \pm 0.00$ | $56.87 \pm 1.30$ | $0.97 \pm 0.00$ | $0.99 \pm 0.00$ | $71.19 \pm 0.92$ | $0.99 \pm 0.00$ |
| | Prevalence-Adj. | $0.99 \pm 0.00$ | $74.42 \pm 0.39$ | $0.98 \pm 0.00$ | $1.00 \pm 0.00$ | $86.18 \pm 0.30$ | $0.99 \pm 0.00$ |
| | Energy-based | $0.97 \pm 0.00$ | $\mathbf{53.13} \pm 1.53$ | $0.97 \pm 0.00$ | $0.99 \pm 0.00$ | $\mathbf{68.02} \pm 1.60$ | $0.99 \pm 0.00$ |
| RAPS | Standard | $0.96 \pm 0.00$ | $61.92 \pm 1.24$ | $0.97 \pm 0.00$ | $0.99 \pm 0.00$ | $76.89 \pm 1.40$ | $0.99 \pm 0.00$ |
| | Prevalence-Adj. | $0.99 \pm 0.00$ | $73.38 \pm 0.28$ | $0.98 \pm 0.00$ | $1.00 \pm 0.00$ | $86.52 \pm 0.40$ | $0.99 \pm 0.00$ |
| | Energy-based | $0.97 \pm 0.00$ | $\mathbf{54.13} \pm 1.37$ | $0.97 \pm 0.00$ | $0.99 \pm 0.00$ | $\mathbf{69.14} \pm 1.92$ | $0.99 \pm 0.00$ |
| SAPS | Standard | $0.96 \pm 0.00$ | $61.42 \pm 1.18$ | $0.97 \pm 0.00$ | $0.99 \pm 0.00$ | $76.01 \pm 1.11$ | $0.99 \pm 0.00$ |
| | Prevalence-Adj. | $0.99 \pm 0.00$ | $73.12 \pm 0.36$ | $0.98 \pm 0.00$ | $1.00 \pm 0.00$ | $86.15 \pm 0.29$ | $0.99 \pm 0.00$ |
| | Energy-based | $0.97 \pm 0.00$ | $\mathbf{54.02} \pm 1.58$ | $0.97 \pm 0.00$ | $0.99 \pm 0.00$ | $\mathbf{69.01} \pm 1.67$ | $0.99 \pm 0.00$ |
| | **CIFAR-100-LT ($\lambda = 0.03$, severe imbalance)** | | | | | | |
| LAC | Standard | $0.96 \pm 0.00$ | $51.39 \pm 1.25$ | $0.97 \pm 0.00$ | $0.98 \pm 0.00$ | $67.76 \pm 0.84$ | $0.99 \pm 0.00$ |
| | Prevalence-Adj. (PAS) | $0.98 \pm 0.00$ | $62.93 \pm 0.99$ | $0.98 \pm 0.00$ | $1.00 \pm 0.00$ | $75.03 \pm 0.46$ | $0.99 \pm 0.00$ |
| | Energy-based | $0.96 \pm 0.00$ | $\mathbf{50.77} \pm 1.35$ | $0.97 \pm 0.00$ | $0.98 \pm 0.00$ | $\mathbf{66.87} \pm 0.95$ | $0.99 \pm 0.00$ |
| APS | Standard | $0.96 \pm 0.00$ | $53.95 \pm 1.21$ | $0.97 \pm 0.00$ | $0.99 \pm 0.00$ | $69.98 \pm 1.29$ | $0.99 \pm 0.00$ |
| | Prevalence-Adj. | $0.99 \pm 0.00$ | $79.13 \pm 0.69$ | $0.98 \pm 0.00$ | $1.00 \pm 0.00$ | $86.14 \pm 0.28$ | $0.99 \pm 0.00$ |
| | Energy-based | $0.96 \pm 0.01$ | $\mathbf{52.00} \pm 1.48$ | $0.97 \pm 0.00$ | $0.99 \pm 0.00$ | $\mathbf{69.77} \pm 1.34$ | $0.99 \pm 0.00$ |
| RAPS | Standard | $0.96 \pm 0.00$ | $60.09 \pm 1.28$ | $0.97 \pm 0.00$ | $0.98 \pm 0.00$ | $75.77 \pm 1.34$ | $0.99 \pm 0.00$ |
| | Prevalence-Adj. | $0.99 \pm 0.00$ | $78.45 \pm 0.47$ | $0.98 \pm 0.00$ | $1.00 \pm 0.00$ | $86.09 \pm 0.42$ | $0.99 \pm 0.00$ |
| | Energy-based | $0.96 \pm 0.01$ | $\mathbf{53.67} \pm 1.54$ | $0.97 \pm 0.00$ | $0.99 \pm 0.00$ | $\mathbf{71.64} \pm 1.17$ | $0.99 \pm 0.00$ |
| SAPS | Standard | $0.96 \pm 0.00$ | $59.61 \pm 1.28$ | $0.97 \pm 0.00$ | $0.98 \pm 0.00$ | $75.59 \pm 1.15$ | $0.99 \pm 0.00$ |
| | Prevalence-Adj. | $0.99 \pm 0.00$ | $78.38 \pm 0.51$ | $0.98 \pm 0.00$ | $1.00 \pm 0.00$ | $86.02 \pm 0.33$ | $0.99 \pm 0.00$ |
| | Energy-based | $0.96 \pm 0.00$ | $\mathbf{53.68} \pm 1.51$ | $0.97 \pm 0.00$ | $0.98 \pm 0.00$ | $\mathbf{71.72} \pm 1.30$ | $0.99 \pm 0.00$ |

Table 12: Performance on fully imbalanced CIFAR-100-LT for lower confidence levels ($\alpha \in \{0.1, 0.05\}$). For each method, we compare the **Standard** baseline, the **Prevalence-Adj.** variant, and our **Energy-based** variant. Lower **Set Size** is better.

| Method | Variant | $\alpha = 0.1$ | | | $\alpha = 0.05$ | | |
| --- | --- | --- | --- | --- | --- | --- | --- |
| | | **Cov** | **Size** | **MacroCov** | **Cov** | **Size** | **MacroCov** |
| **CIFAR-100-LT ($\lambda = 0.005$, mild imbalance)** | | | | | | | |
| LAC | Standard | $0.90 \pm 0.01$ | $7.12 \pm 0.26$ | $0.90 \pm 0.01$ | $0.95 \pm 0.01$ | $13.08 \pm 0.55$ | $0.95 \pm 0.01$ |
| | Prevalence-Adj. (PAS) | $0.90 \pm 0.01$ | $7.26 \pm 0.30$ | $0.90 \pm 0.01$ | $0.95 \pm 0.01$ | $13.39 \pm 0.60$ | $0.95 \pm 0.01$ |
| | Energy-based | $0.90 \pm 0.01$ | $\mathbf{6.96} \pm 0.23$ | $0.90 \pm 0.01$ | $0.95 \pm 0.01$ | $\mathbf{12.74} \pm 0.54$ | $0.95 \pm 0.01$ |
| APS | Standard | $0.90 \pm 0.01$ | $8.48 \pm 0.42$ | $0.90 \pm 0.01$ | $0.95 \pm 0.01$ | $17.55 \pm 1.03$ | $0.95 \pm 0.01$ |
| | Prevalence-Adj. | $0.90 \pm 0.01$ | $8.81 \pm 0.35$ | $0.90 \pm 0.01$ | $0.95 \pm 0.01$ | $17.95 \pm 0.82$ | $0.95 \pm 0.01$ |
| | Energy-based | $0.90 \pm 0.01$ | $\mathbf{7.39} \pm 0.22$ | $0.90 \pm 0.01$ | $0.95 \pm 0.01$ | $\mathbf{13.42} \pm 0.56$ | $0.95 \pm 0.01$ |
| RAPS | Standard | $0.90 \pm 0.01$ | $8.95 \pm 0.44$ | $0.90 \pm 0.01$ | $0.95 \pm 0.01$ | $18.89 \pm 1.06$ | $0.95 \pm 0.01$ |
| | Prevalence-Adj. | $0.90 \pm 0.01$ | $9.14 \pm 0.46$ | $0.90 \pm 0.01$ | $0.95 \pm 0.01$ | $19.19 \pm 0.89$ | $0.95 \pm 0.01$ |
| | Energy-based | $0.90 \pm 0.01$ | $\mathbf{7.58} \pm 0.25$ | $0.90 \pm 0.01$ | $0.95 \pm 0.01$ | $\mathbf{13.33} \pm 0.56$ | $0.95 \pm 0.01$ |
| SAPS | Standard | $0.90 \pm 0.01$ | $8.63 \pm 0.47$ | $0.90 \pm 0.01$ | $0.95 \pm 0.01$ | $18.22 \pm 1.09$ | $0.95 \pm 0.01$ |
| | Prevalence-Adj. | $0.90 \pm 0.01$ | $8.83 \pm 0.46$ | $0.90 \pm 0.01$ | $0.95 \pm 0.01$ | $18.50 \pm 0.91$ | $0.95 \pm 0.01$ |
| | Energy-based | $0.90 \pm 0.01$ | $\mathbf{7.53} \pm 0.28$ | $0.90 \pm 0.01$ | $0.95 \pm 0.00$ | $\mathbf{13.30} \pm 0.52$ | $0.95 \pm 0.00$ |
| **CIFAR-100-LT ($\lambda = 0.01$)** | | | | | | | |
| LAC | Standard | $0.89 \pm 0.01$ | $11.37 \pm 0.28$ | $0.89 \pm 0.01$ | $0.95 \pm 0.01$ | $20.09 \pm 0.71$ | $0.95 \pm 0.01$ |
| | Prevalence-Adj. (PAS) | $0.91 \pm 0.01$ | $12.20 \pm 0.24$ | $0.90 \pm 0.01$ | $0.96 \pm 0.00$ | $22.03 \pm 0.63$ | $0.95 \pm 0.00$ |
| | Energy-based | $0.89 \pm 0.01$ | $\mathbf{11.12} \pm 0.29$ | $0.89 \pm 0.01$ | $0.95 \pm 0.01$ | $\mathbf{19.73} \pm 0.70$ | $0.95 \pm 0.01$ |
| APS | Standard | $0.89 \pm 0.00$ | $13.76 \pm 0.30$ | $0.90 \pm 0.00$ | $0.95 \pm 0.01$ | $25.34 \pm 0.95$ | $0.95 \pm 0.01$ |
| | Prevalence-Adj. | $0.91 \pm 0.01$ | $14.68 \pm 0.36$ | $0.90 \pm 0.01$ | $0.96 \pm 0.01$ | $26.62 \pm 0.96$ | $0.95 \pm 0.01$ |
| | Energy-based | $0.89 \pm 0.00$ | $\mathbf{11.29} \pm 0.22$ | $0.89 \pm 0.00$ | $0.95 \pm 0.01$ | $\mathbf{20.80} \pm 0.70$ | $0.95 \pm 0.01$ |
| RAPS | Standard | $0.89 \pm 0.00$ | $14.72 \pm 0.19$ | $0.90 \pm 0.00$ | $0.95 \pm 0.01$ | $26.91 \pm 1.03$ | $0.95 \pm 0.01$ |
| | Prevalence-Adj. | $0.90 \pm 0.01$ | $15.03 \pm 0.54$ | $0.90 \pm 0.01$ | $0.95 \pm 0.00$ | $27.14 \pm 0.84$ | $0.95 \pm 0.00$ |
| | Energy-based | $0.89 \pm 0.00$ | $\mathbf{11.22} \pm 0.18$ | $0.89 \pm 0.00$ | $0.95 \pm 0.01$ | $\mathbf{20.95} \pm 0.77$ | $0.95 \pm 0.01$ |
| SAPS | Standard | $0.89 \pm 0.00$ | $14.22 \pm 0.19$ | $0.90 \pm 0.00$ | $0.95 \pm 0.01$ | $26.19 \pm 0.98$ | $0.95 \pm 0.01$ |
| | Prevalence-Adj. | $0.90 \pm 0.01$ | $14.55 \pm 0.43$ | $0.90 \pm 0.01$ | $0.95 \pm 0.00$ | $26.65 \pm 0.84$ | $0.95 \pm 0.00$ |
| | Energy-based | $0.89 \pm 0.00$ | $\mathbf{11.26} \pm 0.17$ | $0.89 \pm 0.00$ | $0.95 \pm 0.01$ | $\mathbf{20.73} \pm 0.80$ | $0.95 \pm 0.01$ |
| **CIFAR-100-LT ($\lambda = 0.02$)** | | | | | | | |
| LAC | Standard | $0.86 \pm 0.01$ | $24.97 \pm 0.67$ | $0.88 \pm 0.01$ | $0.93 \pm 0.01$ | $38.81 \pm 1.07$ | $0.94 \pm 0.01$ |
| | Prevalence-Adj. (PAS) | $0.91 \pm 0.00$ | $30.25 \pm 0.46$ | $0.91 \pm 0.00$ | $0.96 \pm 0.00$ | $46.98 \pm 1.15$ | $0.96 \pm 0.00$ |
| | Energy-based | $0.86 \pm 0.01$ | $\mathbf{24.65} \pm 0.67$ | $0.88 \pm 0.01$ | $0.93 \pm 0.01$ | $\mathbf{38.46} \pm 0.94$ | $0.94 \pm 0.01$ |
| APS | Standard | $0.87 \pm 0.01$ | $26.69 \pm 0.71$ | $0.88 \pm 0.01$ | $0.93 \pm 0.01$ | $42.41 \pm 1.32$ | $0.94 \pm 0.01$ |
| | Prevalence-Adj. | $0.92 \pm 0.00$ | $37.05 \pm 0.63$ | $0.91 \pm 0.00$ | $0.97 \pm 0.00$ | $60.22 \pm 0.92$ | $0.96 \pm 0.00$ |
| | Energy-based | $0.86 \pm 0.01$ | $\mathbf{25.07} \pm 0.76$ | $0.88 \pm 0.01$ | $0.93 \pm 0.01$ | $\mathbf{39.10} \pm 0.96$ | $0.94 \pm 0.01$ |
| RAPS | Standard | $0.86 \pm 0.01$ | $28.44 \pm 0.78$ | $0.88 \pm 0.01$ | $0.93 \pm 0.01$ | $46.22 \pm 1.45$ | $0.94 \pm 0.01$ |
| | Prevalence-Adj. | $0.92 \pm 0.00$ | $35.42 \pm 0.69$ | $0.91 \pm 0.00$ | $0.97 \pm 0.00$ | $59.12 \pm 0.77$ | $0.96 \pm 0.00$ |
| | Energy-based | $0.86 \pm 0.01$ | $\mathbf{25.37} \pm 0.70$ | $0.88 \pm 0.01$ | $0.93 \pm 0.01$ | $\mathbf{39.63} \pm 0.98$ | $0.94 \pm 0.01$ |
| SAPS | Standard | $0.86 \pm 0.01$ | $28.26 \pm 0.76$ | $0.88 \pm 0.01$ | $0.93 \pm 0.01$ | $45.73 \pm 1.28$ | $0.94 \pm 0.01$ |
| | Prevalence-Adj. | $0.92 \pm 0.00$ | $35.69 \pm 0.66$ | $0.91 \pm 0.00$ | $0.97 \pm 0.00$ | $59.11 \pm 0.95$ | $0.96 \pm 0.00$ |
| | Energy-based | $0.86 \pm 0.01$ | $\mathbf{25.41} \pm 0.71$ | $0.88 \pm 0.01$ | $0.93 \pm 0.01$ | $\mathbf{39.72} \pm 1.00$ | $0.94 \pm 0.01$ |
| **CIFAR-100-LT ($\lambda = 0.03$, severe imbalance)** | | | | | | | |
| LAC | Standard | $0.84 \pm 0.01$ | $24.43 \pm 0.63$ | $0.87 \pm 0.01$ | $0.92 \pm 0.01$ | $38.72 \pm 0.77$ | $0.94 \pm 0.01$ |
| | Prevalence-Adj. (PAS) | $0.90 \pm 0.01$ | $34.18 \pm 1.00$ | $0.90 \pm 0.01$ | $0.95 \pm 0.01$ | $49.60 \pm 1.43$ | $0.95 \pm 0.01$ |
| | Energy-based | $0.84 \pm 0.01$ | $\mathbf{24.06} \pm 0.64$ | $0.87 \pm 0.01$ | $0.92 \pm 0.01$ | $\mathbf{38.04} \pm 0.90$ | $0.94 \pm 0.01$ |
| APS | Standard | $0.85 \pm 0.01$ | $25.64 \pm 0.94$ | $0.88 \pm 0.01$ | $0.92 \pm 0.01$ | $39.43 \pm 1.23$ | $0.94 \pm 0.01$ |
| | Prevalence-Adj. | $0.95 \pm 0.00$ | $48.47 \pm 0.65$ | $0.93 \pm 0.00$ | $0.98 \pm 0.00$ | $66.59 \pm 0.73$ | $0.97 \pm 0.01$ |
| | Energy-based | $0.84 \pm 0.01$ | $\mathbf{24.43} \pm 0.88$ | $0.87 \pm 0.01$ | $0.92 \pm 0.01$ | $\mathbf{38.78} \pm 1.12$ | $0.94 \pm 0.01$ |
| RAPS | Standard | $0.85 \pm 0.01$ | $28.00 \pm 1.00$ | $0.87 \pm 0.01$ | $0.92 \pm 0.01$ | $43.81 \pm 1.37$ | $0.94 \pm 0.01$ |
| | Prevalence-Adj. | $0.95 \pm 0.01$ | $46.46 \pm 0.88$ | $0.93 \pm 0.01$ | $0.98 \pm 0.00$ | $65.14 \pm 0.70$ | $0.97 \pm 0.01$ |
| | Energy-based | $0.84 \pm 0.01$ | $\mathbf{24.74} \pm 0.69$ | $0.87 \pm 0.01$ | $0.92 \pm 0.01$ | $\mathbf{39.74} \pm 1.17$ | $0.94 \pm 0.01$ |
| SAPS | Standard | $0.85 \pm 0.01$ | $27.64 \pm 0.85$ | $0.88 \pm 0.01$ | $0.92 \pm 0.01$ | $43.29 \pm 1.47$ | $0.94 \pm 0.01$ |
| | Prevalence-Adj. | $0.95 \pm 0.00$ | $46.81 \pm 0.81$ | $0.93 \pm 0.01$ | $0.98 \pm 0.00$ | $65.25 \pm 0.69$ | $0.97 \pm 0.01$ |
| | Energy-based | $0.84 \pm 0.01$ | $\mathbf{24.76} \pm 0.63$ | $0.87 \pm 0.01$ | $0.92 \pm 0.01$ | $\mathbf{39.73} \pm 1.24$ | $0.94 \pm 0.01$ |

## L CLASS-CONDITIONAL CONFORMAL PREDICTION

Beyond marginal coverage guarantee, we evaluate the proposed Energy-based nonconformity scores within the class-conditional setting. Class-conditional Conformal Prediction (CP) operates by partitioning the calibration dataset according to the true labels. Nonconformity score quantiles are then computed independently for each class using its respective calibration subset. The objective is to achieve *class-conditional coverage*, defined as:

$$\mathbb{P}(Y_{\text{test}} \in \mathcal{C}(X_{\text{test}}) \mid Y_{\text{test}} = y) \geq 1 - \alpha, \quad \text{for all } y \in \mathcal{Y}, \tag{58}$$

This condition ensures that for every class $y \in \mathcal{Y}$, the probability of the true label being included in the prediction set $\mathcal{C}(\mathbf{X}_{\text{test}})$ is at least $1 - \alpha$.

We evaluate performance on the Places365 dataset at miscoverage levels $\alpha \in \{0.05, 0.1\}$ using average set size, CovGap, and SSCV (for details of these refer to E.3. As shown in Table 13, Energy-based variants consistently yield more efficient prediction sets, reflected in reduced average set sizes across all base nonconformity functions. Importantly, this improvement in efficiency does not compromise class-conditional validity and CovGap is preserved or slightly improved in most cases. While we report SSCV for completeness, given its use in prior work, and note that it is often maintained or even improved in our experiments, we emphasize that it is not a reliable measure of conditional coverage quality.

Table 13: Class-conditional performance comparison of different nonconformity score functions and their Energy-based variants on the Places365 dataset at miscoverage levels $\alpha = 0.05$ and $\alpha = 0.1$. For Set Size, CovGap, and SSCV, lower values indicate better performance. **Bold** values denote the best result within each method family. Results are averaged over 10 trials with a ResNet-50 model.

| Method | | $\alpha = 0.1$ | | | | $\alpha = 0.05$ | | | |
| --- | --- | --- | --- | --- | --- | --- | --- | --- | --- |
| | | Coverage | Set Size ↓ | CovGap ↓ | SSCV ↓ | Coverage | Set Size ↓ | CovGap ↓ | SSCV ↓ |
| APS | w/o Energy | $0.89 \pm 0.00$ | $9.13 \pm 0.14$ | **5.03** $\pm 0.14$ | $0.119 \pm 0.015$ | $0.95 \pm 0.00$ | $21.50 \pm 0.59$ | $3.25 \pm 0.12$ | $0.124 \pm 0.039$ |
| | w/ Energy | $0.89 \pm 0.00$ | **8.65** $\pm 0.17$ | $5.09 \pm 0.18$ | **0.100** $\pm 0.00$ | $0.95 \pm 0.00$ | **19.20** $\pm 0.49$ | **3.24** $\pm 0.10$ | **0.087** $\pm 0.027$ |
| RAPS | w/o Energy | $0.89 \pm 0.00$ | $9.11 \pm 0.18$ | $4.98 \pm 0.15$ | **0.100** $\pm 0.00$ | $0.95 \pm 0.00$ | $22.03 \pm 0.59$ | $3.24 \pm 0.13$ | **0.090** $\pm 0.024$ |
| | w/ Energy | $0.89 \pm 0.00$ | **8.59** $\pm 0.17$ | **4.95** $\pm 0.19$ | $0.112 \pm 0.011$ | $0.95 \pm 0.00$ | **19.11** $\pm 0.47$ | **3.22** $\pm 0.11$ | $0.119 \pm 0.020$ |
| SAPS | w/o Energy | $0.89 \pm 0.00$ | $8.92 \pm 0.20$ | $4.98 \pm 0.20$ | **0.100** $\pm 0.00$ | $0.95 \pm 0.00$ | $21.68 \pm 0.58$ | $3.26 \pm 0.14$ | **0.060** $\pm 0.019$ |
| | w/ Energy | $0.89 \pm 0.00$ | **8.62** $\pm 0.20$ | **4.87** $\pm 0.13$ | $0.103 \pm 0.01$ | $0.95 \pm 0.00$ | **18.99** $\pm 0.49$ | **3.17** $\pm 0.11$ | $0.083 \pm 0.013$ |

## M EFFECT OF RAPS HYPERPARAMETERS $\lambda$ AND $k_{\text{REG}}$

In this section, we evaluate the sensitivity of the Regularized Adaptive Prediction Sets (RAPS) method, and its energy-based variant, to their two core hyperparameters: the regularization weight $\lambda$ and the penalty threshold $k_{reg}$. Table 14 compares the average prediction set size of the standard RAPS baseline against our proposed Energy-based RAPS on the Places365 dataset using a ResNet-50 model, across a grid of parameter configurations.

Table 14: Comparison of average prediction set sizes for varying regularization parameters ($k_{reg}$ and $\lambda$). Lower set size is better. Values are reported as: Size (RAPS→ Energy-based RAPS). The **bold** value highlights the superior (smaller) set size.

| | Average Set Size (w/o Energy → w/ Energy) ↓ | | | | | | |
| --- | --- | --- | --- | --- | --- | --- | --- |
| | Regularization Penalty ($\lambda$) | | | | | | |
| $k_{reg}$ | 0 | 0.05 | 0.1 | 0.2 | 0.5 | 0.7 | 1.0 |
| 1 | $14.08 \to$ **12.88** | $13.11 \to$ **12.88** | $13.60 \to$ **12.24** | $14.07 \to$ **12.61** | $14.30 \to$ **12.90** | $14.30 \to$ **12.95** | $14.30 \to$ **13.00** |
| 2 | $14.09 \to$ **12.86** | $13.13 \to$ **12.83** | $13.58 \to$ **12.34** | $14.05 \to$ **12.63** | $14.30 \to$ **12.97** | $14.30 \to$ **13.05** | $14.30 \to$ **13.11** |
| 5 | $14.01 \to$ **12.85** | $13.11 \to$ **13.06** | $13.61 \to$ **12.67** | $14.07 \to$ **12.86** | $14.30 \to$ **13.22** | $14.30 \to$ **13.29** | $14.30 \to$ **13.34** |
| 10 | $14.05 \to$ **12.90** | $13.13 \to$ **13.07** | $13.58 \to$ **13.57** | $14.07 \to$ **13.46** | $14.29 \to$ **13.62** | $14.30 \to$ **13.74** | $14.30 \to$ **13.82** |
| 50 | $14.00 \to$ **12.86** | $13.97 \to$ **12.91** | $14.02 \to$ **12.85** | $14.00 \to$ **12.85** | $14.02 \to$ **12.88** | $14.05 \to$ **12.87** | $13.90 \to$ **12.88** |

# N    EFFECT OF $T_{\text{CALIBRATION}}$ AND $\tau_{\text{ENERGY}}$

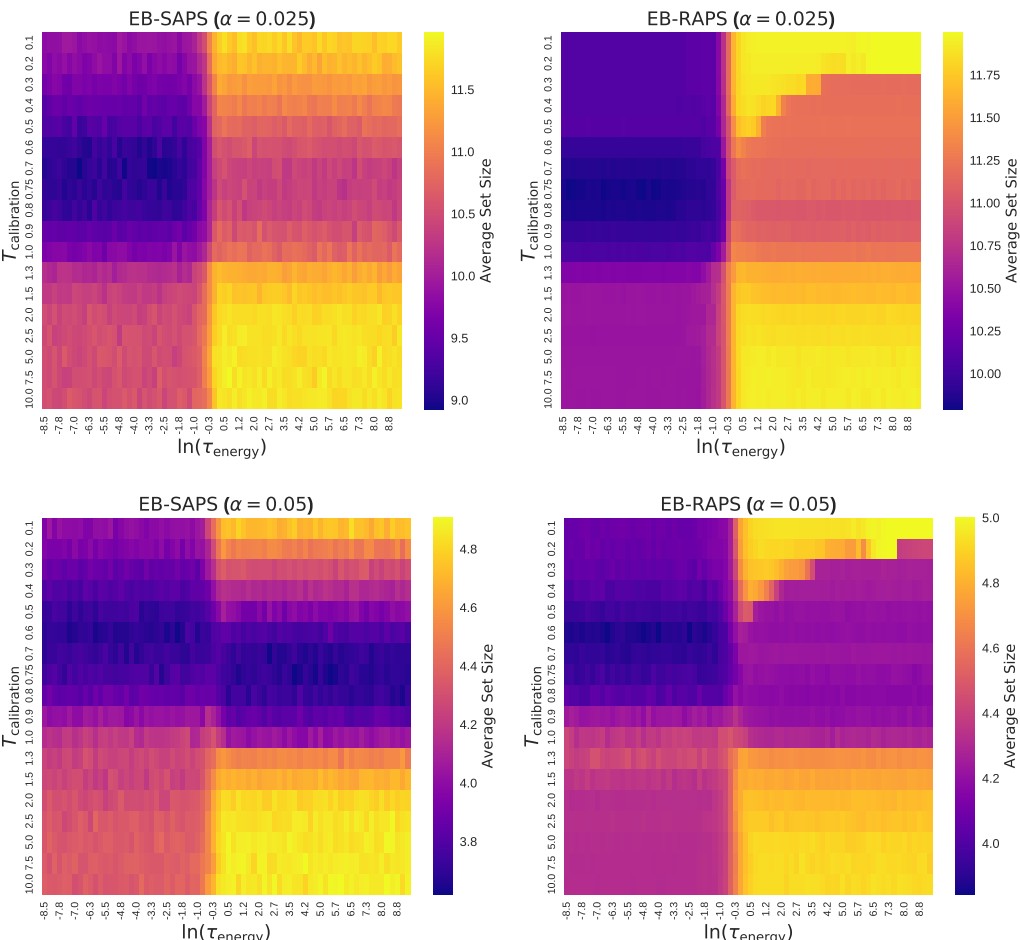

Figure 8: Average Set Size heatmap for different hyperparameter settings across Energy-based variants of `RAPS` and `SAPS`.

Sensitivity heatmaps illustrate how $T$ (temperature in the calibrated softmax) and $\tau$ (temperature in the energy calculation) affect the average set size on the ImageNet dataset for Energy-Based `RAPS` and Energy-Based `SAPS`, with $\alpha \in \{0.025, 0.05\}$. As $\ln(\tau)$ increases, the effect of energy-based reweighting gradually diminishes. Consequently, for larger values of $\tau$, the model converges to the baseline method. For instance, Energy-Based RAPS with a large positive $\ln(\tau)$ behaves almost identically to standard RAPS. As shown in Figure 8, across different values of $T$ (softmax probability calibration), energy-based variants of the methods (corresponding to smaller values of $\ln(\tau)$) produce more informative prediction sets compared to their baseline counterparts (associated with larger values of $\ln(\tau)$).

## O    SENSITIVITY ANALYSIS OF THE SOFTPLUS PARAMETER $\beta$

We analyze the impact of the sharpness parameter $\beta$ on the efficiency of the generated prediction sets. As defined in Equation 9, this parameter controls the approximation of the scaling factor to the ReLU function. We evaluate the average prediction set size across a wide range of $\beta$ values for the CIFAR-100 dataset using a ResNet-56 model.

Figure 9 presents the results for Energy-based APS, Energy-based RAPS, and Energy-based SAPS at miscoverage levels $\alpha = 0.05$ and $\alpha = 0.025$. As $\beta$ approaches zero, the term $\frac{1}{\beta} \log \left( 1 + e^{-\beta F(x)} \right)$ yields scaling factors that are inseparable across samples. Due to this loss of distinction, the performance converges to the baseline without energy.

However, as $\beta$ increases, the performance stabilizes and remains constant across several orders of magnitude. This behavior aligns with the theoretical motivation that the scaling factor need only approximate the ReLU function to handle rare negative free energy values while preserving the signal for positive values. Consequently, precise tuning of this parameter is unnecessary. Selecting a sufficiently large value is a safe option to achieve the performance benefits of Energy-based conformal classification.

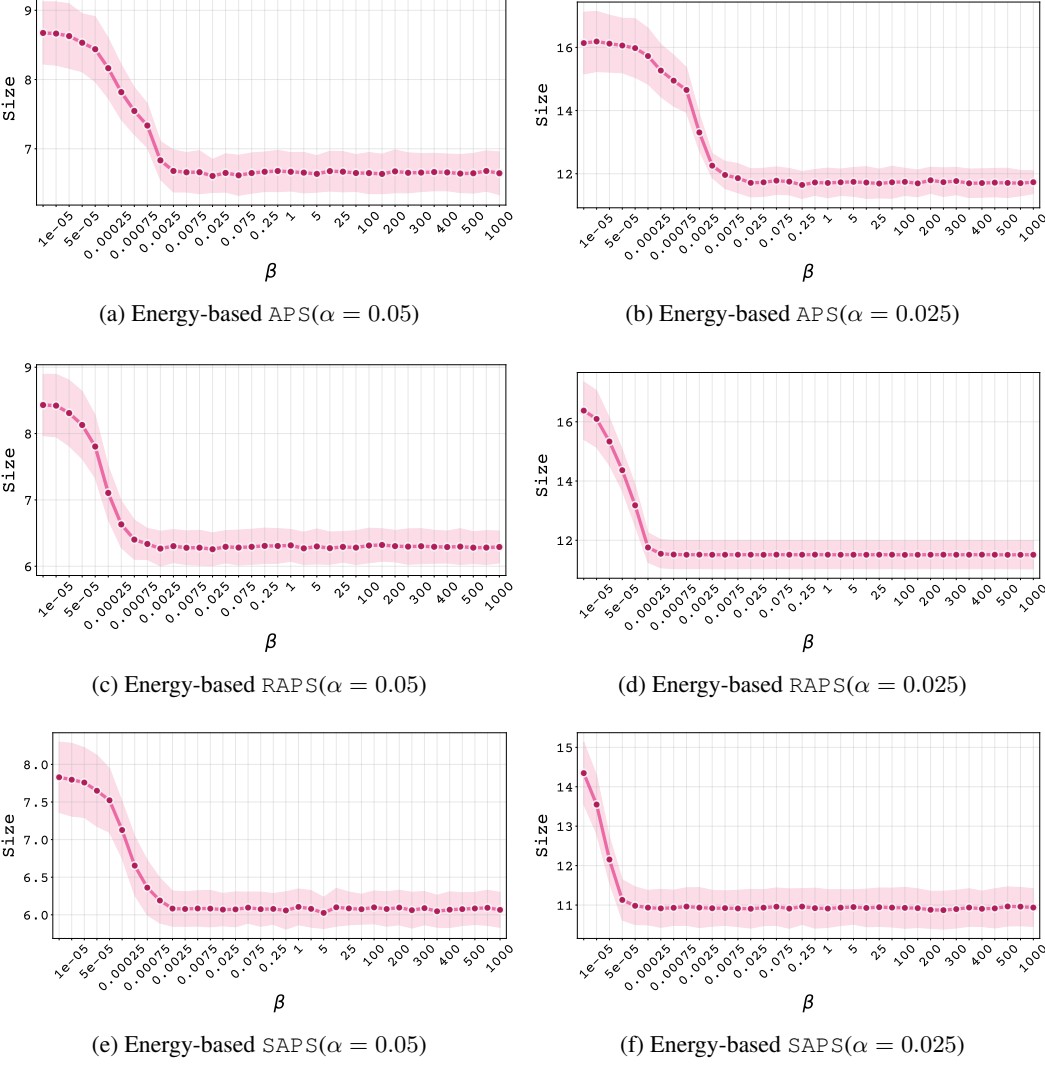

(a) Energy-based APS($\alpha = 0.05$)       (b) Energy-based APS($\alpha = 0.025$)

(c) Energy-based RAPS($\alpha = 0.05$)      (d) Energy-based RAPS($\alpha = 0.025$)

(e) Energy-based SAPS($\alpha = 0.05$)      (f) Energy-based SAPS($\alpha = 0.025$)

Figure 9: Ablation study of the parameter $\beta$ on CIFAR-100 with ResNet-56. The plots show the average prediction set size (shaded regions indicate standard deviation) as a function of $\beta$. Performance stabilizes for sufficiently large values of $\beta$, indicating that the method does not require specific tuning of this parameter.

## P ON THE RELIABILITY OF CONFORMAL CLASSIFIERS WHEN FACED WITH OOD TEST SAMPLES

A reliable system for uncertainty quantification is often expected to satisfy two primary objectives, as outlined in the Appendix of Angelopoulos & Bates (2021):

1. Flag out-of-distribution (OOD) inputs to avoid making predictions on unfamiliar data.

2. If an input is deemed in-distribution, output a prediction set that contains the true class with user-specified probability.

A practical strategy to achieve this is a two-stage pipeline: first, an OOD detector screens each input. If an input is flagged as OOD, the system can abstain (e.g., by returning an empty set). If deemed in-distribution, the input is passed to a conformal predictor to generate a valid prediction set. This separation, while effective, requires deploying and managing two distinct models.

The utility of the energy-based paradigm becomes particularly evident in scenarios where a dedicated OOD detection module is not available to filter inputs. In such cases, a standard conformal predictor, relying solely on softmax outputs, can be misleading. An OOD input might still produce a single, high-confidence softmax score, leading the base conformal method to output a small, high-confidence prediction set (e.g., {'Tuberculosis'}) for a non-medical image. This false confidence is a critical failure mode.

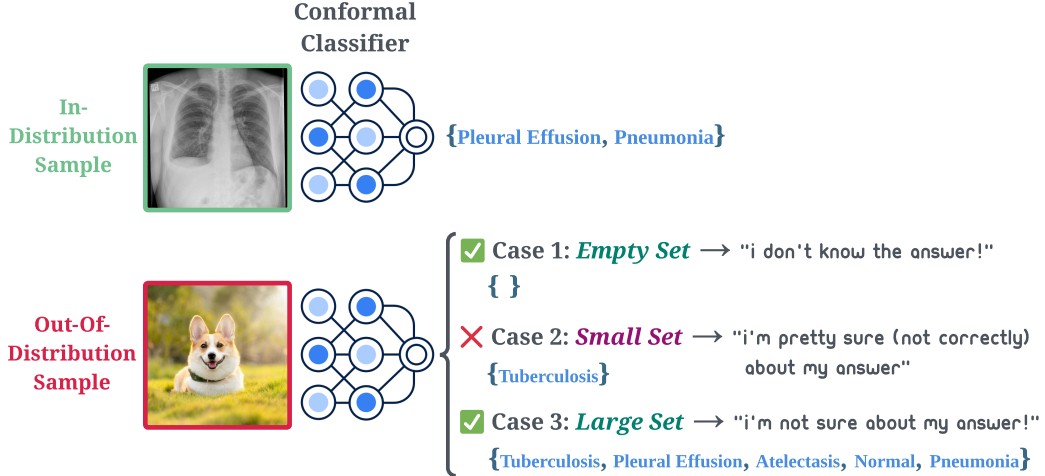

Figure 10: Conceptual diagram of a reliable conformal classifier facing an OOD input. The desired behaviors are to produce an empty or large set, both signaling uncertainty, and to avoid producing a small set that implies false confidence.

The energy-based approach addresses this vulnerability by incorporating a more reliable measure of model uncertainty. The key difference in behavior is:

- For *familiar ID inputs*, model confidence is high, resulting in a high negative free energy. The prediction sets are thus appropriately small and efficient, similar to the base variant.

- For *unfamiliar OOD inputs*, the model's uncertainty is captured by a low negative free energy. The energy-based reweighting dampens scores so that more classes fall below the calibrated quantile. This results in the predictor generating a large prediction set.

This is a clear improvement over the base variant, which is prone to producing small, overconfident sets for such inputs. Therefore, even when the system is not configured to abstain, the large prediction set generated by the energy-based method provides a more robust and honest signal of uncertainty. It reduces the risk of overconfident and incorrect predictions on OOD data, making it a more reliable choice in deployments without a separate OOD detector. This property validates

the use of free energy as a model-aware signal that overcomes the limitations of standard softmax probabilities, as noted in prior work (Liu et al., 2020; Wang et al., 2021).

To illustrate the importance of the desiderata outlined in Section 3.3, consider a conformal classifier trained to identify medical conditions from chest X-rays. If this model is fed with an image of a completely unrelated subject, such as a corgi, a reliable classifier must signal its unfamiliarity with the input. As visualized in Figure 10, this signal correctly manifests in two ways:

1. An **empty set** ($\emptyset$), which communicates: *"I don't know the answer."*
2. A **large set** (e.g., {'Tuberculosis', 'Pleural Effusion', 'Atelectasis', ...}), which communicates: *"I am uncertain about the answer."*

In contrast, producing a small, non-empty set (e.g., {'Tuberculosis'}) is misleading, as it incorrectly signals high confidence in a prediction that is likely wrong.

Our experimental results in Section 3.3 confirm this behavior in practice. We observed an increase in the average prediction set size for the energy-based variants of the nonconformity scores.

## Q  ENERGY-BASED REWEIGHTING VS. ENTROPY-BASED REWEIGHTING

A common measure of uncertainty in a classifier's output is the *Shannon Entropy* of its softmax probability distribution. As described by Luo & Colombo (2024), let $\mathbf{f}(x)$ be the logit vector produced by the classifier for an input $x$, and let $\hat{\pi}(k \mid x)$ be the softmax probability for class $k$. The Entropy $\mathbb{H}(x)$ is given by:

$$
\begin{aligned}
\mathbb{H}(x) &= -\sum_{k=1}^{K} \hat{\pi}(k \mid x) \log \hat{\pi}(k \mid x) \\
&= -\sum_{k=1}^{K} \hat{\pi}(k \mid x) \left( f_k(x) - \log \sum_{j=1}^{K} \exp(f_j(x)) \right) \\
&= -\sum_{k=1}^{K} \hat{\pi}(k \mid x) f_k(x) + \left( \log \sum_{j=1}^{K} \exp(f_j(x)) \right) \sum_{k=1}^{K} \hat{\pi}(k \mid x) \\
&= -\sum_{k=1}^{K} \hat{\pi}(k \mid x) f_k(x) + \log \sum_{j=1}^{K} \exp(f_j(x))
\end{aligned}
\tag{59}
$$

If one were to consider an "Entropy-based reweighting" for conformal scores, it would likely utilize this $\mathbb{H}(x)$ or a function thereof. However, the decomposition in Equation 59 reveals two distinct components influencing the Entropy value. The first term, $-\sum_{k=1}^{K} \hat{\pi}(k \mid x) f_k(x)$, depends on the alignment of softmax probabilities with the logit values. The second term is the `logsumexp` (LSE) of the logits: $L(x) = \log \sum_{j=1}^{K} \exp(f_j(x))$.

This LSE term, $L(x)$, is particularly relevant as it is directly related to the concept of free energy, which forms the basis of our proposed Energy-based nonconformity scores. It captures the overall magnitude or scale of the raw logits. Critically, while the softmax probabilities $\hat{\pi}(k \mid x)$ also depend on these logits, the LSE term $L(x)$ is calculated purely from the logits $f_j(x)$ without $\hat{\pi}(k \mid x)$ appearing as explicit factors within its own sum, unlike in the definition of entropy.

Indeed, softmax entropy is not well-suited for capturing epistemic uncertainty (Mukhoti et al., 2021). The distinction and potential advantage of using an Energy-based measure over the Entropy $\mathbb{H}(x)$ is illustrated in Figure 2. As shown, both the "easy" and "hard" samples can yield high softmax confidence for the predicted class, resulting in very low Entropy values (close to zero) for both. This suggests that Entropy alone might not adequately distinguish between an input for which the model is genuinely certain (high overall logit values, "easy sample") and an input where the model is less certain overall but still produces a peaky softmax distribution ("hard sample" with high softmax for one class). In contrast, the negative energy scores clearly differentiate these two cases: the easy sample exhibits a significantly higher negative energy score (32.49) compared to the hard

sample (14.08). This indicates that the Energy-based metric, by reflecting the overall scale of logit activation, provides a more nuanced signal of the model's underlying certainty.

Our proposed Energy-based Nonconformity Scores in Section 2.3 leverages this energy signal by reweighting a standard nonconformity score with $\text{softplus}(-F(x))$. The rationale is that $F(x)$ offers a more direct and potentially more robust indication of the model's overall certainty about an input $x$ than the Entropy $\mathbb{H}(x)$, which can be saturated (i.e., near zero) for different levels of underlying model certainty. Empirical comparisons supporting the benefits of Energy-reweighted scores over Entropy-reweighted alternatives is provided in Table 15.

Table 15: Performance comparison of different nonconformity score functions and their Energy-based and Entropy-based variants on ImageNet using a ResNet-50 classifier at miscoverage levels $\alpha = 0.05$ and $\alpha = 0.1$. Results are averaged over 10 trials and reported as empirical coverage and average prediction set size. **Bold** values indicate the best performance within each group. The Entropy-based variants of adaptive method, APS, RAPS, and SAPS are defined as $S_{\mathbb{H}}(x, y) = \frac{S(x,y)}{\mathbb{H}(x)}$, and for LAC it is defined as $S_{\mathbb{H}}(x, y) = -\hat{\pi}(y|x) \cdot \mathbb{H}(x)$.

| | | $\alpha = 0.1$ | | $\alpha = 0.05$ | |
|---|---|---|---|---|---|
| **Family** | | **Coverage** | **Set Size ↓** | **Coverage** | **Set Size ↓** |
| LAC | baseline | $0.898_{\pm 0.002}$ | $1.487_{\pm 0.013}$ | $0.950_{\pm 0.002}$ | $2.682_{\pm 0.039}$ |
| | w/ Energy | $0.898_{\pm 0.002}$ | $\mathbf{1.485}_{\pm 0.012}$ | $0.949_{\pm 0.002}$ | $\mathbf{2.680}_{\pm 0.043}$ |
| | w/ Entropy | $0.898_{\pm 0.002}$ | $1.496_{\pm 0.014}$ | $0.949_{\pm 0.002}$ | $2.696_{\pm 0.043}$ |
| APS | baseline | $0.899_{\pm 0.002}$ | $1.605_{\pm 0.022}$ | $0.950_{\pm 0.002}$ | $4.007_{\pm 0.164}$ |
| | w/ Energy | $0.899_{\pm 0.002}$ | $\mathbf{1.599}_{\pm 0.022}$ | $0.950_{\pm 0.002}$ | $\mathbf{3.842}_{\pm 0.159}$ |
| | w/ Entropy | $0.898_{\pm 0.003}$ | $2.159_{\pm 0.042}$ | $0.949_{\pm 0.002}$ | $4.990_{\pm 0.083}$ |
| RAPS | baseline | $0.898_{\pm 0.003}$ | $1.764_{\pm 0.030}$ | $0.949_{\pm 0.001}$ | $4.222_{\pm 0.056}$ |
| | w/ Energy | $0.898_{\pm 0.003}$ | $\mathbf{1.763}_{\pm 0.033}$ | $0.949_{\pm 0.001}$ | $\mathbf{3.889}_{\pm 0.057}$ |
| | w/ Entropy | $0.898_{\pm 0.003}$ | $2.108_{\pm 0.052}$ | $0.949_{\pm 0.002}$ | $4.811_{\pm 0.076}$ |
| SAPS | baseline | $0.898_{\pm 0.002}$ | $1.664_{\pm 0.034}$ | $0.949_{\pm 0.002}$ | $3.659_{\pm 0.073}$ |
| | w/ Energy | $0.898_{\pm 0.002}$ | $\mathbf{1.662}_{\pm 0.029}$ | $0.949_{\pm 0.002}$ | $\mathbf{3.654}_{\pm 0.064}$ |
| | w/ Entropy | $0.898_{\pm 0.003}$ | $2.101_{\pm 0.039}$ | $0.949_{\pm 0.002}$ | $4.702_{\pm 0.075}$ |

## R    X-CONDITIONAL COVERAGE ANALYSIS

Complementing the class-conditional analysis, we further assess the robustness of our method by evaluating validity conditional on the input features $X$. While exact conditional coverage is impossible in a distribution-free setting (Vovk, 2012; Foygel Barber et al., 2021), we employ the Worst-Slab Coverage (WSC) metric (Cauchois et al., 2021) to approximate the maximum coverage violation over local regions of the input space. We computed WSC using 100 random projections on the Places365 dataset.

Table 16 reports the Average Set Size and WSC. The results demonstrate that Energy-based variants achieve a notable reduction in prediction set size compared to the baselines while maintaining comparable WSC scores. This confirms that the proposed energy-based variants improve efficiency (smaller sets) without compromising local validity or "safety" on difficult slices of the input space.

Table 16: X-conditional performance comparison on the Places365 dataset (ResNet-50) using Worst-Slab Coverage (WSC). We compare Set Size and WSC at miscoverage levels $\alpha = 0.1$ and $\alpha = 0.05$. Lower values are better for Set Size; higher values (closer to $1 - \alpha$) are better for WSC. **Bold** values denote the best Set Size within each method family. The Energy-based variants reduce set size without degrading conditional coverage.

| | | $\alpha = 0.1$ | | $\alpha = 0.05$ | |
|---|---|---|---|---|---|
| **Method** | **Variant** | **Set Size ↓** | **WSC ↑** | **Set Size ↓** | **WSC ↑** |
| APS | w/o Energy | $9.02_{\pm 0.11}$ | $0.892_{\pm 0.00}$ | $20.98_{\pm 0.28}$ | $0.949_{\pm 0.00}$ |
| | w/ Energy | $\mathbf{8.54}_{\pm 0.18}$ | $0.893_{\pm 0.00}$ | $\mathbf{18.77}_{\pm 0.31}$ | $0.948_{\pm 0.00}$ |
| RAPS | w/o Energy | $8.99_{\pm 0.15}$ | $0.892_{\pm 0.00}$ | $21.46_{\pm 0.27}$ | $0.950_{\pm 0.00}$ |
| | w/ Energy | $\mathbf{8.47}_{\pm 0.18}$ | $0.891_{\pm 0.00}$ | $\mathbf{18.65}_{\pm 0.31}$ | $0.948_{\pm 0.00}$ |
| SAPS | w/o Energy | $8.80_{\pm 0.15}$ | $0.891_{\pm 0.00}$ | $21.13_{\pm 0.29}$ | $0.949_{\pm 0.00}$ |
| | w/ Energy | $\mathbf{8.49}_{\pm 0.15}$ | $0.891_{\pm 0.00}$ | $\mathbf{18.55}_{\pm 0.34}$ | $0.949_{\pm 0.00}$ |

## S    STATISTICAL SIGNIFICANCE TESTS

We validate our efficiency gains using a one-tailed Welch's t-test ($H_1 : \mu_{\text{Energy-Based}} < \mu_{\text{Base}}$). Tables 17 and 18 report the $p$-values. Results show statistically significant improvements across most settings, particularly at stricter $\alpha$ levels and on imbalanced data.

Table 17: Statistical significance ($p$-values from one-tailed Welch's t-test) for the experiments on balanced datasets presented in Table 1. **Bold** values indicate statistical significance ($p < 0.05$).

| Method | Comparison | $\alpha = 0.1$ $p$-value | $\alpha = 0.05$ $p$-value | $\alpha = 0.025$ $p$-value | $\alpha = 0.01$ $p$-value |
|---|---|---|---|---|---|
| **CIFAR-100 (ResNet-56)** | | | | | |
| `APS` | w/o vs. w/ `Energy` | 0.398 | **0.001** | **< 0.0001** | **< 0.0001** |
| `RAPS` | w/o vs. w/ `Energy` | 0.500 | **< 0.0001** | **< 0.0001** | **< 0.0001** |
| `SAPS` | w/o vs. w/ `Energy` | 0.500 | **< 0.0001** | **< 0.0001** | **< 0.0001** |
| **ImageNet (ResNet-50)** | | | | | |
| `APS` | w/o vs. w/ `Energy` | 1.000 | **0.036** | **< 0.0001** | **< 0.0001** |
| `RAPS` | w/o vs. w/ `Energy` | 0.268 | **< 0.0001** | **< 0.0001** | **< 0.0001** |
| `SAPS` | w/o vs. w/ `Energy` | 0.169 | 0.378 | **< 0.0001** | **< 0.0001** |
| **Places365 (ResNet-50)** | | | | | |
| `APS` | w/o vs. w/ `Energy` | **< 0.0001** | **< 0.0001** | **< 0.0001** | **< 0.0001** |
| `RAPS` | w/o vs. w/ `Energy` | **< 0.0001** | **< 0.0001** | **< 0.0001** | **< 0.0001** |
| `SAPS` | w/o vs. w/ `Energy` | **< 0.0001** | **< 0.0001** | **< 0.0001** | **< 0.0001** |

Table 18: Statistical significance ($p$-values) for the experiments on imbalanced data presented in Table 2 ($\lambda = 0.005$) and Table 10 ($\lambda = 0.02$). **Bold** values indicate statistical significance ($p < 0.05$).

| Method | Comparison | $\alpha = 0.1$ $p$-value | $\alpha = 0.05$ $p$-value | $\alpha = 0.025$ $p$-value | $\alpha = 0.01$ $p$-value |
|---|---|---|---|---|---|
| **CIFAR-100-LT ($\lambda = 0.005$)** | | | | | |
| `APS` | w/o vs. w/ `Energy` | **< 0.0001** | **< 0.0001** | **< 0.0001** | **< 0.0001** |
| `RAPS` | w/o vs. w/ `Energy` | **< 0.0001** | **< 0.0001** | **< 0.0001** | **< 0.0001** |
| `SAPS` | w/o vs. w/ `Energy` | **< 0.0001** | **< 0.0001** | **< 0.0001** | **< 0.0001** |
| **CIFAR-100-LT ($\lambda = 0.02$)** | | | | | |
| `APS` | w/o vs. w/ `Energy` | **< 0.0001** | **< 0.0001** | **< 0.0001** | **< 0.0001** |
| `RAPS` | w/o vs. w/ `Energy` | **< 0.0001** | **< 0.0001** | **< 0.0001** | **< 0.0001** |
| `SAPS` | w/o vs. w/ `Energy` | **< 0.0001** | **< 0.0001** | **< 0.0001** | **< 0.0001** |

# T    DETAILED RESULTS ON IMAGENET

The table 19 provides an evaluation of 3 nonconformity score across 16 different model architectures on the ImageNet validation set. We compare adaptive baseline methods (APS, RAPS, and SAPS) against their respective Energy-based counterparts.

Table 19: Comparison of average prediction set sizes and accuracy for conformal methods on ImageNet at confidence level of 95% ($\alpha = 0.05$). Lower average set size is better. Set sizes are shown as baseline (w/o Energy) $\rightarrow$ Energy-based (w/ Energy). The **bold** value highlights the superior (smaller) set size within each pair. The results are reported as the median of means over 10 trials.

| Model | Accuracy | | Average Set Size (w/o Energy $\rightarrow$ w/ Energy) $\downarrow$ | | |
|---|---|---|---|---|---|
| | Top-1 | Top-5 | **APS** | **RAPS** | **SAPS** |
| ResNet152 | 0.783 | 0.940 | 3.82 $\rightarrow$ **3.48** | 3.25 $\rightarrow$ **3.17** | 2.87 $\rightarrow$ **2.83** |
| ResNet101 | 0.774 | 0.936 | 3.97 $\rightarrow$ **3.70** | 3.60 $\rightarrow$ **3.32** | 3.15 $\rightarrow$ **3.03** |
| ResNet50 | 0.761 | 0.929 | 4.09 $\rightarrow$ **3.97** | 4.16 $\rightarrow$ **3.84** | 3.70 $\rightarrow$ **3.63** |
| ResNet34 | 0.733 | 0.914 | 9.86 $\rightarrow$ **8.29** | 9.94 $\rightarrow$ **7.73** | 9.57 $\rightarrow$ **7.41** |
| ResNet18 | 0.698 | 0.891 | 14.23 $\rightarrow$ **12.06** | 15.16 $\rightarrow$ **11.43** | 14.56 $\rightarrow$ **10.93** |
| VGG19 | 0.742 | 0.918 | 8.41 $\rightarrow$ **7.14** | 8.85 $\rightarrow$ **6.85** | 8.36 $\rightarrow$ **6.71** |
| VGG16 | 0.734 | 0.915 | 8.70 $\rightarrow$ **7.34** | 9.06 $\rightarrow$ **7.01** | 8.67 $\rightarrow$ **6.89** |
| VGG13 | 0.716 | 0.904 | 10.77 $\rightarrow$ **9.29** | 11.83 $\rightarrow$ **8.78** | 11.60 $\rightarrow$ **8.58** |
| VGG11 | 0.704 | 0.898 | 12.36 $\rightarrow$ **10.55** | 13.42 $\rightarrow$ **10.09** | 13.10 $\rightarrow$ **9.62** |
| ViT-B/16 | 0.811 | 0.953 | 4.70 $\rightarrow$ **4.29** | 4.10 $\rightarrow$ **3.56** | 3.53 $\rightarrow$ **3.24** |
| ViT-B/32 | 0.759 | 0.925 | 9.62 $\rightarrow$ **8.25** | 8.52 $\rightarrow$ **7.28** | 7.93 $\rightarrow$ **6.68** |
| Swin_s | 0.832 | 0.964 | **2.79** $\rightarrow$ 2.81 | 3.13 $\rightarrow$ **2.87** | 2.74 $\rightarrow$ **2.65** |
| Swin_t | 0.815 | 0.958 | **3.36** $\rightarrow$ 3.38 | 3.64 $\rightarrow$ **3.42** | 3.28 $\rightarrow$ **3.18** |
| EfficientNet_b4 | 0.834 | 0.966 | 5.87 $\rightarrow$ **4.99** | 4.87 $\rightarrow$ **4.28** | 4.33 $\rightarrow$ **3.93** |
| EfficientNet_v2_m | 0.851 | 0.972 | 5.93 $\rightarrow$ **5.60** | 5.16 $\rightarrow$ **5.16** | 4.86 $\rightarrow$ **4.69** |
| ShuffleNet_v2_x1_0 | 0.694 | 0.883 | 19.17 $\rightarrow$ **14.79** | 19.63 $\rightarrow$ **14.48** | 19.40 $\rightarrow$ **14.07** |
| **Average** | **0.765** | **0.929** | **7.98 $\rightarrow$ 6.87** | **8.02 $\rightarrow$ 6.45** | **7.60 $\rightarrow$ 6.13** |

# U    AI USAGE CLARIFICATION

Large Language Models were used to polish the writing of this manuscript by improving grammar, spelling, sentence flow, and overall readability. All research design, analysis, and interpretation were fully carried out and decided by the authors.

