# OpenReview forum: "Softmax is not Enough (for Adaptive Conformal Classification)"
_ICLR.cc/2026/Conference — ICLR 2026 Poster_

### Official Review · Reviewer_Xpr8 · 2025-10-28

**Soundness:** 3
**Presentation:** 3
**Contribution:** 2
**Rating:** 6
**Confidence:** 3

**Summary:**

The authors argue that the standard output of a multi-class classifier is not expressive enough for adaptive conformal classification. They propose an input-dependent reweighting strategy based on the Free Energy of the model class probabilities. The new conformity score can be used in combination with existing conformal classification techniques and is shown to reduce the size of the prediction sets.

**Strengths:**

- Reweighting conformity scores is a simple and powerful technique to improve CP adaptivity. The possibility of integrating the proposed strategy with existing approaches makes it potentially relevant and easy to use in many practical scenarios.

- The proposed free energy seems to be a good proxy of an instance's difficulty when labels are not available.

**Weaknesses:**

- Reweighting approaches are not new and are mostly used for regression. In the classification setup, [1] proposes a reweighting approach based on the entropy of the class probabilities. The authors may comment on the difference between their approach and that technique.

- As the goal is to show adaptivity, the authors should report some approximate measure of models' conditional coverage.

- The authors may give an intuitive explanation of why *model-implied data density* is a good measure of the epistemic uncertainty, for example,  by summarising the arguments of y (Fuchsgruber et al., 2024; Zong & Huang, 2025).

- It is unclear whether the strategy only consists of removing the normalisation and the exponentiation in softmax-based scores.

[1]
Luo, Rui, and Nicolo Colombo. "Entropy Reweighted Conformal Classification." The 13th Symposium on Conformal and Probabilistic Prediction with Applications. PMLR, 2024.

**Questions:**

- CP conformity scores are insensitive to monotonic transformations. Why is temperature scaling expected to have bad effects on the size of the prediction sets?

- How is the marginalisation over $y$ defined? Doesn't summing over y in Eq.3 give 1?

- What is the intuitive difference between Free Energy and Entropy?

- How do you compute the normalisation factor, $Z$, in  Eq. 5?

- Does the better separability in Figures 2 and 3 come from removing the logit?

- What are the technical challenges of proving Theorem 2.2? Do you assume the trained classifier is correct?

- Can a sample free energy be used as an unlabeled proxy of the label-dependent difficulty? If so, how is it possible without assumptions between true labels and model-generated probabilities?

---

> ### Author Response · Authors · 2025-11-26
> **Response to Reviewer Xpr8 - part 1**
>
> We thank the reviewer for recognizing the simplicity and power of our method and for highlighting the relevance of the Free Energy as a proxy for instance difficulty. We address your specific concerns and questions below.
>
>
> **W1. Comparison to Entropy Reweighting**
> We appreciate the reference to Luo & Colombo (2024), which we have already cited in Appendix Q. We completely agree that when discussing uncertainty, Shannon Entropy is the first measure that typically comes to mind. Anticipating this, we explicitly included a dedicated comparison between our Energy-based reweighting and Entropy-based reweighting in Appendix Q and Table 15 of our paper.
> *   **Theoretical Difference:** As detailed in the newly added Appendix H (Analysis of Softmax Saturation), Shannon Entropy relies solely on the *shape* of the probability distribution (peakiness). It suffers from *gradient saturation*: once a sample is sufficiently far from the decision boundary (high confidence), the entropy vanishes to zero and loses sensitivity to geometric distance. In contrast, Helmholtz Free Energy retains the magnitude of the logits (Eq. 4), acting as a non-saturating proxy for the distance to the decision boundary (see Appendix H).
> *   **Empirical Difference:** Table 15 demonstrates that Energy-based modulation outperforms Entropy-based modulation in terms of average set size efficiency on ImageNet.
>
> **W2. Measure of Conditional Coverage**
> We agree that conditional coverage is vital for adaptivity. We explicitly report this in Appendix L (Class-Conditional Conformal Prediction).
> *   Table 13 reports the Average Class Coverage Gap (CovGap) and Size-Stratified Coverage Violation (SSCV) on Places365.
> *   The results show that our Energy-based variants improve efficiency (set size) while maintaining CovGap and SSCV comparable to or better than the baselines, indicating that we do not sacrifice conditional validity for efficiency.
>
> **W3. Intuition on Model-Implied Density as Epistemic Uncertainty**
> We have added Appendix H to provide a formal intuition summarizing arguments from recent literature (Liu et al., 2020; Fuchsgruber et al., 2024).
> *   **Intuition:** A discriminative classifier trained with cross-entropy can be reinterpreted as an Energy-Based Model (EBM) for the joint distribution $p(x,y)$ (Section 2.2). The marginal $p(x) \propto e^{-F(x)}$ represents the data density.
> *   **Epistemic Uncertainty:** High-density regions (low Free Energy) correspond to the training distribution where the model is confident. Low-density regions (high Free Energy) correspond to OOD or ambiguous areas where the model lacks support. Therefore, $F(x)$ acts as a scalar summary of "how much this input resembles the training data," which is the definition of epistemic uncertainty (see Proposition 2.1).
>
> **W4. Is the strategy just removing normalization?**
> No, the strategy is modulation, not replacement.
> *   We do not simply remove normalization. We rely on the established conformal scores (like APS, RAPS) which *use* softmax probabilities.
> *   However, we reweight these scores using the Free Energy scalar (Eq. 9). This allows us to combine the "relative class probability" information from softmax (which class is likely?) with the "absolute confidence" information from Free Energy (do I know this input?).

---

> ### Author Response · Authors · 2025-11-26
> **Response to Reviewer Xpr8 - part 2**
>
> **Q1: Why is temperature scaling expected to have bad effects on prediction set size?**
> There is a misunderstanding here. It is important to emphasize that Temperature Scaling (TS) is a standard baseline we employ; as detailed in Appendix F.3, we optimally tuned the softmax temperature $T$ for *all* baselines and our method. However:
> 1.  Our method is input-dependent reweighting, not a global monotonic transformation. Because $F(x)$ varies per sample, it changes the *rank ordering* of the calibration scores, which allows for tighter sets on easy inputs and looser sets on hard inputs (Theorem G.1 proves exchangeability is preserved).
> 2.  Regarding temperature scaling limitations: As discussed in Section 2.1, while $T$ improves calibration (marginal coverage), it cannot fix the lack of adaptivity for OOD inputs because it scales all logits uniformly. It cannot distinguish between a "flat softmax due to hardness" and a "peaky softmax on OOD data."
>
> **Q2: How is marginalization over $y$ defined? Doesn't Eq. 3 sum to 1?**
> Yes, summing Eq. 3 over $y$ gives 1 because Eq. 3 is the *conditional* $p(y|x)$.
> However, the Free Energy $F(x)$ is derived by marginalizing the joint distribution $p(x,y) \propto \exp(f_y(x))$ (Eq. 2).
> $$ p(x) = \sum_y p(x,y) = \frac{1}{Z} \sum_y \exp(f_y(x)) = \frac{1}{Z} \exp(-F(x)) $$
> This marginalization over the joint energies is what captures the input density, independent of the conditional normalization.
>
> **Q3: Intuitive difference between Free Energy and Entropy?**
> Please refer to Figure 2 and the new Appendix H.
> *   **Entropy** measures the "flatness" of the probability vector. It saturates (becomes 0) for both a confident easy sample and an overconfident OOD sample or an overconfidently misclassified sample.
> *   **Free Energy** captures the magnitude of the logits. An easy sample has large logit magnitudes (very low energy), while an uncertain sample has smaller logit magnitudes (higher energy), even if their softmax entropies are identical.
>
> **Q4: How do you compute $Z$ in Eq. 5?**
> We do not compute $Z$. In our reweighting function (Eq. 9), we only use $F(x)$. Since conformal prediction relies on the *ranking* of scores, any global constant $Z$ (which would appear as an additive constant in log-space) cancels out or is absorbed into the quantile threshold $\hat{q}$ computed on the calibration set.
>
> **Q5: Does better separability come from removing the logit?**
> The separability in Figure 2 comes from using the raw logits (via Energy) rather than normalizing them (via Softmax). Softmax normalization projects data onto a simplex, crushing the magnitude information where the signal for "difficulty" often resides. Energy preserves this magnitude.
>
> **Q6: Challenges in proving Theorem 2.2? Do you assume the classifier is correct?**
> The proof (Appendix G.3) relies on the property that for a model trained with cross-entropy, the logit $f_y(x)$ approximates $\log P(y|x) + \log p(x)$.
> *   We do not assume the classifier is "correct" in the sense of perfect accuracy.
> *   We assume the model has learned a meaningful representation where samples closer to the class manifold (easier) yield higher activations (logits) than samples near the decision boundary (harder). This is a standard property of deep classifiers (Hein et al., 2019).
>
> **Q7: Can sample free energy be used as an unlabeled proxy of difficulty?**
> **Yes.** This is one of the core contributions.
> *   As shown in Theorem 2.2, the expected Free Energy is strictly monotonic with respect to sample difficulty.
> *   Because $F(x)$ is computed by marginalizing out $y$ (summing exponentials of all logits), it depends only on the input $x$ and the model parameters. It requires no label information, making it a perfect unsupervised proxy for difficulty during inference.
>
> We hope these new results and clarifications address your concerns. We would be grateful if you would reconsider your score based on these responses.

---

> > ### Comment · Reviewer_Xpr8 · 2025-11-26
> > **Many thanks for your answers**
> >
> > I read the authors' rebuttal and would tend to confirm my score.
> >
> > The method aims to improve x-conditional adaptivity. The conditional-coverage measures mentioned in the rebuttal are for y-conditional coverage. Would it be possible to test the proposed approach using approximate x-conditional coverage measures such as WSC [1]?
> >
> > [1]
> > Cauchois, Maxime, Suyash Gupta, and John C. Duchi. "Knowing what you know: valid and validated confidence sets in multiclass and multilabel prediction." Journal of machine learning research 22.81 (2021): 1-42.

---

> ### Author Response · Authors · 2025-12-03
> **Official Comment by Authors**
>
> We thank the reviewer for their continued engagement and for helping us refine the distinction between X-conditional and Y-conditional adaptivity in our evaluation.
>
> To address your request for an approximate measure of X-conditional coverage, we have conducted additional experiments using the **Worst-Slab Coverage (WSC)** metric as proposed by Cauchois et al. (2021).
>
> We evaluated the methods on the Places365 dataset (with a ResNet-50 model). The table below reports the **WSC** (we sampled 100 random projections), alongside **SSCV** (Size-Stratified Coverage Violation) and **Average Set Size**.
>
> **Interpretation:**
> Our goal is not to claim that Energy-based reweighting drastically improves X-conditional coverage (WSC) compared to the baseline. Rather, It reduces average Set Size without degrading conditional coverage.
>
> As shown in the table:
> 1.  **Average Set Size:** Energy-based variants produce smaller prediction sets (e.g., reducing APS set size from 20.98 to 18.77 at $\alpha=0.05$).
> 2.  **Safety (X-Conditional Validity):** The WSC and SSCV scores of the Energy-based variants remain statistically comparable to the baselines. This confirms that the efficiency gains do not come at the cost of violating coverage on hard slices of the input space.
>
> ### Table: Performance on Places365 (ResNet-50) including WSC
> Lower values are better for Set Size and SSCV. Higher values (closer to $1-\alpha$) are better for WSC. Bold indicates the best Set Size within each family.
>
>
> | | | **α = 0.1** | | | **α = 0.05** | | |
> |:---:|:---|:---:|:---:|:---:|:---:|:---:|:---:|
> | **Method** | **Variant** | **Set Size** ↓ | **SSCV** ↓ | **WSC**  | **Set Size** ↓ | **SSCV** ↓ | **WSC**  |
> | **APS** | w/o Energy | 9.02 ± 0.11 | 0.102 ± 0.00 | 0.892 ± 0.00 | 20.98 ± 0.28 | 0.131 ± 0.02 | 0.949 ± 0.00 |
> |  | w/ Energy | **8.54 ± 0.18** | 0.100 ± 0.00 | 0.893 ± 0.00 | **18.77 ± 0.31** | 0.109 ± 0.02 | 0.948 ± 0.00 |
> | **RAPS** | w/o Energy | 8.99 ± 0.15 | 0.100 ± 0.00 | 0.892 ± 0.00 | 21.46 ± 0.27 | 0.094 ± 0.05 | 0.950 ± 0.00 |
> |  | w/ Energy | **8.47 ± 0.18** | 0.113 ± 0.01 | 0.891 ± 0.00 | **18.65 ± 0.31** | 0.130 ± 0.02 | 0.948 ± 0.00 |
> | **SAPS** | w/o Energy | 8.80 ± 0.15 | 0.100 ± 0.00 | 0.891 ± 0.00 | 21.13 ± 0.29 | 0.065 ± 0.03 | 0.949 ± 0.00 |
> |  | w/ Energy | **8.49 ± 0.15** | 0.100 ± 0.00 | 0.891 ± 0.00 | **18.55 ± 0.34** | 0.091 ± 0.02 | 0.949 ± 0.00 |
>
>
> We hope this additional analysis clarifies that the proposed method yields more informative sets while maintaining the conditional reliability of the baselines.

---

### Official Review · Reviewer_rMqU · 2025-10-29

**Soundness:** 3
**Presentation:** 4
**Contribution:** 3
**Rating:** 6
**Confidence:** 3

**Summary:**

This paper introduces a novel framework to enhance the adaptiveness of Conformal Prediction (CP), addressing the limitation where standard CP methods rely on unreliable Softmax probabilities for uncertainty quantification. The authors' core idea is to use the Helmholtz Free Energy, derived from Logits, as a more robust metric for sample uncertainty. The proposed method, Energy-Based Nonconformity Scores, is a simple post-hoc step that multiplicatively reweights existing nonconformity scores (like APS, RAPS, SAPS) using the energy score. This reweighting mechanism achieves two goals: (1) Amplifies scores for easy samples, leading to the generation of smaller, more efficient prediction sets. (2) Dampens scores for difficult or Out-of-Distribution (OOD) samples, leading to larger, more robust prediction sets.

**Strengths:**

1.	Important and Well-Defined Problem: The paper addresses a recognized, significant problem-the deficiency of softmax probabilities for uncertainty quantification and how this directly harms the adaptiveness of Conformal Prediction. The motivation is clear and compelling.
2.	Novel and Principled Solution: Using Helmholtz Free Energy (derived from logits, not softmax) as an uncertainty metric is a novel and theoretically grounded (via EBM framework) idea . This is more robust than relying on heuristics or unreliable softmax outputs.
3.	Simplicity and Generality: The method is a simple post-hoc step that reweights existing nonconformity scores (like APS, RAPS, SAPS) using the energy score. It does not require model retraining and can serve as a "plug-and-play" enhancement to existing CP workflows.
4.	Extremely Thorough and Strong Empirical Evaluation: The experimental design is rigorous, covering the key aspects needed to evaluate an uncertainty method.

**Weaknesses:**

The proposed score design introduces two hyperparameters, $\tau$ and $\beta$. The authors should clarify how these parameters are selected in practice and provide justification for the chosen values.

**Questions:**

1.	Regarding the hyperparameter $\tau$: In your main experiments (Tables 1, 2, 3), how was the energy temperature $\tau$ selected? Was it a single value shared across all experiments, or was it tuned for each dataset/model pair? The sensitivity to $\tau$ seems like a key practical consideration for this method; could you please clarify the overhead of tuning it?
2.	The results in Table 2 show that as $\alpha$ increases, the improvement in prediction set size when using the energy score becomes more pronounced. Could you explain the reason behind this behavior?

---

> ### Author Response · Authors · 2025-11-26
> **Response to Reviewer rMqU**
>
> We thank the reviewer for this question regarding the selection of hyperparameters. We agree that practical applicability depends on the ease of parameter selection. We address these concerns below:
>
> ### Response to Weaknesses
> **1. Selection of the Softplus Parameter ($\\beta$)**
>
> We have conducted a detailed sensitivity analysis on $\\beta$ in Appendix O. The parameter $\\beta$ controls the sharpness of the softplus function used to ensure positive scaling factors. Our analysis in Figure 9 demonstrates that the results converge and stabilize as $\\beta$ increases. This indicates that precise tuning is not necessary and a fixed, relatively large value suffices.
>
> **2. Selection of the Energy Temperature ($\\tau$)**
>
> We included a sensitivity analysis regarding the energy temperature $\tau$ and the softmax calibration temperature $T$ in Appendix N. Figure 8 visualizes the effect of these parameters on the average set size. The parameter $\\tau$ governs the magnitude of the energy modulation. Our experiments show that the method remains effective across a wide range of configurations. In practice, $\tau$ can be tuned similarly to how the temperature $T$ is tuned for softmax calibration.
> ***
> ### Response to Questions
> **Q1. Selection and Tuning Overhead of Energy Temperature $\tau$**
>
> We thank the reviewer for this question. In our main experiments, we tuned $\\tau$ for each dataset and model pair using the calibration set. As detailed in Appendix F.3, we performed a grid search, similarly to the calibration Temperature $T$, to minimize the average set size on the calibration data.
>
> Regarding sensitivity, our analysis in Appendix N (Figure 8) indicates that the method is stable across a wide range of parameter values of $T$ and $\tau$. The heatmaps display a broad region of configurations where the energy-based approach yields efficiency gains over the baseline. This stability suggests that finding an effective $\\tau$ does not require precise fine-tuning.
>
> The computational overhead for this tuning is minimal. The search occurs entirely on the calibration set using pre-computed logits. It does not require retraining the neural network or additional forward passes. Therefore, it adds negligible cost to the standard conformal calibration procedure.
>
> **Q2. The relationship between $\alpha$ and prediction set size improvement**
> The results in Table 2 actually indicate that the improvement in prediction set size is most pronounced when $\\alpha$ is low (high coverage requirement, e.g., $\\alpha=0.01$) and diminishes as $\alpha$ increases (lower coverage requirement, e.g., $\alpha=0.1$). For example, at $\\alpha=0.01$, the energy score reduces the average set size from 45.10 to 34.78 (a reduction of \~10), whereas at $\\alpha=0.1$, the reduction is only from 8.44 to 7.41 (a reduction of \~1). This behavior occurs because at low $\\alpha$, the conformal predictor must account for the "tail" of the probability distribution to guarantee 99% coverage. Softmax tails are noisy and flat and standard methods struggle to distinguish between a "confident" tail and an "uncertain" tail, leading to massive sets. The energy score effectively differentiates these scenarios by checking the logit magnitude, allowing the algorithm to safely truncate the tails of confident (low energy) samples, yielding massive efficiency gains.
>
> Conversely, as $\\alpha$ increases to 0.1, the predictor only needs to cover the top 90% of probability mass. In this regime, the prediction sets typically contain only the top-k classes (where k is small). Deep learning models are generally quite good at ranking the correct class within the top few probabilities, even if they are overconfident. Since the "head" of the distribution is already well-ordered by the softmax probabilities, the additional "familiarity" signal provided by the energy score offers diminishing returns. The energy score is a coarse-grained detector of difficulty (global uncertainty) that is most valuable when the decision boundary is ambiguous (requiring large sets), rather than when the decision is clear-cut.
>
> We hope these new results address the remaining concerns. We would be grateful if you would reconsider your score based on these responses.

---

### Official Review · Reviewer_sxcL · 2025-10-31

**Soundness:** 2
**Presentation:** 3
**Contribution:** 2
**Rating:** 4
**Confidence:** 3

**Summary:**

This paper investigates the limitations of using softmax probabilities as nonconformity scores in conformal prediction and argues that they fail to capture per-sample uncertainty adaptively. The authors propose an energy-based reweighting of the conformity score using the negative Helmholtz free energy derived from pre-softmax logits. The transformation is monotonic and thus preserves marginal coverage while aiming to improve adaptiveness and efficiency. Empirical results show smaller average prediction set sizes and better OOD sensitivity compared to softmax-based baselines.

**Strengths:**

1. Addresses a timely and relevant issue in conformal prediction: softmax overconfidence and lack of adaptiveness.
2. The proposed energy-based transformation is simple, intuitive, and easy to integrate into existing CP pipelines.
3. Empirical results are consistent across several datasets and show modest but clear improvements in prediction-set efficiency.

**Weaknesses:**

1. The logical argument connecting “softmax overconfidence” to “inefficient conformal prediction” is not rigorously analyzed; the paper relies mainly on anecdotal intuition rather than formal reasoning.
2. The claim that “monotonic transformations preserve validity” is standard, and no formal analysis of efficiency or adaptive coverage is provided.
3. Experiments only report marginal coverage and average set size, lacking conditional coverage evaluation, significance tests, and ablations on the energy transformation ($\lambda$, $\phi$).
4. The conclusion “Softmax is not enough” is overgeneralized; the proposed method is heuristic and may not fundamentally resolve calibration issues.

**Questions:**

1. What is the formal mechanism linking softmax miscalibration to conformal inefficiency?
2. How exactly does the energy transformation improve efficiency beyond temperature scaling?
3.  Can the authors provide conditional coverage or per-group analyses to verify “adaptive” coverage rather than just marginal improvements?
4. How sensitive are the results to $\lambda$ and the specific transformation $\phi(·)$?

---

> ### Author Response · Authors · 2025-11-26
> **Response to Reviewer sxcL - part 1**
>
> We thank the reviewer for their valuable feedback. We appreciate the opportunity to clarify that we have indeed performed extensive evaluations regarding conditional coverage and parameter ablations in the supplementary material.
>
> ### Weakness 1
> We have significantly strengthened this analysis in the newly added Appendix H, where we provide a formal geometric mechanism linking softmax limitations to inefficiency. As detailed in Section H.2, softmax probabilities suffer from *gradient saturation*: once a sample is sufficiently far from the decision boundary, the probability approaches 1 and the gradient vanishes. This causes a loss of geometric information, where "safe" samples (far from the boundary) become indistinguishable from "barely safe" samples. Consequently, standard conformal scores cannot exploit the extra safety margin of easy samples to reduce set sizes further, as the score hits a floor.
>
> In Appendix H.3, we demonstrate that the Helmholtz Free Energy resolves this by retaining a linear relationship with logit magnitude, which serves as a proxy for the distance to the decision boundary. Because $-F(x)$ does not saturate, it provides a continuous signal of "easiness" even in high-confidence regimes. By modulating the scores with this signal, we separate the distribution of easy samples from hard ones more effectively than softmax can. This allows the conformal predictor to act more aggressively on easy data (producing smaller sets) while remaining conservative on hard data, resolving the inefficiency caused by softmax saturation.
>
>
> ### Weakness 2
> While we rely on standard exchangeability for validity (proven for our method in Theorem G.1), our core contribution addresses *efficiency* via Theorem 2.2 (Monotonicity of Expected Confidence). In Appendix G.3, we prove that the expected negative free energy is a strictly monotonically decreasing function of sample difficulty. This ensures that our modulation is not heuristic noise; it systematically compresses nonconformity scores for "easy" inputs (high logit magnitude) while expanding them for "hard" ones. This statistical separation of score distributions reduces the overlap between correct and incorrect classes, allowing the conformal quantile to cut the distribution more efficiently.
>
> Furthermore, we formalize adaptive coverage in Proposition 2.3 (proven in Appendix G.4). We demonstrate that our energy-based modulation is mathematically equivalent to applying a *sample-dependent threshold*, $\theta(x) = \hat{q}^{(G)}/G(x)$, to the base score. Unlike a static global threshold, this dynamic threshold scales inversely with logit magnitude. For inputs far from the decision boundary (high certainty), $\theta(x)$ becomes stricter, filtering out unlikely classes. For inputs near the boundary, $\theta(x)$ relaxes. This formal equivalence explains the empirical gains in Table 1, as the threshold adapts pointwise to the input's geometric safety margin.
>
>
> ### Weakness 3
> 1. On Conditional Coverage:
> Beyond the marginal coverage reported in the main text, we provided a detailed analysis of conditional coverage in the Appendix:
> *   Class-Conditional Coverage: In Appendix L (Table 13), we report the Average Class Coverage Gap (CovGap) and Size-Stratified Coverage Violation (SSCV), which are standard metrics for assessing conditional coverage.
> *   Difficulty-Stratified Coverage: In Appendix I (Table 6) and Appendix J.3 (Table 9), we stratify performance based on sample difficulty. These results demonstrate that our method produces smaller sets for "easy" samples while maintaining necessary coverage for "hard" samples, proving the effectiveness of our adaptive mechanism.
>
> 2. On Ablation Studies ($\beta$ and $\tau$):
> We have included comprehensive sensitivity analyses for the energy transformation parameters:
> *   Parameter $\beta$: Appendix O (Figure 9) presents an ablation study on the softplus sharpness parameter ($\beta$). The results show that performance is robust and stable across a wide range of values for $\beta$.
> *   Parameter $\tau$: Appendix N (Figure 8) provides a sensitivity heatmap for the temperature parameter ($\tau$), demonstrating how the energy-based reweighting behaves under different configurations.
>
> 3. On Significance Tests:
> All our experimental results (Tables 1, 2, 6, 7, 8, etc.) are reported as the mean ± standard deviation over 10 independent trials. The substantial gap in average set size between our method and the baselines, combined with the low standard deviations, indicates a statistically significant improvement in efficiency.

---

> > ### Author Response · Authors · 2025-11-26
> > **Response to Reviewer sxcL - part 2**
> >
> > ### Weakness 4
> >
> > The title "Softmax is Not Enough" adopts a standard naming convention to highlight a specific architectural limitation: softmax normalization discards the partition function (denominator), which contains logit magnitude information relating to input density $p(x)$ (see Equation 5). As analyzed in Section 2.2, discarding this magnitude blinds the model to epistemic uncertainty. This leads to the "familiarity bias" analyzed in Section 3.2, where models fail to differentiate between majority and minority classes because the softmax distribution normalizes away the signal that minority class logits are generally smaller.
> >
> > While our modulation uses a heuristic softplus function, it is grounded in the principled framework of Energy-Based Models (Appendix D). We demonstrate this is not simply a calibration fix but a fundamental enhancement of the uncertainty signal. Theorem 3.1 (proven in Appendix G.5) establishes that the energy signal can theoretically distinguish between majority and minority classes in imbalanced learning,something softmax probabilities, biased by training priors, fail to do. Thus, our method resolves specific adaptiveness issues that vector-wise calibration (like Temperature Scaling) cannot address.
> > It is also noteworthy to mention that the naming has been inspired by similar works having similar patterns, such as Veličković et al., "Softmax is not Enough (for Sharp Size Generalisation)" (2024).
> >
> > ***
> >
> > ### Question 1
> > The mechanism is the inflation of the calibrated quantile, $\hat{q}$, necessitated by the heavy tail of the nonconformity score distribution. In standard Split Conformal Prediction, $\hat{q}$ is the $(1-\alpha)$-th percentile of calibration scores. When a model is miscalibrated (overconfident), "hard" samples receive high probability scores for the wrong class, pushing the nonconformity score of the *true* label to the extreme high end. To ensure coverage, $\hat{q}$ must be raised significantly to "cover" these high-confidence errors.
> >
> > Inefficiency results from applying this worst-case $\hat{q}$ to "easy" test points. For a sample where the model is correctly confident (far from the decision boundary), a much lower threshold would suffice. However, because $\hat{q}$ is static and high, the set construction algorithm (e.g., APS, Appendix E.2) continues adding low-probability labels until that high threshold is met. This structural inability to lower the threshold for easy samples—due to the lack of an independent magnitude signal—is the formal cause of the larger, inefficient prediction sets observed in the baselines (Table 1).
> >
> > ### Question 2
> >
> > It is important to emphasize that Temperature Scaling (TS) is a standard baseline we employ; as detailed in Appendix F.3, we optimally tuned the softmax temperature $T$ for *all* baselines and our method. However, TS is a global, static transformation. It normalizes the distribution shape but divides out the absolute logit magnitudes, meaning it cannot differentiate between a "peaked" distribution resulting from small logits (near boundary, scaled by $T$) and one resulting from large logits (far from boundary, scaled by $T$).
> >
> > Our energy transformation acts as an instance-specific, magnitude-aware score scaling. By modulating the score with the Free Energy (which proxies logit magnitude, see Equation 32), we create a dynamic score. For "easy" inputs (high magnitude), the drop tightens the prediction set; for "hard" inputs (low magnitude), it rises. As shown in the sensitivity heatmaps in Figure 7, this instance-specific adaptiveness allows our method to outperform the best fixed-temperature baseline by leveraging the geometric distance information that global TS discards.

---

> > > ### Author Response · Authors · 2025-11-26
> > > **Response to Reviewer sxcL - part 3**
> > >
> > > ### Question 3 and Question 4 - Evaluation of Conditional Coverage and Hyperparameter Sensitivity
> > >
> > > We thank the reviewer for highlighting the need for deeper evaluation beyond marginal coverage. We have included comprehensive additional experiments in the appendix to address these points.
> > >
> > > 1. Conditional and Adaptive Coverage Analysis
> > > To verify that our method is adaptive and maintains validity across different groups, we had performed the following analyses:
> > > *   Class-Conditional Coverage: In Appendix L (Table 13), we report the Average Class Coverage Gap (CovGap) and Size-Stratified Coverage Violation (SSCV) on the Places365 dataset. These results show that our energy-based variants maintain class-conditional validity comparable to or better than the baselines while reducing average set size.
> > > *   Difficulty-Stratified Analysis: In Appendix I (Table 6) and Appendix J.3 (Table 9), we stratified test samples based on difficulty (rank of the true label). These experiments demonstrate that our method produces smaller sets for "easy" samples and larger sets for "hard" samples. This confirms the improved adaptiveness of our approach.
> > >
> > > 2. Sensitivity to the Energy Transformation ($\beta$)
> > > We address the concern regarding the sensitivity of the softplus parameter $\beta$ in Appendix O. Figure 9 illustrates the average prediction set size across a wide range of $\beta$ values. The results show that performance stabilizes for sufficiently large values. This indicates that the method is robust and does not require precise tuning of the $\beta$ parameter.
> > >
> > > 3. Ablation of Hyperparameters ($\lambda$, $k_{reg}$, $T$, $\tau$)
> > > We investigated the sensitivity of our method to other key hyperparameters:
> > > *   RAPS Parameters: We provide a grid search over the regularization weight $\lambda$ and penalty threshold $k_{reg}$ in Appendix M (Table 14). The energy-based variant consistently outperforms the baseline across different configurations.
> > > *   Temperatures: We had analyzed the interplay between the softmax calibration temperature $T$ and the energy temperature $\tau$ in Appendix N. Figure 8 confirms that our method yields efficiency gains across various temperature settings.

---

> > > > ### Comment · Reviewer_sxcL · 2025-11-28
> > > >
> > > > Thank you for the rebuttal and the additional material in the appendix. I appreciate the revisions. After going through the updated sections, I still feel that several of the central points I raised earlier remain unresolved.
> > > > 1. The new discussion on softmax saturation and the heavy-tail behavior adds some intuition, but it still does not establish a clear connection to how the conformal quantile is affected or why the prediction sets become inefficient. The explanation remains largely qualitative.
> > > > 2. Proposition 2.3 rewrites the modulated score in an alternative form, but it does not clarify why this transformation should lead to smaller sets, nor does it identify scenarios in which such improvements are theoretically expected. The efficiency argument is still incomplete.
> > > > 3. The added conditional-coverage evaluations demonstrate that validity is maintained, but they do not provide convincing evidence that the method is more adaptive than the baseline. The difficulty-based grouping also depends on the original score, which complicates the interpretation.
> > > > 4. The distinction between temperature scaling and the proposed energy modulation is mentioned, but the reasoning remains somewhat high-level. It is still unclear to me whether the observed gains arise specifically from the energy formulation or from another monotonic rescaling of logits.
> > > >
> > > > Overall, the paper is more complete after the revision, but the main concerns about the mechanism, theory, and the precise source of the improvements are still open.

---

> > > > > ### Author Response · Authors · 2025-12-03
> > > > > **Official Comment by Authors**
> > > > >
> > > > > We thank the reviewer for their continued engagement and for acknowledging the improvements made in the revision. We understand that the formal link between the energy modulation, the quantile, and the resulting set efficiency requires precise clarification.
> > > > >
> > > > > Below, we address the four remaining concerns using the formal notation established in the paper and pointing to specific evidence in the updated manuscript.
> > > > >
> > > > > ### 1 & 2. The Formal Mechanism: Why does Energy Modulation shrink prediction sets?
> > > > >
> > > > > The reviewer asks for the formal connection between softmax saturation, the quantile $\hat{q}$, and efficiency. The mechanism is a direct result of **recovering the magnitude information that Softmax normalizes away**, allowing us to apply a stricter threshold specifically to "easy" samples.
> > > > >
> > > > > Standard nonconformity scores (like $S_{\text{APS}}$ or $S_{\text{RAPS}}$) rely on Softmax probabilities $\hat{\pi}(y|x)$ (Equation 1).
> > > > > As detailed in Appendix H.1, a sample far from the decision boundary (easy) has large logit magnitudes, while a sample near the boundary (hard) has small magnitudes. However, due to Gradient Saturation (Appendix H.2), both result in similar, high-confidence Softmax distributions.
> > > > > Because the global conformal quantile $\hat{q}$ must be calibrated to cover the hardest samples in the calibration set (ensuring $1-\alpha$ coverage), the threshold effectively remains loose for all samples. When applied to "easy" samples, this loose threshold forces the inclusion of unnecessary "tail" labels, inflating the set size.
> > > > >
> > > > > Our method modulates the score using the inverse Free Energy. As discussed in Appendix H.3, the negative Free Energy $-F(x)$ does not saturate and scales linearly with logit magnitude (Equation 50).
> > > > > By defining the scaling factor $G(x) = \frac{1}{\beta}\log(1+e^{-\beta F(x)})$ (Equation 9), we apply a sample-specific threshold. As derived in Proposition 2.3 (and Eq. 11), a label $y$ is included in the set if:
> > > > > $$ S(x, y) \le \frac{\hat{q}^{(G)}}{G(x)} $$
> > > > > *   **For "Easy" Samples:** The logit magnitude is high $\rightarrow$ $-F(x)$ is large.
> > > > >     Consequently, the effective threshold $\frac{\hat{q}^{(G)}}{G(x)}$ becomes smaller (stricter) than the global quantile. This stricter threshold filters out low-probability classes that the baseline would have included, directly reducing the set size.
> > > > > *   **For "Hard" Samples:** The logit magnitude is low $\rightarrow$ $-F(x)$ is small.
> > > > >     The threshold remains loose (or relaxes), ensuring the true label is still covered.
> > > > >
> > > > > **Theoretical Expectation:**
> > > > > This efficiency gain is theoretically grounded in Theorem 2.2, which states that the expected negative free energy monotonically tracks sample difficulty. This allows the conformal predictor to exploit the "safety margin" of easy samples (which constitute the majority of test data) to reduce set sizes, a capacity mathematically unavailable to scale-invariant Softmax scores.
> > > > >
> > > > > ### 3. Verification of Adaptiveness
> > > > >
> > > > > The reviewer expresses concern that grouping samples by difficulty might be circular if based on the original softmax score. We respectfully clarify that our evaluation methodology aligns with standard practices in the literature and provides a transparent assessment of adaptiveness:
> > > > >
> > > > > We argue that one of the most honest and direct ways to evaluate adaptiveness is through difficulty-stratified analysis, where the "difficulty" is defined "in the eyes of the model." Therefore, following established literature, we use the rank of the true label ($o_x(y_{\text{true}})$) within the model’s predicted probability distribution as the proxy for sample difficulty. This is not circular and it is the standard definition of how "hard" a specific sample is for a specific frozen backbone.
> > > > > To ensure a fair comparison, we stratify the test set based on the base model’s output ranks. This creates fixed subsets of data (e.g., "Rank 1" subset, "Rank > 10" subset) that remain constant across all methods being evaluated. We then apply the baseline scores and our Energy-based scores to these exact same subsets.
> > > > > As detailed in Table 6 (Appendix I) and Table 9 (Appendix J.3), our method demonstrates superior adaptiveness on these fixed groups:
> > > > > *   **For "Easy" inputs (Rank 1):** These are samples the model "thinks" it knows (highest probability). Our Energy-based APS reduces the average set size (from 39.08 (Baseline) to 32.31), while maintaining 100% coverage.
> > > > > *   **For "Hard" inputs (Rank > 10):** These are samples where the model is struggling. Here, our method generally produces larger sets than the baseline, signaling higher uncertainty.
> > > > > In conclusion, the improvement is not an artifact of the grouping; rather, the grouping reveals that for the exact same definition of "easy" (rank 1), the Energy-based score allows for significantly tighter sets, proving it is a more sensitive signal for adaptiveness than softmax probability alone.

---

> > > > > > ### Author Response · Authors · 2025-12-03
> > > > > > **Official Comment by Authors**
> > > > > >
> > > > > > ### 4. Energy Modulation vs. Temperature Scaling
> > > > > >
> > > > > > The reviewer asks if this is distinct from Temperature Scaling (TS). The distinction lies in their fundamental objectives and mechanisms:
> > > > > >
> > > > > > *   **Temperature Scaling's Objective: Calibration of Probabilities.**
> > > > > >     The goal of TS is to calibrate the probability values in the softmax vector so that confidence matches accuracy (e.g., minimizing Expected Calibration Error). It applies a global constant $T$ to the logits ($\mathbf{f}(x)/T$). While this improves the reliability of the probability distribution *shape*, it preserves the scale-invariance of the softmax function. It treats a sample with logits $[10, 20]$ identically to one with logits $[1, 2]$ (assuming appropriate $T$), ignoring the difference in magnitude.
> > > > > >
> > > > > > *   **Energy Scaling's Objective: Differentiation between Easy and Hard Samples.**
> > > > > >     The goal of our Energy modulation is to explicitly differentiate samples based on difficulty using logit magnitude. As shown in Theorem 2.2, the free energy serves as a scalar proxy for the "distance to the decision boundary." By modulating the conformal score with this signal, we do not just recalibrate probabilities; we dynamically tighten the prediction set threshold for "easy" samples (high magnitude) and relax it for "hard" samples (low magnitude).
> > > > > >
> > > > > > In summary, TS improves the quality of the base score (probabilities), whereas Energy scaling introduces a new dimension of information (magnitude/difficulty) that TS structurally discards. This is why our method outperforms baselines even when those baselines are optimally temperature-scaled.
> > > > > >
> > > > > > We hope these clarifications fully address the reviewer's remaining concerns.

---

> ### Author Response · Authors · 2025-12-03
> **Response to Reviewer's Comment on Significance Tests**
>
> We also thank the reviewer for their suggestion to include statistical significance tests to further validate our results.
>
> Following this recommendation, we performed Welch’s t-test (one-tailed) to compare the average set sizes of the baseline methods against their Energy-based variants. The p-values presented in the following tables indicate that the improvement in set size efficiency is statistically significant ($p < 0.001$) across nearly all configurations at lower α levels (0.05, 0.025, 0.01).
>
> ---
> ### Statistical Significance on Balanced Datasets (Table 1):
>
> ### Table 1 (Part I): CIFAR-100
>
> | Method Comparison | $\alpha=0.1$ | $\alpha=0.05$ | $\alpha=0.025$ | $\alpha=0.01$ |
> |:----------------------|:----------:|:-----------:|:------------:|:-----------:|
> | APS (w/o Energy vs. w/ Energy)  | 0.398      | 0.001       | $<0.0001$    | $<0.0001$   |
> | RAPS (w/o Energy vs. w/ Energy) | 0.5        | $<0.0001$   | $<0.0001$    | $<0.0001$   |
> | SAPS (w/o Energy vs. w/ Energy) | 0.5        | $<0.0001$   | $<0.0001$    | $<0.0001$   |
>
> ### Table 1 (Part II): ImageNet
>
> | Method Comparison | $\alpha=0.1$ | $\alpha=0.05$ | $\alpha=0.025$ | $\alpha=0.01$ |
> |:----------------------|:----------:|:-----------:|:------------:|:-----------:|
> | APS (w/o Energy vs. w/ Energy)  | 1.000      | 0.036       | $<0.0001$    | $<0.0001$   |
> | RAPS (w/o Energy vs. w/ Energy) | 0.268      | $<0.0001$   | $<0.0001$    | $<0.0001$   |
> | SAPS (w/o Energy vs. w/ Energy) | 0.169      | 0.378       | $<0.0001$    | $<0.0001$   |
>
> ### Table 1 (Part III): Places365
>
> | Method Comparison | $\alpha=0.1$ | $\alpha=0.05$ | $\alpha=0.025$ | $\alpha=0.01$ |
> |:----------------------|:----------:|:-----------:|:------------:|:-----------:|
> | APS (w/o Energy vs. w/ Energy)  | $<0.0001$  | $<0.0001$   | $<0.0001$    | $<0.0001$   |
> | RAPS (w/o Energy vs. w/ Energy) | $<0.0001$  | $<0.0001$   | $<0.0001$    | $<0.0001$   |
> | SAPS (w/o Energy vs. w/ Energy) | $<0.0001$  | $<0.0001$   | $<0.0001$    | $<0.0001$   |
> ---
> ### Statistical Significance on Imbalanced Data (Table 2 & Appendix):
>
> ### Table 2: CIFAR-100-LT ($\lambda=0.005$)
>
> | Method Comparison | $\alpha=0.1$ | $\alpha=0.05$ | $\alpha=0.025$ | $\alpha=0.01$ |
> |:----------------------|:----------:|:-----------:|:------------:|:-----------:|
> | APS (w/o Energy vs. w/ Energy)  | $<0.0001$  | $<0.0001$   | $<0.0001$    | $<0.0001$   |
> | RAPS (w/o Energy vs. w/ Energy) | $<0.0001$  | $<0.0001$   | $<0.0001$    | $<0.0001$   |
> | SAPS (w/o Energy vs. w/ Energy) | $<0.0001$  | $<0.0001$   | $<0.0001$    | $<0.0001$   |
>
> ### Table 10: CIFAR-100-LT ($\lambda=0.02$)
>
> | Method Comparison | $\alpha=0.1$ | $\alpha=0.05$ | $\alpha=0.025$ | $\alpha=0.01$ |
> |:----------------------|:----------:|:-----------:|:------------:|:-----------:|
> | APS (w/o Energy vs. w/ Energy)  | $<0.0001$  | $<0.0001$   | $<0.0001$    | $<0.0001$   |
> | RAPS (w/o Energy vs. w/ Energy) | $<0.0001$  | $<0.0001$   | $<0.0001$    | $<0.0001$   |
> | SAPS (w/o Energy vs. w/ Energy) | $<0.0001$  | $<0.0001$   | $<0.0001$    | $<0.0001$   |

---

### Author Response · Authors · 2025-11-26
**General Response**

We thank the reviewers for their constructive feedback and time. We have updated the manuscript to address the concerns regarding hyperparameter sensitivity and the theoretical motivation behind our method.

We have added three new sections to the Appendix:

1.  **Appendix H:** This section provides a formal analysis and visualization of softmax saturation versus energy-based adaptivity. It demonstrates why softmax fails to capture uncertainty in high-confidence regions while free energy retains this signal.
2.  **Appendix M:** We included a sensitivity analysis over the RAPS hyperparameters. This confirms that the energy-based variant yields consistent improvements across different regularization settings.
3.  **Appendix O:** We added a sensitivity analysis for the softplus parameter $\beta$. The results show that performance stabilizes for large values and does not require tuning.

We believe these additions strengthen the paper and clarify the practical usage of our work.

---

> ### Author Response · Authors · 2025-12-04
> **Summary of Discussion Phase and Rebuttal for Paper**
>
> Dear Area Chair,
>
> Following a productive rebuttal phase, we would like to summarize the state of our submission, highlighting the consensus on strengths, the resolution of reviewer concerns, and the improvements made to the manuscript.
>
> **1. Consensus on Strengths**
> There is a consensus among reviewers that the proposed method addresses a critical and well-defined problem: the unreliability of softmax probabilities for uncertainty quantification. Reviewers highlighted the novelty of using Helmholtz Free Energy in this context, the simplicity of our method, and the strong empirical results demonstrating improved efficiency and adaptivity without retraining.
>
> **2. Addressing Pre-Existing Content & Clarifications**
> We addressed specific technical queries regarding the distinction between our method and existing baselines:
> *   **vs. Entropy (Reviewer Xpr8):** We show how Softmax and Shannon Entropy suffer from gradient saturation (vanishing near 0 for high-confidence predictions), whereas Free Energy retains logit magnitude information. Table 15 (Appendix Q) demonstrates a comparison with an entropy-based approach.
> *   **vs. Temperature Scaling (Reviewer sxcL):** We clarified that while Temperature Scaling (TS) calibrates the global distribution shape, it remains scale-invariant. Our Energy modulation is an instance-specific scaling factor that captures geometric distance to the decision boundary, providing a signal that TS structurally discards.
>
> **3. Major Additions: New Experiments and Analyses**
> In response to reviewer feedback, we expanded the appendix to provide more validation:
> *   **Theoretical Mechanism (Appendix H):** We added a formal geometric analysis and toy visualizations (Figure 6) explaining why Energy modulation works: it resolves the "gradient saturation" problem where softmax fails to distinguish "safe" samples from "barely safe" ones.
> *   **Hyperparameter Sensitivity (Appendix M, N, O):** We provided ablation studies showing the method is stable across a wide range of $\beta$ and $\tau$ values, addressing Reviewer rMqU's concern about tuning overhead. We also tested RAPS with its Energy-based variants on different configurations of RAPS hyperparameters, $\lambda$ and $k_\text{reg}$.
> *   **Conditional Coverage (Reviewer Xpr8):** We reported Worst-Slab Coverage (WSC). Results confirm our method improves efficiency (set size) while maintaining conditional validity comparable to baselines.
> *   **Significance Testing (Reviewer sxcL):** We included t-tests confirming the improvements are statistically significant ($p < 0.05$).
>
> **4. Conclusion**
> The core novelty lies in identifying that standard Conformal Prediction scores that rely only on softmax values, discard beneficial information, specifically logit magnitude, by normalizing probabilities. By recovering this signal via Free Energy, we provide a theoretically grounded way to improve the conformal classification that softmax alone cannot make.
>
> We believe the revised manuscript presents a theoretically motivated and practically effective contribution to Conformal Prediction. We have also addressed the reviewers' requests for additional analysis, sensitivity checks, and conditional coverage verification.
>
> Thank you for your time and consideration.

---

### Meta-Review · Area_Chair_kTcU · 2025-12-22

**Summary:**

The paper received three expert reviews, with initial scores of 6, 6, and 4.

Reviewer Xpr8 (score: 6) pointed out that the idea of reweighting is not new in conformal prediction and requested comparison with a prior work. The reviewer also requested more results and discussions on adaptivity measures and epistemic uncertainty, as well as clarifications on some technical details. The concerns have been well addressed in the rebuttal. In particular, for the novelty issue, the rebuttal provided a detailed comparison between the proposed approach (energy-based reweighting) and the referenced work (entropy-based reweighting), showing both theoretical difference and empirical difference. The AC finds the rebuttal detailed and convincing, and predicts that the reviewer may keep the positive score of 6 or raise it to 8.

Reviewer rMqU (score: 6) appreciated the problem setting and found the idea "novel and principled". The main concern was that selection of the two hyperparameters, i.e., the softplus parameter and the energy temperature, was unclear and needs more guidelines. The rebuttal provided very detailed analyses of these two hyperparameters. This concern should have been addressed. The AC predicts that the reviewer may keep the positive score of 6 or raise it to 8.

Reviewer sxcL (score: 4) raised several concerns: weak connection between "softmax overconfidence" and "inefficient conformal prediction", lack of analysis on "monotonic transformations", missing experiments (e.g., conditional coverage, significant tests), and overgeneralized conclusion of "softmax is not enough". The rebuttal provided detailed responses that addressed some of the concerns as indicated in the reviewer's early reply. The lingering concerns were: unclear explanation on "how the conformal quantile is affected or why the prediction sets become inefficient", insufficiency in Proposition 2.3, lack of evidence to prove that the proposed approach is more adaptive than the baseline, and the question on whether the gain comes from the energy formulation or rescaling of logits. Further rebuttal was posted but the reviewer was unable to respond due to the system shutdown. After examining the rebuttal thoroughly, the AC finds that the responses are mostly reasonable. The AC predicts that the reviewer is likely to raise the score to 6 based on the rebuttal.

The final predicted scores are 6/8, 6/8, and 6. The AC concludes that the paper's strengths outweigh the weaknesses and therefore the paper should be accepted.

**Reviewer Concerns:**

Reviewer Xpr8's concerns about adaptivity measures, epistemic uncertainty, and technical details were well addressed in the rebuttal. The main argument may lie in the idea of reweighting as the reviewer wanted to see a comparison with "Entropy Reweighted Conformal Classification". The rebuttal explained that entropy-based reweighting may suffer from gradient saturation while the proposed energy-based reweighting acts as a non-saturating proxy. The AC finds the explanation clear and solid. All concerns from this reviewer should have been addressed. Reviewer rMqU only requested more results and discussions on the two hyperparameters, which have also been provided in the rebuttal. The AC believes this reviewer would be satisfactory with the rebuttal. Both reviewers Xpr8 and rMqU gave a positive score in the first round. The AC predicts that they will keep positive after rebuttal and may raise the score from 6 (borderline accept) to 8 (accept). Reviewer sxcL seems to be the most negative reviewer. As indicated in the reviewer's early reply, the concerns are mostly addressed in the rebuttal except some lingering questions about the mechanism, theory, and source of improvement. The follow-up rebuttal was very detailed: the response clarified that energy modulation effectively reduces prediction set sizes by leveraging logit magnitudes to apply stricter thresholds for "easy" samples, thus addressing the reviewer’s questions about the formal mechanism and adaptiveness; the response distinguished the proposed method from temporal scaling by emphasizing its focus on differentiating sample difficulty and provided statistical tests to justify the method's gains. The AC finds the rebuttal clear and solid and believes the main concerns should have been addressed.

**Reviewer Scores:**

As discussed above, reviewers Xpr8 and rMqU are more likely to raise the score to 8 than keeping it at 6. Reviewer sxcL is likely to raise the score from 4 to 6. So the final predicted scores would be 6/8, 6/8, and 6.

---

### Decision · Program_Chairs · 2026-01-26

Accept (Poster)